# Fair Classification with Adversarial Perturbations

**L. Elisa Celis**
Yale University

**Anay Mehrotra**
Yale University

**Nisheeth K. Vishnoi**
Yale University

## Abstract

We study fair classification in the presence of an omniscient adversary that, given an $\eta$, is allowed to choose an arbitrary $\eta$-fraction of the training samples and arbitrarily perturb their protected attributes. The motivation comes from settings in which protected attributes can be incorrect due to strategic misreporting, malicious actors, or errors in imputation; and prior approaches that make stochastic or independence assumptions on errors may not satisfy their guarantees in this adversarial setting. Our main contribution is an optimization framework to learn fair classifiers in this adversarial setting that comes with provable guarantees on accuracy and fairness. Our framework works with multiple and non-binary protected attributes, is designed for the large class of linear-fractional fairness metrics, and can also handle perturbations besides protected attributes. We prove near-tightness of our framework's guarantees for natural hypothesis classes: no algorithm can have significantly better accuracy and any algorithm with better fairness must have lower accuracy. Empirically, we evaluate the classifiers produced by our framework for statistical rate on real-world and synthetic datasets for a family of adversaries.

## 1 Introduction

It is increasingly common to deploy classifiers to assist in decision-making in applications such as criminal recidivism [50], credit lending [21], and predictive policing [34]. Hence, it is imperative to ensure that these classifiers are fair with respect to protected attributes such as gender and race. Consequently, there has been extensive work on approaches for fair classification [32, 26, 28, 17, 63, 62, 48, 24, 27, 1, 13]. At a high level, a classifier $f$ is said to be "fair" with respect to a protected attribute $Z$ if it has a similar "performance" with respect to a given metric on different protected groups defined by $Z$. Given a fairness metric and a hypothesis class $\mathcal{F}$, fair classification frameworks consider the problem of finding a classifier $f^\star \in \mathcal{F}$ that maximizes accuracy constrained to being fair with respect to the given fairness metric (and $Z$) [8]. To specify fairness constraints, these approaches need protected attributes of training data to be known.

However, protected attributes can be erroneous for various reasons; there could be uncertainties during data collection or data cleaning process [20, 52], or the attributes could be strategically misreported [46]. Further, protected attributes may be missing entirely, as is often the case for racial and ethnic information in healthcare [20] or when data is scraped from the internet as with many image datasets [22, 66, 35]. In these cases, protected attributes can be "imputed" [18, 36, 16], but this can also introduce errors [12]; further, imputation by machine-learning-based methods is known to be fragile to imperceptible changes in the inputs [29] and to have correlated errors across samples [49]. Perturbations in protected attributes, regardless of origin, have been shown to have adverse effects on fair classifiers, affecting their performance on both accuracy and fairness metrics; see e.g., [16, 7].

Towards addressing this problem, several recent works have developed fair classification algorithms for various models of errors in the protected attributes. [44] consider an extension of the "mutually contaminated learning model" [53] where, instead of observing samples from the "true" joint distribution, distributions of observed group-conditional distributions are stochastic mixtures of their true counterparts. [6] consider a binary protected attribute and Bernoulli perturbations that are

35th Conference on Neural Information Processing Systems (NeurIPS 2021).

independent of the labels (and of each other). [14] consider the setting where each sample's protected attribute is independently flipped to a different value with a known probability. [59] considers two approaches to deal with perturbations. In their "soft-weights" approach, they assume perturbations follow a fixed distribution and one has access to an auxiliary data containing independent draws of both the true and perturbed protected attributes. In their distributionally robust approach, for each protected group, its feature and label distributions in the true data and the perturbed data are a known total variation distance away from each other. Finally, in an independent work, [40] study fair classification under the Malicious noise model [56, 39] in which a fraction of the training samples are chosen uniformly at random, and can then be perturbed arbitrarily.

**Our perturbation model.** We extend this line of work by studying fair classification under the following worst-case adversarial perturbation model: Given an $\eta > 0$, after the training samples are independently drawn from a true distribution $\mathcal{D}$, the adversary with unbounded computation power sees all the samples and can use this information to choose any $\eta$-fraction of the samples and perturb their protected attributes arbitrarily. This model is a straightforward adaptation of the perturbation model of [31] to the fair classification setting and we refer to it as the $\eta$-Hamming model. Unlike perturbation models studied before, this model can capture settings where the perturbations are strategic or arbitrarily correlated as can arise in the data collection stage or during imputation of the protected attributes, and in which the errors cannot be "estimated" using auxiliary data. In fact, under this perturbation model, the classifiers outputted by prior works can violate the fairness constraints by a large amount or have an accuracy that is significantly lower than the accuracy of $f^\star$; see Section 5 and Supplementary Material D.2. Taking these perturbed samples, a fairness metric $\Omega$, and a desired fairness threshold $\tau$ as input, the goal is to learn a classifier $f$ with the maximum accuracy with respect to the true distribution $\mathcal{D}$ subject to having a fairness value, $\Omega_{\mathcal{D}}(f)$, of at least $\tau$ with respect to the true distribution $\mathcal{D}$.

**Our contributions.** We present an optimization framework (Definition 4.1) that outputs fair classifiers for the $\eta$-Hamming model and comes with provable guarantees on accuracy and fairness (Theorem 4.3). Our framework works for multiple and non-binary protected attributes, and the large class of linear-fractional fairness metrics (that capture most fairness metrics studied in the literature); see Definition 3.1 and [13]. The framework provably outputs a classifier whose accuracy is within $2\eta$ of the accuracy of $f^\star$ and which violates the fairness constraint by at most $O(\eta/\lambda)$ additively (Theorem 4.3), under the mild assumption that the "performance" of $f^\star$ on each protected group is larger than a known constant $\lambda > 0$ (Assumption 1). Assumption 1 is drawn from the work of [14] for fair classification with stochastic perturbations. While it is not clear if the assumption is necessary in their model, we show that Assumption 1 is necessary for fair classification in the $\eta$-Hamming model: If $\lambda$ is not bounded away from 0, then no algorithm can give a non-trivial guarantee on *both* accuracy and fairness value of the output classifier (Theorem 4.4). Moreover, we prove the near-tightness of our framework's guarantee under Assumption 1: No algorithm can guarantee to output a classifier with accuracy closer than $\eta$ to that of $f^\star$ and any algorithm that violates the fairness constraint by less than $\eta/(20\lambda)$ additively has an accuracy at most $19/20$ (Theorems 4.5 and A.21). Finally, we also extend our framework's guarantees to the Nasty Sample Noise model (Supplementary Material A.1.5). The Nasty Sample Noise model is a generalization of the $\eta$-Hamming model, which was studied by [11] in the context of PAC learning (without any fairness considerations), where the adversary can choose any $\eta$-fraction of the samples, and can arbitrarily perturb both their labels and features.

We implement our framework for logistic loss function with linear classifiers and evaluate its performance on COMPAS [3], Adult [23], and a synthetic dataset (Section 5). We generate perturbations of these datasets admissible in the $\eta$-Hamming model and compare the performance of our approach to several baselines [44, 6, 59, 14, 40] with statistical rate and false-positive rate as fairness metrics.[1] On the synthetic dataset, we compare against a method developed for fair classification under stochastic perturbations [14] and demonstrate the comparative strength of the $\eta$-Hamming model; our results show that [14]'s framework achieves a significantly lower accuracy than our framework for the same statistical rate. Empirical results on COMPAS and Adult show that the classifier output by our framework can attain better statistical rate and false-positive rate than the accuracy maximizing classifier on the true distribution, with a small loss in accuracy. Further, our framework has a similar (or better) fairness-accuracy trade-off compared to all baselines we consider in a variety of settings, and is not dominated by any other approach (Figure 1 and Figures 7 and 8 in Supplementary Material E.2).

---

[1]Let $q_\ell(f, \mathrm{SR})$ (respectively $q_\ell(f, \mathrm{FPR})$) be the fraction of positive predictions (respectively false-positive predictions) by $f$ in the $\ell$-th protected group. $f$'s statistical rate (respectively false-positive rate) is the ratio of the minimum value to the maximum value of $q_\ell(f, \mathrm{SR})$ (respectively $q_\ell(f, \mathrm{FPR})$) over all protected groups.

**Techniques.** The starting point of our optimization framework (Definition 4.1) is the "standard" optimization program for fair classification in the *absence* of any perturbations: Given a fairness metric $\Omega$ and a desired fairness threshold $\tau$ as input, find $f^\star \in \mathcal{F}$ that maximizes the accuracy on the *given data* $\widehat{S}$ constrained to a fairness value at least $\tau$ on the given data. However, when $\widehat{S}$ is given to us by an $\eta$-Hamming adversary, this standard program, which imposes the fairness constraints with respect to the perturbed data $\widehat{S}$, may output a classifier with an accuracy/fairness-value worse than that of $f^\star$ when measured with respect to $\mathcal{D}$. But, observe that the difference in accuracies of a classifier when measured with respect to the given data $\widehat{S}$ and data sampled from $\mathcal{D}$ is at most $\eta$. Thus, if $f^\star \in \mathcal{F}$ is feasible for the standard optimization program, this observation (used twice) implies that the accuracy of the output classifier measured with respect to $\mathcal{D}$ is within $2\eta$ of the accuracy of $f^\star$ measured with respect $\mathcal{D}$ (Equation (8)). However, without any modifications, the classifier output by the standard optimization program could still have a fairness value much lower than $\tau$ with respect to $\mathcal{D}$ (see Example A.27). To bypass this, we introduce the notion of $s$-stability that allows us to lower bound the fairness value of a classifier with respect to $\mathcal{D}$ given its fairness value on $\widehat{S}$. Roughly, $f \in \mathcal{F}$ is said to be $s$-stable with respect to a fairness metric if for any $\widehat{S}$ that is generated by an $\eta$-Hamming adversary, the ratio of fairness value of $f$ with respect to $\mathcal{D}$ and with respect to $\widehat{S}$ is between $s$ and $1/s$ (see Definition 4.7). It follows that any $s$-stable classifier that has fairness value $\tau' > 0$ with respect to $\widehat{S}$, has fairness value at least $s \cdot \tau'$ with respect to $\mathcal{D}$. Hence, an optimization program that ensures that all feasible classifiers are $s$-stable (for a suitable choice of $s$) and have fairness value at least $\tau' > 0$ with respect to $\widehat{S}$, comes with a guarantee that any feasible classifier has a fairness value at least $s \cdot \tau'$ (with respect to $\mathcal{D}$). If such an optimization program could further ensure that $f^\star$ is feasible for it, then by arguments presented above, the classifier output by this optimization program would satisfy required guarantees on both fairness and accuracy (Lemma 4.9). The issue is that, to directly enforce $s$-stability, one needs to compute the fairness values of classifiers with respect to $\mathcal{D}$, but this is not possible in the absence of samples from $\mathcal{D}$. We overcome this by present a "proxy" constraint on the classifier (Equation (5)) that involves only $\widehat{S}$ and ensures that any classifier that satisfies it is $s$-stable. Moreover, $f^\star$ satisfies this constraint under Assumption 1. Overall, modifying Program (2) to include this constraint (Equation (5)) with a suitable value of $s$, and setting an appropriate fairness threshold $\tau$ so that $f^\star$ remains feasible, leads us to our framework.

## 2 Related work

In this section, we situate this paper in relation to lines of work which also consider fair classification with perturbed protected attributes; additional related work (e.g., on fair classification in the absence of protected attributes) are presented in Supplementary Material D.1.

[44] give a framework which comes with provable guarantees on the accuracy and fairness value of output classifiers for a binary protected attribute and either statistical rate or equalized-odds fairness metrics. [6] identify conditions on the distribution of perturbations under which the post-processing algorithm of [32] improves the fairness value of the accuracy-maximizing classifier with respect to equalized-odds on the true distribution with a binary protected attribute. [59] consider a non-binary protected attribute. In their "soft-weights" approach, they give provable guarantees on the accuracy (with respect to $f^\star$) and fairness value of the output classifier *in expectation* and in their distributionally robust approach, they give provable guarantees on the fairness value of the output classifiers.[2] [14] give provable guarantees on the accuracy and fairness value of output classifiers for multiple non-binary protected attributes and the class of linear-fractional metrics. All of the aforementioned works [44, 6, 59, 14] consider stochastic perturbation models, which are weaker than the model considered in this paper. Further, compared to [44, 6], our approach (and that of [14]) can handle multiple categorical protected attributes and multiple linear-fractional metrics (which include statistical rate and can ensure equalized-odds constraints). Compared to [6, 59], our work (and those of [44, 14]) give provable guarantees on the accuracy (with respect to $f^\star$) and fairness value of output classifiers *with high probability*. In another related work, [40] give an algorithm for a binary protected attribute which, under the realizable assumption (i.e., assuming there exists a classifier with perfect accuracy), outputs a classifier with guarantees on accuracy and fairness value with respect to the true-positive rate fairness metric. They study the Malicious noise model, which can modify a uniformly randomly selected subset of samples arbitrarily; this is weaker than the Nasty Sample Noise model [11, 4], and hence, than the model considered in this paper. Further, our

---

[2]Supplementary Material D.2.3 gives an example where [59]'s distributionally robust approach outputs a classifier whose accuracy is arbitrarily close to $1/2$.

framework works without the realizable assumption (i.e., in the agnostic setting), can handle multiple and non-binary protected attributes, and can ensure fairness with respect to multiple linear-fractional metrics (which include true-positive rate).

Another line of work has studied PAC learning in the presence of adversarial (and stochastic) perturbations in the data, without considerations of fairness [39, 2, 11, 15, 5]; see also [4]. In particular, [11] study PAC learning (without fairness constraints) under the Nasty Sample Noise model. They use the empirical risk minimization framework (see, e.g., [54]) run on the perturbed samples to output a classifier. Our framework Program (ErrTolerant) finds empirical risk minimizing classifiers that satisfy fairness constraints on the perturbed data, and that are also "stable" for the given fairness metric. While both frameworks show that the accuracy of the respective output classifiers is within $2\eta$ of the respective optimal classifiers when the data is unperturbed, the optimal classifiers can be quite different. For instance, while [11]'s framework is guaranteed to output a classifier with high accuracy, it can perform poorly on fairness metrics; see Section 5 and Supplementary Material D.2.1.

# 3   Model

Let the data domain be $D := \mathcal{X} \times \{0, 1\} \times [p]$, where $\mathcal{X}$ is the set of non-protected features, $\{0, 1\}$ is the set of binary labels, and $[p]$ is the set of $p$ protected attributes. Let $\mathcal{D}$ be a distribution over $D$. Let $\mathcal{F} \subseteq \{0, 1\}^{\mathcal{X} \times [p]}$ be a hypothesis class of binary classifiers. For $f \in \mathcal{F}$, let $\mathrm{Err}_{\mathcal{D}}(f) := \Pr_{(X,Y,Z) \sim \mathcal{D}}[f(X, Z) \neq Y]$ denote $f$'s predictive error on draws from $\mathcal{D}$. In the vanilla classification problem, the learner $\mathcal{L}$'s goal is to find a classifier with minimum error: $\mathrm{argmin}_{f \in \mathcal{F}} \mathrm{Err}_{\mathcal{D}}(f)$. In the fair classification problem, the learner is restricted to pick classifiers that have a "similar performance" conditioned on $Z = \ell$ for all $\ell \in [p]$. We consider the following class of metrics.

**Definition 3.1 (Linear/linear-fractional metrics [13]).** *Given $f \in \mathcal{F}$ and two events $\mathcal{E}(f)$ and $\mathcal{E}'(f)$, that can depend on $f$, define the performance of $f$ on $Z = \ell$ ($\ell \in [p]$) as $q_\ell(f) := \Pr_{\mathcal{D}}[\mathcal{E}(f) \mid \mathcal{E}'(f), Z = \ell]$. If $\mathcal{E}'$ depends on $f$, then $q_\ell(f)$ is said to be linear-fractional, otherwise linear.*

Definition 3.1 captures most of the performance metrics considered in the literature. For instance, for $\mathcal{E} := (f = 1)$ and $\mathcal{E}' := \emptyset$, we get statistical rate (a linear metric).[3] For $\mathcal{E} := (f = 1)$ and $\mathcal{E}' := (Y = 0)$, we get false-positive rate (also a linear metric). For $\mathcal{E} := (Y = 0)$ and $\mathcal{E}' := (f = 1)$, we get false-discovery rate (a linear-fractional metric). Given a performance metric $q$, the corresponding fairness metric is defined as

$$\Omega_{\mathcal{D}}(f) := \frac{\min_{\ell \in [p]} q_\ell(f)}{\max_{\ell \in [p]} q_\ell(f)}. \tag{1}$$

When $\mathcal{D}$ is the empirical distribution over samples $S$, we use $\Omega(f, S)$ to denote $\Omega_{\mathcal{D}}(f)$. The goal of fair classification, given a fairness metric $\Omega$ and a threshold $\tau \in (0, 1]$, is to (approximately) solve:

$$\min_{f \in \mathcal{F}} \mathrm{Err}_{\mathcal{D}}(f) \quad \text{s.t.,} \quad \Omega_{\mathcal{D}}(f) \geq \tau. \tag{2}$$

If samples from $\mathcal{D}$ are available, then one could try to solve this program. However, as discussed in Section 1, we do not have access to the *true* protected attribute $Z$, but instead only see a perturbed version, $\widehat{Z} \in [p]$, generated by the following adversary.

$\eta$-**Hamming model.** Given an $\eta \in [0, 1]$, let $\mathcal{A}(\eta)$ denote the set of all adversaries in the $\eta$-Hamming model. Any adversary $A \in \mathcal{A}(\eta)$ is a randomized algorithm with *unbounded* computation resources that knows the true distribution $\mathcal{D}$ and the algorithm of the learner $\mathcal{L}$. In this model, the learner $\mathcal{L}$ queries $A$ for $N \in \mathbb{N}$ samples from $\mathcal{D}$ *exactly once*. On receiving the request, $A$ draws $N$ independent samples $S := \{(x_i, y_i, z_i)\}_{i \in [N]}$ from $\mathcal{D}$, then $A$ uses its knowledge of $\mathcal{D}$ and $\mathcal{L}$ to choose an arbitrary $\eta \cdot N$ samples ($\eta \in [0, 1]$) and perturb their protected attribute arbitrarily to generate $\widehat{S} := \{(x_i, y_i, \widehat{z}_i)\}_{i \in [N]}$. Finally, $A$ gives these perturbed samples $\widehat{S}$ to $\mathcal{L}$.

**Learning model.** Given $\widehat{S}$ and the $\eta$, the learner $\mathcal{L}$ would like to (approximately) solve Program (2).

**Definition 3.2 ($(\varepsilon, \nu)$-learning).** *Given bounds on error $\varepsilon \in (0, 1)$ and constraint violation $\nu \in (0, 1)$, a learner $\mathcal{L}$ is said to $(\varepsilon, \nu)$-learn a hypothesis class $\mathcal{F} \subseteq \{0, 1\}^{\mathcal{X} \times [p]}$ with perturbation rate $\eta \in [0, 1]$ and confidence $\delta \in (0, 1)$ if for all*

---

[3] We overload the notation $f$ to denote both the classifier as well as its prediction, and the terms, statistical rate and false-positive rate, to refer to both the linear/linear-fractional metric $q$ and the resulting fairness metric $\Omega$.

- *distributions $\mathcal{D}$ over $\mathcal{X} \times \{0, 1\} \times [p]$ and*
- *adversaries $A \in \mathcal{A}(\eta)$,*

*there exists a threshold $N_0(\varepsilon, \nu, \delta, \eta) \in \mathbb{N}$, such that with probability at least $1 - \delta$ over the draw of $N \geq N_0(\varepsilon, \nu, \delta, \eta)$ iid samples $S \sim \mathcal{D}$, given $\eta$ and the perturbed samples $\widehat{S} := A(S)$, $\mathcal{L}$ outputs $f \in \mathcal{F}$ that satisfies $\mathrm{Err}_{\mathcal{D}}(f) - \mathrm{Err}_{\mathcal{D}}(f^\star) \leq \varepsilon$ and $\Omega_{\mathcal{D}}(f) \geq \tau - \nu$, where $f^\star$ is the optimal solution of Program (2) (i.e., $f^\star := \arg\min_{f \in \mathcal{F}} \mathrm{Err}_{\mathcal{D}}(f)$, s.t., $\Omega_{\mathcal{D}}(f) \geq \tau$).*

Given a finite number of perturbed samples, Definition 3.2 requires the learner to output a classifier that violates the fairness constraints additively by at most $\nu$ and that has a predictive error at most $\varepsilon$ smaller than that of $f^\star$, with probability at least $1 - \delta$. Like PAC learning [56], for a given hypothesis class $\mathcal{F}$, Definition 3.2 requires the learner to succeed on all distributions $\mathcal{D}$.

**Problem 1** (**Fair classification with adversarial perturbations**). *Given a hypothesis class $\mathcal{F} \subseteq \{0, 1\}^{\mathcal{X} \times [p]}$, a fairness metric $\Omega$, a threshold $\tau \in [0, 1]$, a perturbation rate $\eta \in [0, 1]$, and perturbed samples $\widehat{S}$, the goal is to $(\varepsilon, \nu)$-learn $\mathcal{F}$ for small $\varepsilon, \nu \in (0, 1)$.*

## 4 Theoretical results

In this section, we present our results on learning fair classifiers under the $\eta$-Hamming model. Our optimization framework (Program (ErrTolerant)) is a careful modification of Program (2). The main difficulty is that, unlike Program (2), it only has access to the perturbed samples $\widehat{S}$, and the ratio of a classifier's fairness with respect to the true distribution $\mathcal{D}$ and with respect to $\widehat{S}$ can be arbitrarily small (see Example A.27 in Supplementary Material A.5). To overcome this, our framework ensures that all feasible classifiers are "stable" (Definition 4.7). Then, as mentioned in Section 1, imposing the fairness constraint on $\widehat{S}$ guarantees (approximate) fairness on the true distribution $\mathcal{D}$. The accuracy guarantee follows by ensuring that the optimal solution of Program (2), $f^\star \in \mathcal{F}$, is feasible for our framework. To ensure this, we require Assumption 1 that also appeared in [14].

**Assumption 1.** *There is a known constant $\lambda > 0$ such that $\min_{\ell \in [p]} \mathrm{Pr}_{\mathcal{D}}[\mathcal{E}(f^\star), \mathcal{E}'(f^\star), Z = \ell] \geq \lambda$.*

It can be shown that this assumption implies that $\lambda$ is also a lower bound on the performances $q_1(f^\star), \ldots, q_p(f^\star)$ that depend on $\mathcal{E}$ and $\mathcal{E}'$. We expect $\lambda$ to be a non-vanishing positive constant in applications. For example, if $q$ is statistical rate, the minority protected group makes at least 20% of the population (i.e., $\min_{\ell \in [p]} \mathrm{Pr}_{\mathcal{D}}[Z = \ell] \geq 0.2$), and for all $\ell \in [p]$, $\mathrm{Pr}[f^\star = 1 \mid Z = \ell] \geq 1/2$, then $\lambda \geq 0.1$. In practice, $\lambda$ is not known exactly, but it can be set based on the context (e.g., see Section 5 and [14]). We show that Assumption 1 is necessary for the $\eta$-Hamming model (see Theorem 4.4).

**Definition 4.1** (**Error-tolerant program**). *Given a fairness metric $\Omega$ and corresponding events $\mathcal{E}$ and $\mathcal{E}'$ (as in Definition 3.1), a perturbation rate $\eta \in [0, 1]$, and constants $\lambda, \Delta \in (0, 1]$, we define the error-tolerant program for perturbed samples $\widehat{S}$, whose empirical distribution is $\widehat{D}$, as*

$$\min_{f \in \mathcal{F}} \quad \mathrm{Err}_{\widehat{D}}(f), \tag{ErrTolerant} \tag{3}$$

$$\text{s.t.,} \quad \Omega(f, \widehat{S}) \geq \tau \cdot \left( \frac{1 - (\eta + \Delta)/\lambda}{1 + (\eta + \Delta)/\lambda} \right)^2, \tag{4}$$

$$\forall \ell \in [p], \; \mathrm{Pr}_{\widehat{D}}\left[ \mathcal{E}(f), \mathcal{E}'(f), \widehat{Z} = \ell \right] \geq \lambda - \eta - \Delta. \tag{5}$$

$\Delta$ acts as a relaxation parameter in Program (ErrTolerant), which can be fixed in terms of the other parameters; see Theorem 4.3. Equation (4) ensures all feasible classifiers satisfy fairness constraints with respect to the perturbed samples $\widehat{S}$. Equation (5) ensures that all feasible classifiers are $(1 - O(\eta/\lambda))$-stable (see Definition 4.7). As mentioned in Section 1, this suffices to ensure that all feasible classifiers are fair with respect to $S$. Finally, to ensure the accuracy guarantee the thresholds in the RHS of Equations (4) and (5) are carefully tuned to ensure that $f^\star$ is feasible for Program (ErrTolerant); see Lemma 4.9. We refer the reader to the proof overview of Theorem 4.3 at the end of this section for further discussion of Program (ErrTolerant).

Before presenting our result we require the definition of the Vapnik–Chervonenkis (VC) dimension.

**Definition 4.2.** *Given a finite set $A$, define the collection of subsets $\mathcal{F}_A := \{\{a \in A \mid f(a) = 1\} \mid f \in \mathcal{F}\}$. We say that $\mathcal{F}$ shatters a set $B$ if $|\mathcal{F}_B| = 2^{|B|}$. The VC dimension of $\mathcal{F}$, $\mathrm{VC}(\mathcal{F}) \in \mathbb{N}$, is the largest integer such that there exists a set $C$ of size $\mathrm{VC}(\mathcal{F})$ that is shattered by $\mathcal{F}$.*

Our first result bounds the accuracy and fairness of an optimal solution $f_{\mathrm{ET}}$ of Program (ErrTolerant) for any hypothesis class $\mathcal{F}$ with a finite VC dimension using $O(\mathrm{VC}(\mathcal{F}))$ samples.

**Theorem 4.3** (**Main result**). *Suppose Assumption 1 holds with constant $\lambda > 0$ and $\mathcal{F}$ has VC dimension $d \in \mathbb{N}$. Then, for all perturbation rates $\eta \in (0, \lambda/2)$, fairness thresholds $\tau \in (0, 1]$, bounds on error $\varepsilon > 2\eta$ and constraint violation $\nu > 8\eta\tau/(\lambda-2\eta)$, and confidence parameters $\delta \in (0, 1)$ with probability at least $1 - \delta$, the optimal solution $f_{\mathrm{ET}} \in \mathcal{F}$ of Program (ErrTolerant) with parameters $\eta$, $\lambda$, and $\Delta := O\left(\min\left\{\varepsilon - 2\eta, \nu - 8\eta\tau/(\lambda-2\eta), \lambda - 2\eta\right\}\right)$, and $N = \mathrm{poly}(d, 1/\Delta, \log(p/\delta))$ perturbed samples from the $\eta$-Hamming model satisfies $\mathrm{Err}_{\mathcal{D}}(f_{\mathrm{ET}}) - \mathrm{Err}_{\mathcal{D}}(f^\star) \leq \varepsilon$ and $\Omega_{\mathcal{D}}(f_{\mathrm{ET}}) \geq \tau - \nu$.*

Thus, Theorem 4.3 shows that any procedure that outputs $f_{\mathrm{ET}}$, given with a sufficiently large number of perturbed samples, $(\varepsilon, \nu)$-learns $\mathcal{F}$ for any $\varepsilon > 2\eta$ and $\nu = O((\eta \cdot \tau)/\lambda)$. Theorem 4.3 can be extended to provably satisfy multiple linear-fractional metrics (at the same time) and work for multiple non-binary protected attributes; see Theorem B.2 in Supplementary Material B.1. Moreover, Theorem 4.3 also holds for the Nasty Sample Noise model. The proof of this result is implicit in the proof of Theorem 4.3; we present the details in Supplementary Material A.1.5. Finally, Program (ErrTolerant) only requires an estimate of one parameter, $\lambda$. (Since $\eta$ is known, $\tau$ is fixed by the user, and $\Delta$ can be set in terms of the other parameters.) If for each $\ell \in [p]$, we also have estimates of $\lambda_\ell := \Pr_{\mathcal{D}}[\mathcal{E}(f^\star), \mathcal{E}'(f^\star), Z = \ell]$ and $\gamma_\ell := \Pr_{\mathcal{D}}[\mathcal{E}'(f^\star), Z = \ell]$, then we can use this information to "tighten" Program (ErrTolerant) to the following program:

$$\min_{f \in \mathcal{F}} \quad \mathrm{Err}_{\widehat{D}}(f), \qquad\qquad\qquad\qquad \text{(ErrTolerant+)} \quad (6)$$
$$\text{s.t.,} \quad \Omega(f, \widehat{S}) \geq \tau \cdot s,$$
$$\forall\, \ell \in [p], \; \Pr_{\widehat{D}}[\mathcal{E}(f), \mathcal{E}'(f), \widehat{Z} = \ell] \geq \lambda_\ell - \eta - \Delta.$$

where the scaling parameter $s \in [0, 1]$ is the solution of the following optimization program

$$\min_{\eta_1, \eta_2, \ldots, \eta_p \geq 0} \; \min_{\ell, k \in [p]} \frac{1 - \eta_\ell/\lambda_\ell}{1 + (\eta_k - \eta_\ell)/\gamma_\ell} \cdot \frac{1 + (\eta_\ell - \eta_k)/\gamma_k}{1 + \eta_\ell/\lambda_k}, \quad \text{s.t.,} \quad \sum_{\ell \in [p]} \eta_\ell \leq \eta + \Delta. \quad (7)$$

If the classifiers in $\mathcal{F}$ do not use the protected attributes for prediction, then we can show that Program (ErrTolerant+) has a fairness guarantee of $(1 - s) + 4\eta\tau/(\lambda-2\eta)$ (which is always smaller than $8\eta\tau/(\lambda-2\eta)$) and an accuracy guarantee of $2\eta$. We prove this result in Supplementary Material B.2. Thus, in applications where one can estimate $\lambda_1, \ldots, \lambda_p$ and $\gamma_1, \ldots, \gamma_p$, Program (ErrTolerant+) offers better fairness guarantee than Program (ErrTolerant) (up to constants).

The proof of Theorem 4.3 appears in Supplementary Material A.1.

As for computing $f_{\mathrm{ET}}$, note that in general, Program (ErrTolerant) is a nonconvex optimization problem. In our simulations, we use the standard solver SLSQP in SciPy [57] to heuristically find $f_{\mathrm{ET}}$; see Supplementary Material E.1. Theoretically, for any arbitrarily small $\alpha > 0$, the techniques from [13] can be used to find an $f \in \mathcal{F}$ that has the optimal objective value for Program (ErrTolerant) and that additively violates its fairness constraint (4) by at most $\alpha$ by solving a set of $O(1/(\lambda\alpha))$ convex programs; details appear in Supplementary Material C.

**Impossibility results.** We now present results complementing the guarantees of Theorem 4.3.

**Theorem 4.4** (**No algorithm can guarantee high accuracy *and* fairness without Assumption 1**). *For all perturbation rates $\eta \in (0, 1]$, thresholds $\tau \in (1/2, 1)$, confidence parameters $\delta \in [0, 1/2)$, and bounds on the error $\varepsilon \in [0, 1/2)$ and constraint violation $\nu \in [0, \tau - 1/2)$, if the fairness metric is statistical rate, then it is impossible to $(\varepsilon, \nu)$-learn any hypothesis class $\mathcal{F} \subseteq \{0, 1\}^{\mathcal{X} \times [p]}$ that shatters a set of 6 points of the form $\{x_A, x_B, x_C\} \times [2] \subseteq \mathcal{X} \times [p]$ for some distinct $x_A, x_B, x_C \in \mathcal{X}$.*

Suppose that $\tau = 0.8$, say to encode the 80% Then, Theorem 4.4 shows that for any $\eta > 0$, any $\mathcal{F}$ satisfying the condition in Theorem 4.4 is not $(\varepsilon, \nu)$-learnable for any $\varepsilon < 1/2$ and $\nu < \tau - 1/2 = 3/10$. Intuitively, the condition on $\mathcal{F}$ avoids "simple" hypothesis classes. It is similar to the conditions considered by works on PAC learning with adversarial perturbations [11, 39], and holds for common hypothesis classes such as decision-trees and SVMs (Remark A.28 in Supplementary Material A.5). Thus, even if $\eta$ is vanishingly small, without additional assumptions, any $\mathcal{F}$ satisfying mild assumptions is not $(\varepsilon, \nu)$-learnable for any $\varepsilon < 1/2$ and $\nu < 3/10$, justifying Assumption 1. The proof of Theorem 4.4 appears in Supplementary Material A.2.

**Theorem 4.5** (**Fairness guarantee of Theorem 4.3 is optimal up to a constant factor**). *For all perturbation rates $\eta \in (0, 1]$, confidence parameter $\delta \in [0, 1/2)$, and a (known) constant $\lambda \in (0, 1/4]$,*

*if the fairness metric is statistical rate and $\tau = 1$, then given the promise that Assumption 1 holds with constant $\lambda$, for any bounds $\varepsilon < 1/4 - 2\eta/5$ and $v < \eta/(10\lambda) \cdot (1 - 4\lambda) - O\left(\eta^2/\lambda^2\right)$ it is impossible to $(\varepsilon, \nu)$-learn any hypothesis class $\mathcal{F} \subseteq \{0,1\}^{\mathcal{X} \times [p]}$ that shatters a set of 10 points of the form $\{x_A, x_B, x_C, x_D, x_E\} \times [2] \subseteq \mathcal{X} \times [p]$ for some distinct $x_A, x_B, x_C, x_D, x_E \in \mathcal{X}$.*

Suppose that $\lambda < 1/8$ and $\eta < 1/2$, then Theorem 4.5 shows that for any $\eta > 0$, any learner $\mathcal{L}$ that has a fairness guarantee $\nu < \eta/(20\lambda) - O(\eta^2/\lambda^2)$, must have a poor error bound, of at least $\varepsilon \geq 1/4 - 2\eta/5 \geq 1/20$, to $(\varepsilon, \nu)$-learn any $\mathcal{F}$ that satisfies a mild assumption. When $\eta/\lambda$ is small, this shows that any learner with a fairness guarantee $\nu = o(\eta/\lambda)$ must have an error guarantee at least $1/4 - 2\eta/5 \gg 2\eta$. Thus, Theorem 4.5 shows that one cannot improve the fairness guarantee in Theorem 4.3 by more than a constant amount without deteriorating the error guarantee from $2\eta$ to $1/4 - 2\eta/5$. Like Theorem 4.4, the condition on $\mathcal{F}$ in Theorem 4.5 avoids "simple" hypothesis classes and holds for common hypothesis (Remark A.28 in Supplementary Material A.5). Finally, complementing our accuracy guarantee, we prove that for any $\varepsilon < \eta$, no algorithm can $(\varepsilon, \nu)$-learn any hypothesis classes $\mathcal{F}$ satisfying mild assumptions (Theorem A.21 in Supplementary Material A.4); its proof appears in Supplementary Material A.4. Thus, the accuracy guarantee in Theorem 4.5 is optimal up to a factor of 2. The proof of Theorem 4.5 appears in Supplementary Material A.3.

**Proof overview of Theorem 4.3.** We explain the key ideas behind Program (ErrTolerant) and how they connect with the proof of Theorem 4.3. Our goal is to construct error-tolerant constraints using perturbed samples $\widehat{S}$ such that the classifier $f_{\mathrm{ET}}$, that has the smallest error on $\widehat{S}$ subject to satisfying these constraints, has accuracy $2\eta$-close to that of $f^\star$ and that additively violates the fairness constraints by at most $O(\eta/\lambda)$.

*Step 1: Lower bound on the accuracy of $f_{\mathrm{ET}}$.* This step relies on Lemma 4.6.

**Lemma 4.6.** *For any bounded function $g \colon \{0,1\}^2 \times [p] \to [0,1]$, $\delta, \Delta \in (0,1)$, and adversaries $A \in \mathcal{A}(\eta)$, given $N = \mathrm{poly}(1/\Delta, \mathrm{VC}(\mathcal{F}), \log 1/\delta)$ true samples $S \sim \mathcal{D}$ and corresponding perturbed samples $A(S) \coloneqq \{(x_i, y_i, \widehat{z}_i)\}_{i \in [N]}$, with probability at least $1 - \delta$, it holds that*

$$\forall f \in \mathcal{F}, \quad \left| \frac{1}{N} \sum_{i \in [N]} g(f(x_i, \widehat{z}_i), y_i, \widehat{z}_i) - \mathbb{E}_{(X,Y,Z) \sim \mathcal{D}} [g(f(X, Z), Y, Z)] \right| \leq \Delta + \eta.$$

The proof of Lemma 4.6 follows from generalization bounds for bounded functions (e.g., see [54]) and because the $\eta$-Hamming model perturbs at most $\eta \cdot N$ samples. Let $g$ be the 0-1 loss (i.e., $g(\widetilde{y}, y, z) \coloneqq \mathbb{I}[\widetilde{y} \neq y]$), then for all $f \in \mathcal{F}$, Lemma 4.6 shows that the error of $f$ on samples drawn from $\mathcal{D}$ and samples in $\widehat{S}$ are close: $|\mathrm{Err}_{\mathcal{D}}(f) - \mathrm{Err}(f, \widehat{S})| \leq \Delta + \eta$. Thus, intuitively, minimizing $\mathrm{Err}(f, \widehat{S})$ could be a good strategy to minimize $\mathrm{Err}_{\mathcal{D}}(f)$. Then, if $f^\star$ is feasible for Program (ErrTolerant), we can bound the error of $f_{\mathrm{ET}}$: Since $f_{\mathrm{ET}}$ is optimal for Program (ErrTolerant), its error on $\widehat{S}$ is at most the error of $f^\star$ on $\widehat{S}$. Using this and applying Lemma 4.6 we get that

$$\mathrm{Err}_{\mathcal{D}}(f_{\mathrm{ET}}) \leq \mathrm{Err}(f_{\mathrm{ET}}, \widehat{S}) + \eta + \Delta \ \leq \ \mathrm{Err}(f^\star, \widehat{S}) + \eta + \Delta \leq \mathrm{Err}_{\mathcal{D}}(f^\star) + 2(\eta + \Delta). \tag{8}$$

*Step 2: Lower bound on the fairness of $f_{\mathrm{ET}}$.* One could try to bound the fairness of $f_{\mathrm{ET}}$ using the same approach as Step 1, i.e., show that for all $f \in \mathcal{F}$: $|\Omega_{\mathcal{D}}(f) - \Omega(f, \widehat{S})| \leq O(\eta/\lambda)$. Then ensuring that $f$ has a high fairness on $\widehat{S}$ implies that it also has high fairness on $S$ (up to an $O(\eta/\lambda)$ factor). However, such a bound does not hold for any $\mathcal{F}$ satisfying mild assumptions (see Example A.27). The first idea is to prove a similar (in fact, stronger multiplicative) bound on a specifically chosen *subset* of $\mathcal{F}$ (consisting of "stable" classifiers). Toward this, we define:

**Definition 4.7.** *A classifier $f \in \mathcal{F}$ is said to be $s$-stable for fairness metric $\Omega$, if for all adversaries $A \in \mathcal{A}(\eta)$ and confidence parameters $\delta \in (0,1)$, given $\mathrm{polylog}(1/\delta)$ samples $S \sim \mathcal{D}$, with probability at least $1 - \delta$, it holds that $\Omega_{\mathcal{D}}(f)/\Omega(f, \widehat{S}) \in [s, 1/s]$, where $\widehat{S} \coloneqq A(S)$.*

If an $s$-stable classifier $f$ has fairness $\tau$ on $\widehat{S}$, then it has a fairness at least $\tau \cdot s$ on $\mathcal{D}$ with high probability. Thus, if we have a condition such that any feasible $f \in \mathcal{F}$ satisfying this condition is $s$-stable, then any classifier satisfying this condition and the fairness constraint, $\Omega(\cdot, \widehat{S}) \geq \tau/s$, must have a fairness at least $\tau$ on $\mathcal{D}$ with high probability. The key idea is coming up such constraints.

**Lemma 4.8.** *Any classifier $f \in \mathcal{F}$ that satisfies $\min_{\ell \in [p]} \mathrm{Pr}_{\mathcal{D}} [\mathcal{E}(f), \mathcal{E}'(f), \widehat{Z} = \ell] \geq \lambda + \eta + \Delta$, is $\left(\frac{1 - (\eta + \Delta)/\lambda}{1 + (\eta + \Delta)/\lambda}\right)^2$-stable for fairness metric $\Omega$ (defined by events $\mathcal{E}$ and $\mathcal{E}'$).*

*Step 3: Requirements for the error-tolerant program.* Building on Steps 1 and 2, we prove:

**Lemma 4.9.** *If the following conditions hold then, $\mathrm{Err}_{\mathcal{D}}(f_{\mathrm{ET}}) - \mathrm{Err}_{\mathcal{D}}(f^\star) \leq 2\eta$ and $\Omega_{\mathcal{D}}(f_{\mathrm{ET}}) \geq \tau - O(\eta/\lambda)$: (C1) $f^\star$ is feasible for Program (ErrTolerant), and all $f \in \mathcal{F}$ feasible for Program (ErrTolerant) are (C2) $s$-stable for $s = 1 - O(\eta/\lambda)$, and (C3) satisfy $\Omega(f, \widehat{S}) \geq \tau \cdot (1 - O(\eta/\lambda))$.*

Thus, it suffices to find error-tolerant constraints that satisfy conditions (C1) to (C3). Condition (C3) can be satisfied by adding the constraint $\Omega(\cdot, \widehat{S}) \geq \tau'$, for $\tau' = \tau \cdot (1 - O(\eta/\lambda))$. From Lemma 4.8, condition (C2) follows by using the constraint in $\min_{\ell \in [p]} \Pr_{\mathcal{D}} [\mathcal{E}(f), \mathcal{E}'(f), \widehat{Z} = \ell] \geq \lambda'$, for $\lambda' \geq \Theta(\lambda)$. It remains to pick $\tau'$ and $\lambda'$ such that condition (C1) also holds. The tension in setting $\tau'$ and $\lambda'$ is that if they are too large then condition (C1) does not hold, and if they are too small, then conditions (C2) and (C3) do not hold. In the proof we show that $\tau' := \tau \cdot (\frac{1-(\eta+\Delta)/\lambda}{1+(\eta+\Delta)/\lambda})^2$ and $\lambda' := \lambda - \eta - \Delta$ suffice to satisfy conditions (C1) to (C3) (this is where we use Assumption 1).

Overall the main technical idea is to identify the notion of $s$-stable classifiers and sufficient conditions for a classifier to be $s$-stable; combining these conditions with the fairness constraints on $\widehat{S}$, ensures that $f_{\mathrm{ET}}$ has high fairness on $S$, and carefully tuning the thresholds so that $f^\star$ is likely to be feasible for Program (ErrTolerant) ensures that $f_{\mathrm{ET}}$ has an accuracy close to $f^\star$.

**Proof overviews of Theorems 4.4 and 4.5.** Our proofs are inspired by [39, Theorem 1] and [11, Theorem 1] which consider PAC learning with adversarial corruptions. In both Theorems 4.4 and 4.5, for some $\varepsilon, \nu \in [0, 1]$, the goal is to show that given samples perturbed by an $\eta$-Hamming adversary, under some additional assumptions, no learner $\mathcal{L}$ can output a classifier that has accuracy $\varepsilon$-close to the accuracy of $f^\star$ and that additively violates the fairness constraints by at most $\nu$. Say a classifier $f \in \mathcal{F}$ is "good" if it satisfies these required guarantees. The approach is to construct two or more distributions $\mathcal{D}_1, \mathcal{D}_2, \ldots, \mathcal{D}_m$ that satisfy the following conditions: (C1) For any $\ell, k$, given a iid draw $S$ from $\mathcal{D}_\ell$, an $\eta$-Hamming adversary can add perturbations such that with high probability $\widehat{S}$ is distributed according to iid samples from $\mathcal{D}_k$. Thus $\mathcal{L}$, who only sees $\widehat{S}$, with high probability, cannot identify the original distribution of $S$ and is forced to output a classifier that is good for all $\mathcal{D}_1, \ldots, \mathcal{D}_m$. The next condition ensures that this is not possible. (C2) No classifier $f \in \mathcal{F}$ is good for all $\mathcal{D}_1, \ldots, \mathcal{D}_m$, and for each $\mathcal{D}_i$ ($i \in [m]$), there is at least one good classifier $f_i \in \mathcal{F}$. (The latter-half ensures that the fairness and accuracy requirements are not vacuously satisfied.) Thus, for every $\mathcal{L}$ there is a distribution in $\mathcal{D}_1, \ldots, \mathcal{D}_m$ for which $\mathcal{L}$ outputs a bad classifier. (Note that even if the learner is randomized, it must fail with probability at least $1/m$.) Finally, the assumptions on $\mathcal{F}$ ensure that condition (C2) is satisfiable. For instance, if $\mathcal{F}$ has less than $m$ hypothesis, then condition (C2) cannot be satisfied.

The key idea in the proofs is to come up with distributions satisfying the above conditions. [39, 11] follow the same outline in the context of PAC learning, however, as we also consider fairness constraints, our constructions end up being very different from their constructions. Our constructions are specific to the statistical rate fairness metric. However, one can still apply the general approach outlined above to other fairness metrics by constructing a suitable set of distributions. Full details appear in Supplementary Materials A.2 and A.3.

## 5 Empirical results

We implement our approach using the logistic loss function with linear classifiers and evaluate its performance on real world and synthetic data.

**Metrics and baselines.** The selection of an appropriate fairness metric is context-dependent and beyond the scope of this work [55]; for illustrative purposes we (arbitrarily) consider the statistical rate (SR) and compare an implementation of our framework (Program (ErrTolerant+)), **Err-Tol,** with state-of-the-art fair classification frameworks for statistical rate under stochastic perturbations: **LMZV** [44] and **CHKV** [14]. **LMZV** and **CHKV** take parameters $\delta_L, \tau \in [0, 1]$ as input; these parameters control the desired fairness, where decreasing $\delta_L$ or increasing $\tau$ increases the desired fairness. We also compare against **KL** [40], which controls for true-positive rate (TPR) in the presence of a Malicious adversary, and **AKM** [6] that is the post-processing method of [32] and controls for equalized-odds fairness constraints. We also compare against the optimal unconstrained classifier, **Uncons;** this is the same as [11]'s algorithm for PAC-learning in the Nasty Sample Noise Model without fairness constraints. We provide additional comparisons using our framework with false-positive rate as the fairness metric with additional baselines and using the Adult data [23] in Supplementary Material E.

**Implementation details.** We use a randomly generated 70-30 train ($S$) test ($T$) split of the data, and generate the perturbed data $\widehat{S}$ from $S$ for a (known) perturbation rate $\eta$. We train each algorithm on $\widehat{S}$, and report the accuracy (acc) and statistical rate (SR) of the output classifiers on the (unperturbed) test data $T$. **Err-Tol** is given the perturbation rate $\eta$ and uses the SLSQP solver in SciPy [57] to solve Program (ErrTolerant+). To advantage the baselines in our comparison, we provide them with even more information as needed by their approaches: **LMZV** and **CHKV** are given group-specific pertur-

Table 1: *Simulation on synthetic data:* We run **CHKV** and **Err-Tol** with $\tau = 0.8$ on synthetic data and report their average accuracy (acc) and statistical rate (SR) with standard deviation in parentheses. The result shows that prior approaches can fail to satisfy their guarantees under the $\eta$-Hamming model.

| | acc ($\eta$=0%) | SR ($\eta$=0%) | acc ($\eta$=3%) | SR ($\eta$=3%) | acc ($\eta$=5%) | SR ($\eta$=5%) |
|---|---|---|---|---|---|---|
| **Unconstrained** | 1.00 (.001) | .799 (.001) | 1.00 (.000) | .799 (.002) | 1.00 (.001) | .800 (.001) |
| **CHKV** ($\tau$=.8) | 1.00 (.001) | .800 (.002) | .859 (.143) | .787 (.015) | .799 (.139) | .795 (.049) |
| **Err-Tol** ($\tau$=.8) | .985 (.065) | .800 (.001) | 1.00 (.001) | .799 (.002) | .999 (.002) | .799 (.004) |

bation rates: for each $\ell \in [p]$, $\eta_\ell := \Pr_D[\widehat{Z} \neq Z \mid Z = \ell]$, and **KL** is given $\eta$ and for each $\ell \in [p]$, the probability $\Pr_D[Z = \ell, Y = 1]$; where $D$ is the empirical distribution of $S$. **Err-Tol** implements Program (ErrTolerant+) which requires estimates of $\lambda_\ell$ and $\gamma_\ell$ for all $\ell \in [p]$. As a heuristic, we set $\gamma_\ell = \lambda_\ell := \Pr_{\widehat{D}}[Z = \ell]$, where $\widehat{D}$ is the empirical distribution of $\widehat{S}$. We find that these estimates suffice, and expect that a more refined approach would only improve the performance of **Err-Tol.**

**Adversaries.** We consider two $\eta$-Hamming adversaries (which we call $A_{\text{TN}}$ and $A_{\text{FN}}$); each one computes the "optimal fair classifier" $f^\star$, which has the highest accuracy (on $S$) subject to having statistical rate at least $\tau$ on $S$. $A_{\text{TN}}$ considers the set of all true negatives of $f^\star$ that have protected attribute $Z = 1$, selects the $\eta \cdot |S|$ samples that are furthest from the decision boundary of $f^\star$, and perturbs their protected attribute to $\widehat{Z} = 2$. $A_{\text{FN}}$ is similar, except that it considers the set of false negatives of $f^\star$. Both adversaries try to increase the performance of $f^\star$ on $Z = 1$ in $\widehat{S}$ by removing the samples that $f^\star$ predicts as negative; thus, increasing $f^\star$'s statistical rate. The adversary's hope is that choosing samples far from the decision boundary would (falsely) give the appearance of a high statistical rate on $\widehat{S}$. This would make a fair classification framework output unfair classifiers with higher accuracy. Note that these are not intended to be "worst-case" adversaries; as **Err-Tol** comes with provable guarantees, we expect it to perform well against other adversaries while other approaches may have even poorer performance.

**Simulation on synthetic data.** We first show empirically that perturbations by the $\eta$-Hamming adversary can be prohibitively disruptive for methods that attempt to correct for stochastic noise. We consider synthetic data with 1,000 samples from two equally-sized protected groups; each sample has a binary protected attribute, two continuous features $x_1, x_2 \in \mathbb{R}$, and a binary label. Conditioned on the protected attribute, $(x_1, x_2)$ are independent draws from a mixture of 2D Gaussians (see Figure 4). This distribution and the labels are such that a) one group has a higher likelihood of a positive label than the other, and b) **Uncons** has a near-perfect accuracy ($> 99\%$) and a statistical rate of 0.8 on $S$. Similar to **Uncons,** we consider a fairness constraint of $\tau = 0.8$. Thus, in the absence of noise, this is an "easy case:" where **Uncons** satisfies the fairness constraints. We generate $\widehat{S}$ using $A_{\text{TN}}$, and compare against **CHKV,** which was developed for correcting stochastic perturbations.[4]

*Results.* The fairness and statistical rate averaged over 50 iterations are reported in Table 1 as a function of the perturbation $\eta$. At $\eta = 0$, both **CHKV** and **Err-Tol** nearly-satisfy the fairness constraint (SR $\geq 0.79$) and have a near-perfect accuracy (acc $\geq 0.98$). However, as $\eta$ increases, while **CHKV** retains the same statistical rate ($\sim 0.8$), it loses a significant amount of accuracy ($\sim 20\%$). In contrast, **Err-Tol** has high accuracy and fairness (acc $\geq 0.99$ and SR $\geq 0.79$) for all $\eta$ considered. Hence, this shows that stochastic approaches may fail to satisfy their guarantees under the $\eta$-Hamming model.

**Simulations on real-world data.** In this simulation, we show that our framework can outperform each baseline with respect to the accuracy-fairness trade-off under perturbations from the adversaries we consider, and does not under-perform compared to baselines under perturbations from either adversary. The COMPAS data in [9] contains 6,172 samples with 10 binary features and a label that is 1 if the individual did not recidivate and 0 otherwise; the statistical rate of **Uncons** on COMPAS is 0.78. We take gender (coded as binary) as the protected attribute, and set the fairness constraint on the statistical rate to be $\tau = 0.9$ for **Err-Tol** and all baselines. We consider both adversaries $A_{\text{TN}}$ and $A_{\text{FN}}$, and a perturbation rate of $\eta = 3.5\%$, as 3.5% is roughly the smallest value for $\eta$ necessary to ensure that the optimal fair classifier $f^\star$ for $\tau = 0.9$ (on $S$) has a statistical rate less than 0.78 on $\widehat{S}$.

*Results.* The accuracy and statistical rate (SR) of **Err-Tol** and baselines for $\tau \in [0.7, 1]$ and $\delta_L \in [0, 0.1]$ and averaged over 100 iterations are reported in Figure 1. For both adversaries, **Err-Tol** attains a better statistical rate than the unconstrained classifier (**Uncons**) for a small trade-off in accuracy. For adversary $A_{\text{TN}}$ (Figure 1(a)), **Uncons** has statistical rate (0.80) and accuracy (0.67). In contrast,

---

[4]We also attempted to compare against **AKM**, **KL**, and **LMZV**. But they did not converge to $f^\star$ even on the unperturbed synthetic data, and hence, we did not include these results as it would be an unfair comparison.

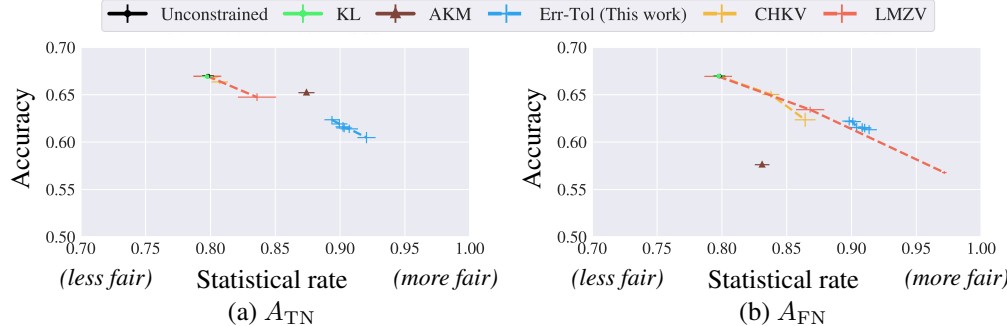

Figure 1: *Simulations on COMPAS data:* Perturbed data is generated using adversary $A_{\text{TN}}$ (a) and $A_{\text{FN}}$ (b) as described in Section 5 with $\eta = 3.5\%$. All algorithms are run on the perturbed data varying the fairness parameters ($\tau \in [0.7, 1]$ and $\delta_L \in [0, 0.1]$). The $y$-axis depicts accuracy and the $x$-axis depicts statistical rate (SR); both values are computed over the unperturbed test set. We observe that for both adversaries our approach **Err-Tol,** attains a better fairness than the unconstrained classifier with a natural trade-off in accuracy. Further, **Err-Tol** achieves a better fairness-accuracy trade-off than each baseline on at least one of (a) or (b). Error bars represent the standard error of the mean.

**Err-Tol** achieves high statistical rate (0.92) with a trade-off in accuracy (0.60). In comparison, **AKM** has a higher accuracy (0.65) but a lower statistical rate (0.87), and other baselines have an even lower statistical rate ($\leq 0.84$) with accuracy comparable to **AKM.** For adversary $A_{\text{FN}}$ (Figure 1(b)), **Uncons** has statistical rate (0.80) and accuracy (0.67), while **Err-Tol** has a significantly higher SR (0.91) and accuracy (0.61). This significantly outperforms **AKM** which has statistical rate (0.83) and accuracy (0.58). **LMZV** achieves the highest statistical rate (0.97) with a natural reduction in accuracy to (0.57). In this case, **Err-Tol** has similar accuracy to statistical rate trade-off as **LMZV,** but achieves a lower maximum statistical rate (0.91). Meanwhile, **Err-Tol** has a significantly higher statistical rate trade-off than **CHKV** at the same accuracy. We further evaluate our framework under stochastic perturbations in Supplementary Material E (specifically, against the perturbation model of [14]) and observe similar statistical rate and accuracy trade-offs as approaches [14, 44] tailored for stochastic perturbations.

**Remark 5.1** (**Range of fairness parameters in the simulation**). *Among baselines,* **AKM, KL,** *and* **Uncons** *do not take the desired-fairness value as input, so they appear as points in Figure 1. For all other methods (***CHKV, Err-Tol,** *and* **LMZV***), we vary the fairness parameters starting from the tightest constraints (i.e., $\tau = 1$ and $\delta_L = 0$) and relax the constraints until all algorithms' achieved statistical rate matches the achieved statistical rate of the unconstrained classifier (this happens around $\tau = 0.7$ and $\delta_L = 0.1$). We do not relax the fairness parameters further because the resulting problem is equivalent to the unconstrained classification problem. (This is because the unconstrained classifier, which has the highest accuracy, satisfies the fairness constraints for $\tau \leq 0.7$ and $\delta_L \geq 0.1$).*

## 6 Limitations and conclusion

This work extends fair classification to real-world settings where perturbations in the protected attributes may be correlated or affect arbitrary subsets of samples. We consider the $\eta$-Hamming model and give a framework that outputs classifiers with provable guarantees on both fairness and accuracy; this framework works for categorical protected attributes and the class of linear-fractional fairness metrics. We show near-tightness of our framework's guarantee and extend it to the Nasty Sample Noise model, which can perturb both labels and features. Empirically, classifiers produced by our framework achieve high fairness at a small cost to accuracy and outperform existing approaches.

Compared to existing frameworks for fair classification with stochastic perturbations, our framework requires less information about the perturbations. That said, in a few applications, e.g., the randomized response procedure [60], where the perturbations are independent across samples and identically distributed according to a *known* distribution, frameworks for fair classification with stochastic perturbations can perform better. Further, like existing frameworks, our framework's efficacy will depend on an appropriate choice of parameters; e.g., an overly conservative $\lambda$ can decrease accuracy and an optimistic $\lambda$ can decrease fairness. A careful assessment both pre- and post-deployment would be important to avoid negative social implications in a misguided attempt to do good [45].

Finally, we note that discrimination is a systematic problem and our work only addresses one part of it; this work would be effective as one piece of a broader approach to mitigate and rectify biases.

**Acknowledgements.** This research was supported in part by an NSF CAREER Award (IIS-2045951), a J.P. Morgan Faculty Award, and an AWS MLRA Award.

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
