# Contents

# A Proofs

## A.1 Proof of Theorem 4.3

Recall that we assume Assumption 1 holds with constant $\lambda > 0$ and the VC dimension of $\mathcal{F}$ is finite, say $d \in \mathbb{N}$. Our goal is to prove that for all perturbation rates $\eta \in (0, \lambda/2)$, fairness thresholds $\tau \in (0, 1]$, bounds on error $\varepsilon > 2\eta$ and bounds on constraint violation $\nu > 8\eta\tau/(\lambda-2\eta)$, and confidence parameters $\delta \in (0, 1)$, given sufficiently many perturbed samples from an $\eta$-Hamming adversary, with probability at least $1 - \delta$, it holds that

$$\mathrm{Err}_{\mathcal{D}}(f_{\mathrm{ET}}) - \mathrm{Err}_{\mathcal{D}}(f^{\star}) \leq \varepsilon,$$
$$\Omega_{\mathcal{D}}(f_{\mathrm{ET}}) \geq \tau - \nu,$$

where $f_{\mathrm{ET}}$ is an optimal solution of Program (ErrTolerant). In the proof, we set the following parameters

$$\Delta_0 := \min\left\{2\eta - \varepsilon, \frac{8\eta\tau}{\lambda - 2\eta} - \nu\right\} \quad \text{and} \quad \delta_0 := \frac{\delta}{2p + 1}. \tag{9}$$

We set $\Delta := \Delta_0 \cdot \frac{(\lambda-2\eta)^2}{32\cdot3}$ in Program (ErrTolerant) and require at least $N$ samples (perturbed by an $\eta$-Hamming adversary), where $N$ satisfies

$$N = \Theta\left(\frac{1}{\Delta^2 \cdot (\lambda - 2\eta)^4} \cdot \left(d\log\left(\frac{d}{\Delta^2 \cdot (\lambda - 2\eta)^4}\right) + \log\left(\frac{p}{\delta_0}\right)\right)\right). \tag{10}$$

Note that these values satisfy the requirements $\Delta = O(\min\{\varepsilon - 2\eta, \nu - 8\eta\tau/(\lambda-2\eta), \lambda - 2\eta\})$ and $N = \mathrm{poly}(d, 1/\Delta, \log(p/\delta))$.

**Remark A.1.** *All probabilities and expectations in all proofs are with respect to the draw of $(X, Y, Z)$. Given a distribution $\mathcal{P}$, we use $\mathrm{Pr}_{\mathcal{P}}[\cdot]$ to denote $\mathrm{Pr}_{(X,Y,Z)\sim\mathcal{P}}[\cdot]$ and $\mathbb{E}_{\mathcal{P}}[\cdot]$ to denote $\mathbb{E}_{(X,Y,Z)\sim\mathcal{P}}[\cdot]$. If $\mathcal{P}$ denotes the distribution of perturbed samples, then we use $\mathrm{Pr}_{\mathcal{P}}[\cdot]$ to denote $\mathrm{Pr}_{(X,Y,\widehat{Z})\sim\mathcal{P}}[\cdot]$ and $\mathbb{E}_{\mathcal{P}}[\cdot]$ to denote $\mathbb{E}_{(X,Y,\widehat{Z})\sim\mathcal{P}}[\cdot]$; The difference between the two will be clear from context.*

### A.1.1 Preliminaries: Generalization bound

We use Lemma A.2 in the proof of Theorem 4.3. See [54, Section 28.1] for a proof.

**Lemma A.2 (Concentration of mean of bounded functions).** *For any bounded function $g: \{0, 1\} \times \{0, 1\} \times [p] \to [0, 1]$ and constants $\Delta, \delta_0 \in (0, 1)$, given $N \geq \Theta\left(\Delta^{-2} \cdot \left(\mathrm{VC}(\mathcal{F}) \cdot \log\left(\mathrm{VC}(\mathcal{F})/\Delta\right) + \log\left(1/\delta_0\right)\right)\right)$ samples $S$ iid from $\mathcal{D}$, with probability at least $1 - \delta_0$, it holds that*

$$\forall f \in \mathcal{F}, \quad |\mathbb{E}_D[g(f(X, Z), Y, Z)] - \mathbb{E}_{\mathcal{D}}[g(f(X, Z), Y, Z)]| \leq \Delta,$$

*where $D$ is the empirical distribution of $S$.*

### A.1.2 Step 1: Lower bound on the accuracy of $f_{\mathrm{ET}}$

This step relies on Lemma A.3, which is the formal version of Lemma 4.6. We use Lemma A.3 in the proof of Lemma A.4 to lower bound the accuracy of $f_{\mathrm{ET}}$.

**Lemma A.3 (Bound on difference in means of bounded functions on $\mathcal{D}$ and on $\widehat{S}$).** *For any bounded function $g: \{0, 1\} \times \{0, 1\} \times [p] \to [0, 1]$ and constants $\Delta, \delta_0 \in (0, 1)$, given $N \geq \Theta\left(\Delta^{-2} \cdot \left(\mathrm{VC}(\mathcal{F}) \cdot \log\left(\mathrm{VC}(\mathcal{F})/\Delta\right) + \log\left(1/\delta_0\right)\right)\right)$ samples $S$ iid from $\mathcal{D}$, and corresponding perturbed samples $A(S) := \{(x_i, y_i, \widehat{z}_i)\}_{i\in[N]}$, with probability at least $1 - \delta$, it holds that*

$$\forall f \in \mathcal{F}, \quad \left|\mathbb{E}_{\widehat{D}}[g(f(X, \widehat{Z}), Y, \widehat{Z})] - \mathbb{E}_{\mathcal{D}}[g(f(X, Z), Y, Z)]\right| \leq \Delta + \eta,$$

*where $\widehat{D}$ is the empirical distribution of $\widehat{S}$.*

*Proof.* Let $S := \{(x_i, z_i, y_i)\}_{i\in[N]}$ and $\widehat{S} := \{(x_i, \widehat{z}_i, y_i)\}_{i\in[N]}$. Using the triangle inequality for absolute value, we have

$$\left|\mathbb{E}_{\widehat{D}}[g(f(X, \widehat{Z}), Y, \widehat{Z})] - \mathbb{E}_{\mathcal{D}}[g(f(X, Z), Y, Z)]\right|$$
$$\leq |\mathbb{E}_D[g(f(X, Z), Y, Z)] - \mathbb{E}_{\mathcal{D}}[g(f(X, Z), Y, Z)]|$$
$$+ \left|\mathbb{E}_{\widehat{D}}[g(f(X, \widehat{Z}), Y, \widehat{Z})] - \mathbb{E}_D[g(f(X, Z), Y, Z)]\right|. \tag{11}$$

We can upper bound the first term in the RHS using Lemma A.2. In particular, we have that with probability at least $1 - \delta$, for all $f \in \mathcal{F}$, it holds that

$$\left| \mathbb{E}_D \left[ g(f(X, Z), Y, Z) \right] - \mathbb{E}_{\mathcal{D}} \left[ g(f(X, Z), Y, Z) \right] \right| \leq \Delta. \tag{12}$$

Further, we can upper bound the second term in the RHS of Equation (11) for all $f \in \mathcal{F}$ as follows

$$\left| \mathbb{E}_{\widehat{D}} \left[ g(f(X, \widehat{Z}), Y, \widehat{Z}) \right] - \mathbb{E}_{\widehat{D}} \left[ g(f(X, \widehat{Z}), Y, \widehat{Z}) \right] \right|$$

$$= \frac{1}{N} \cdot \left| \sum_{i \in [N]} g(f(x_i, \widehat{z}_i), y_i, \widehat{z}_i) - g(f(x_i, z_i), y_i, z_i) \right|$$

$$= \frac{1}{N} \cdot \left| \sum_{i \in [N]: \; z_i \neq \widehat{z}_i} g(f(x_i, \widehat{z}_i), y_i, \widehat{z}_i) - g(f(x_i, z_i), y_i, z_i) \right|$$

(For all $i \in [N]$, where $z_i = \widehat{z}_i$, $g(f(x_i, \widehat{z}_i), y_i, \widehat{z}_i) = g(f(x_i, z_i), y_i, z_i)$.)

$$\leq \frac{1}{N} \cdot \left| \sum_{i \in [N]: \; z_i \neq \widehat{z}_i} 1 \right| \qquad \text{(Using that } g \text{ is bounded by 0 and 1)}$$

$$\leq \eta. \tag{13}$$

Since with probability at least $1 - \delta$, both Equation (12) and (13) hold for all $f \in \mathcal{F}$, substituting them in Equation (11) gives us the required bound. $\qquad \square$

**Lemma A.4.** *If $f^\star$ is feasible for Program (ErrTolerant), then it holds that:*

$$\mathrm{Err}_{\mathcal{D}}(f_{\mathrm{ET}}) - \mathrm{Err}_{\mathcal{D}}(f^\star) \leq 2\eta + \Delta.$$

*Proof.* Let $g$ be the 0-1 loss (i.e., $g(\widetilde{y}, y, z) \coloneqq \mathbb{I}\left[ \widetilde{y} \neq y \right]$), then for all $f \in \mathcal{F}$, Lemma A.3 shows that the error of $f$ on samples drawn from $\mathcal{D}$ and samples in $\widehat{S}$ are close: $|\mathrm{Err}_{\mathcal{D}}(f) - \mathrm{Err}(f, \widehat{S})| \leq \Delta + \eta$. Since $f_{\mathrm{ET}}$ is optimal for Program (ErrTolerant), its error on $\widehat{S}$ is at most the error of $f^\star$ on $\widehat{S}$. Using this and applying Lemma A.3 we get that

$$\mathrm{Err}_{\mathcal{D}}(f_{\mathrm{ET}}) \overset{\text{Lemma A.3}}{\leq} \mathrm{Err}(f_{\mathrm{ET}}, \widehat{S}) + \eta + \Delta$$

$$\leq \mathrm{Err}(f^\star, \widehat{S}) + \eta + \Delta$$

$$\overset{\text{Lemma A.3}}{\leq} \mathrm{Err}_{\mathcal{D}}(f^\star) + 2(\eta + \Delta).$$

$\qquad \square$

### A.1.3 Step 2: Lower bound on the fairness of $f_{\mathrm{ET}}$

In this step, we show that any $f \in \mathcal{F}$ feasible for Program (ErrTolerant) satisfies $\Omega_{\mathcal{D}}(f) \geq \tau - \nu$ (Lemma A.8). This relies on the notion of $s$-stability (Definition A.5) and the fact that any $f \in \mathcal{F}$ feasible for Program (ErrTolerant) is $s$-stable (for $s$ a function of $\eta$ and $\lambda$); Corollary A.7.

**Definition A.5** ($s$-stability)**.** *Given a constant $s \in (0, 1)$ and perturbation rate $\eta \in [0, 1)$, a classifier $f \in \mathcal{F}$ is said to be $s$-stable for fairness metric $\Omega$ with perturbation rate $\eta$, if for all adversaries $A \in \mathcal{A}(\eta)$ and confidence parameters $\delta_0 \in (0, 1)$ given $N = \mathrm{polylog}(1/\delta_0)$ samples $S$ iid from $\mathcal{D}$ and corresponding perturbed samples $\widehat{S} \coloneqq A(S)$, with probability at least $1 - \delta_0$ (over draw of $S \sim \mathcal{D}$), it holds that*

$$\frac{\Omega_{\mathcal{D}}(f)}{\Omega(f, \widehat{S})} \in \left[ s, \frac{1}{s} \right].$$

If an $s$-stable classifier $f$ has fairness $\tau$ on $\widehat{S}$, then it has a fairness at least $\tau \cdot s$ on $\mathcal{D}$ with high probability. Thus, if we have some constraint $C$ such that any feasible $f \in \mathcal{F}$ satisfying constraint $C$ is $s$-stable, then any classifier satisfying constraint $C$ and the fairness constraint, $\Omega(\cdot, \widehat{S}) \geq \tau/s$, must have a fairness at least $\tau$ on $\mathcal{D}$ with high probability. The key idea is coming up such a constraints. First, in Lemma A.6, we give such constraints which use the unperturbed protected attributes $Z$, later in Corollary A.7 we give constraints which only use the perturbed protected attributes $\widehat{Z}$.

**Lemma A.6** (**Sufficient condition for a stable classifier**)**.** *For each $\alpha \in (0, 1)$, any classifier $f \in \mathcal{F}$ satisfying*

$$\min_{\ell \in [p]} \mathrm{Pr}_{\mathcal{D}} \left[ \mathcal{E}(f), \mathcal{E}'(f), Z = \ell \right] \geq \alpha, \tag{14}$$

*is $\left( \frac{1 - (\eta + \Delta)/\alpha}{1 + (\eta + \Delta)/\alpha} \right)^2$-stable with respect to the fairness metric $\Omega$ defined by events $\mathcal{E}$ and $\mathcal{E}'$.*

*Proof of Lemma A.6.* Let $\widehat{D}$ be the empirical distribution of $\widehat{S}$. Let $\mathscr{J}$ be the event that for all $\ell \in [p]$ and all $f \in \mathcal{F}$

$$\left|\Pr_{\widehat{D}}\left[\mathcal{E}(f), \mathcal{E}'(f), \widehat{Z} = \ell\right] - \Pr_{\mathcal{D}}\left[\mathcal{E}(f), \mathcal{E}'(f), Z_i = \ell\right]\right| < \eta + \Delta, \tag{15}$$

$$\left|\Pr_{\widehat{D}}\left[\mathcal{E}'(f), \widehat{Z} = \ell\right] - \Pr_{\mathcal{D}}\left[\mathcal{E}'(f), Z_i = \ell\right]\right| < \eta + \Delta. \tag{16}$$

Using Lemma A.3 with $g(\widetilde{y}, y, z) \coloneqq \mathbb{I}\left[\mathcal{E}(\widetilde{y}), \mathcal{E}'(\widetilde{y}), z = \ell\right]$, we get that with probability at least $1 - \delta_0$, Equation (15) holds for $f \in \mathcal{F}$ and a particular $\ell \in [p]$. Similarly, using Lemma A.3 with $g(\widetilde{y}, y, z) \coloneqq \mathbb{I}\left[\mathcal{E}(\widetilde{y}), z = \ell\right]$, we get that with probability at least $1 - \delta_0$, Equation (16) holds for $f \in \mathcal{F}$ and a particular $\ell \in [p]$. Applying the union bound over all $\ell \in [p]$, we get that

$$\Pr[\mathscr{J}] \geq 1 - 2p\delta_0. \tag{17}$$

Suppose event $\mathscr{J}$ holds. Then, for any $\ell, k \in [p]$, we have

$$\frac{\Pr_{\widehat{D}}[\mathcal{E}(f), \mathcal{E}'(f), \widehat{Z} = \ell]}{\Pr_{\widehat{D}}[\mathcal{E}'(f), \widehat{Z} = \ell]} \cdot \frac{\Pr_{\widehat{D}}[\mathcal{E}'(f), \widehat{Z} = k]}{\Pr_{\widehat{D}}[\mathcal{E}(f), \mathcal{E}'(f), \widehat{Z} = k]}$$

$$\overset{(15),(16)}{\geq} \frac{\Pr_{\mathcal{D}}[\mathcal{E}(f), \mathcal{E}'(f), Z = \ell] - \eta - \Delta}{\Pr_{\mathcal{D}}[\mathcal{E}'(f), Z = \ell] + \eta + \Delta} \cdot \frac{\Pr_{\mathcal{D}}[\mathcal{E}'(f), Z = k] - \eta - \Delta}{\Pr_{\mathcal{D}}[\mathcal{E}(f), \mathcal{E}'(f), Z = k] + \eta + \Delta}. \tag{18}$$

Using Equation (14), we can lower bound the RHS of Equation (18) as follows

$$\frac{\Pr_{\mathcal{D}}[\mathcal{E}(f), \mathcal{E}'(f), Z = \ell] - \eta - \Delta}{\Pr_{\mathcal{D}}[\mathcal{E}'(f), Z = \ell] + \eta + \Delta} \cdot \frac{\Pr_{\mathcal{D}}[\mathcal{E}'(f), Z = k] - \eta - \Delta}{\Pr_{\mathcal{D}}[\mathcal{E}(f), \mathcal{E}'(f), Z = k] + \eta + \Delta}$$

$$\overset{(14)}{\geq} \frac{\Pr_{\mathcal{D}}[\mathcal{E}(f), \mathcal{E}'(f), Z = \ell]}{\Pr_{\mathcal{D}}[\mathcal{E}'(f), Z = \ell]} \cdot \frac{\Pr_{\mathcal{D}}[\mathcal{E}'(f), Z = k]}{\Pr_{\mathcal{D}}[\mathcal{E}(f), \mathcal{E}'(f), Z = k]} \cdot \left(\frac{1 - \frac{1}{\alpha} \cdot (\eta + \Delta)}{1 + \frac{1}{\alpha} \cdot (\eta + \Delta)}\right)^2. \tag{19}$$

Therefore, combining Equations (18) and (19), we have that conditioned on event $\mathscr{J}$

$$\Omega(f, \widehat{S}) = \min_{\ell, k \in [p]} \frac{\Pr_{\widehat{D}}[\mathcal{E}(f), \mathcal{E}'(f), \widehat{Z} = \ell]}{\Pr_{\widehat{D}}[\mathcal{E}'(f), \widehat{Z} = \ell]} \cdot \frac{\Pr_{\widehat{D}}[\mathcal{E}'(f), \widehat{Z} = k]}{\Pr_{\widehat{D}}[\mathcal{E}(f), \mathcal{E}'(f), \widehat{Z} = k]}$$
$$\text{(Using the definition of the fairness metric)}$$

$$\overset{(18),(19)}{\geq} \left(\frac{1 - \frac{1}{\alpha} \cdot (\eta + \Delta)}{1 + \frac{1}{\alpha} \cdot (\eta + \Delta)}\right)^2 \cdot \min_{\ell, k \in [p]} \frac{\Pr_{\mathcal{D}}[\mathcal{E}(f), \mathcal{E}'(f), Z = \ell]}{\Pr_{\mathcal{D}}[\mathcal{E}'(f), Z = \ell]} \cdot \frac{\Pr_{\mathcal{D}}[\mathcal{E}'(f), Z = k]}{\Pr_{\mathcal{D}}[\mathcal{E}(f), \mathcal{E}'(f), Z = k]}$$

$$= \Omega_{\mathcal{D}}(f) \cdot \left(\frac{1 - \frac{1}{\alpha} \cdot (\eta + \Delta)}{1 + \frac{1}{\alpha} \cdot (\eta + \Delta)}\right)^2. \qquad \text{(Using the definition of the fairness metric)}$$

This completes the proof of the upper bound in Definition A.5. It remains to prove the lower bound in Definition A.5 The proof of the lower bound in Definition A.5 is analogous to the proof of the upper bound. For any $\ell, k \in [p]$, we have

$$\frac{\Pr_{\widehat{D}}[\mathcal{E}(f), \mathcal{E}'(f), \widehat{Z} = \ell]}{\Pr_{\widehat{D}}[\mathcal{E}'(f), \widehat{Z} = \ell]} \cdot \frac{\Pr_{\widehat{D}}[\mathcal{E}'(f), \widehat{Z} = k]}{\Pr_{\widehat{D}}[\mathcal{E}(f), \mathcal{E}'(f), \widehat{Z} = k]}$$

$$\overset{(15),(16)}{\leq} \frac{\Pr_{\mathcal{D}}[\mathcal{E}(f), \mathcal{E}'(f), Z = \ell] + \eta + \Delta}{\Pr_{\mathcal{D}}[\mathcal{E}'(f), Z = \ell] - \eta - \Delta} \cdot \frac{\Pr_{\mathcal{D}}[\mathcal{E}'(f), Z = k] + \eta + \Delta}{\Pr_{\mathcal{D}}[\mathcal{E}(f), \mathcal{E}'(f), Z = k] - \eta - \Delta}. \tag{20}$$

Using Equation (14), we can upper bound the RHS of Equation (20) as follows

$$\frac{\Pr_{\mathcal{D}}[\mathcal{E}(f), \mathcal{E}'(f), Z = \ell] - \eta - \Delta}{\Pr_{\mathcal{D}}[\mathcal{E}'(f), Z = \ell] + \eta + \Delta} \cdot \frac{\Pr_{\mathcal{D}}[\mathcal{E}'(f), Z = k] - \eta - \Delta}{\Pr_{\mathcal{D}}[\mathcal{E}(f), \mathcal{E}'(f), Z = k] + \eta + \Delta}$$

$$\overset{(14)}{\leq} \frac{\Pr_{\mathcal{D}}[\mathcal{E}(f), \mathcal{E}'(f), Z = \ell]}{\Pr_{\mathcal{D}}[\mathcal{E}'(f), Z = \ell]} \cdot \frac{\Pr_{\mathcal{D}}[\mathcal{E}'(f), Z = k]}{\Pr_{\mathcal{D}}[\mathcal{E}(f), \mathcal{E}'(f), Z = k]} \cdot \left(\frac{1 + \frac{1}{\alpha} \cdot (\eta + \Delta)}{1 - \frac{1}{\alpha} \cdot (\eta + \Delta)}\right)^2. \tag{21}$$

Therefore, combining Equations (18) and (19), we have that conditioned on event $\mathscr{J}$

$$\Omega_{\mathcal{D}}(f) = \min_{\ell,k\in[p]} \frac{\Pr_{\mathcal{D}}[\mathcal{E}(f),\mathcal{E}'(f),Z=\ell]}{\Pr_{\mathcal{D}}[\mathcal{E}'(f),Z=\ell]} \cdot \frac{\Pr_{\mathcal{D}}[\mathcal{E}'(f),Z=k]}{\Pr_{\mathcal{D}}[\mathcal{E}(f),\mathcal{E}'(f),Z=k]}$$

(Using the definition of the fairness metric)

$$\overset{(18),(19)}{\geq} \left(\frac{1-\frac{1}{\alpha}\cdot(\eta+\Delta)}{1+\frac{1}{\alpha}\cdot(\eta+\Delta)}\right)^2 \cdot \min_{\ell,k\in[p]} \frac{\Pr_{\widehat{D}}[\mathcal{E}(f),\mathcal{E}'(f),\widehat{Z}=\ell]}{\Pr_{\widehat{D}}[\mathcal{E}'(f),\widehat{Z}=\ell]} \frac{\Pr_{\widehat{D}}[\mathcal{E}'(f),\widehat{Z}=k]}{\Pr_{\widehat{D}}[\mathcal{E}(f),\mathcal{E}'(f),\widehat{Z}=k]}$$

$$= \Omega_{\mathcal{D}}(f,\widehat{S}) \cdot \left(\frac{1-\frac{1}{\alpha}\cdot(\eta+\Delta)}{1+\frac{1}{\alpha}\cdot(\eta+\Delta)}\right)^2 . \qquad \text{(Using the definition of the fairness metric)}$$

$\square$

**Corollary A.7.** *For all $\alpha \in (0,1)$, any classifier $f \in \mathcal{F}$ satisfying*

$$\min_{\ell\in[p]} \Pr_{\mathcal{D}}\left[\mathcal{E}(f),\mathcal{E}'(f),\widehat{Z}=\ell\right] \geq \alpha, \tag{22}$$

*is $(\frac{1-(\eta+\Delta)/(\alpha-\eta-\Delta)}{1+(\eta+\Delta)/(\alpha-\eta-\Delta)})^2$-stable with respect to the fairness metric $\Omega$ defined by events $\mathcal{E}$ and $\mathcal{E}'$.*

*Proof.* Suppose event $\mathscr{J}$ defined in Lemma A.6 occurs. Then by the first condition of $\mathscr{J}$, Equation (15), we have that $\left|\Pr_{\mathcal{D}}\left[\mathcal{E}(f),\mathcal{E}'(f),Z=\ell\right]-\Pr_{\widehat{D}}\left[\mathcal{E}(f),\mathcal{E}'(f),\widehat{Z}=\ell\right]\right| \leq \eta+\Delta$. Combining this with Equation (22), we get that $\Pr_{\widehat{D}}\left[\mathcal{E}(f),\mathcal{E}'(f),\widehat{Z}=\ell\right] \geq \alpha-\eta-\Delta$. Then repeating the proof of Lemma A.6 we get that, conditioned on $\mathscr{J}$, the following holds

$$\left(\frac{1+(\eta+\Delta)/(\alpha-\eta-\Delta)}{1-(\eta+\Delta)/(\alpha-\eta-\Delta)}\right)^2 \leq \frac{\Omega_{\mathcal{D}}(f)}{\Omega(f,\widehat{S})} \leq \left(\frac{1-(\eta+\Delta)/(\alpha-\eta-\Delta)}{1+(\eta+\Delta)/(\alpha-\eta-\Delta)}\right)^2$$

Since $\mathscr{J}$ occurs with probability at least $1-2\delta_0 > 1-\delta$, we get that $f$ is $(\frac{1-(\eta+\Delta)/(\alpha-\eta-\Delta)}{1+(\eta+\Delta)/(\alpha-\eta-\Delta)})^2$-stable. $\square$

Setting $\alpha$ as $\lambda+\eta+\Delta$, we recover Lemma 4.8 from Corollary A.7. Now, we are ready to prove the main result in this step: Lemma A.8.

**Lemma A.8** (**Any feasible solution of Program (ErrTolerant) is approximately fair).** *For all $\Delta, \delta_0 \in (0,1)$, given $N \geq \Theta\big(\Delta^{-2}\cdot\big(\mathrm{VC}(\mathcal{F})\cdot\log\big(\mathrm{VC}(\mathcal{F})/\Delta\big)+\log\big(1/\delta_0\big)\big)\big)$ iid samples $S$ from $\mathcal{D}$, and corresponding perturbed samples $A(S) := \{(x_i,y_i,\widehat{z}_i)\}_{i\in[N]}$, with probability at least $1-2p\delta_0$, any $f \in \mathcal{F}$ feasible for Program (ErrTolerant) satisfies*

$$\Omega_{\mathcal{D}}(f) \geq \tau - \frac{8\eta\tau}{\lambda-2\eta} - \Delta_0.$$

*Proof.* Consider any classifier $f \in \mathcal{F}$ that is feasible for Program (ErrTolerant). $f$ must satisfy

$$\Omega(f,\widehat{S}) \geq \tau \cdot \left(\frac{1-(\eta+\Delta)/\lambda}{1+(\eta+\Delta)/\lambda}\right)^2, \tag{23}$$

$$\forall\,\ell\in[p], \quad \Pr_{\widehat{D}}\left[\mathcal{E}(f),\mathcal{E}'(f),\widehat{Z}=\ell\right] \geq \lambda-\eta-\Delta. \tag{24}$$

Since $f$ satisfies Equation (24), using Lemma 4.8, we get that $f$ is $(\frac{1-(\eta+\Delta)/(\lambda-2\eta-2\Delta)}{1+(\eta+\Delta)/(\lambda-2\eta-2\Delta)})^2$-stable. Thus, with high probability, it holds that

$$\Omega_{\mathcal{D}}(f) \geq \Omega(f,\widehat{S}) \cdot \left(\frac{1-(\eta+\Delta)/(\lambda-2\eta-2\Delta)}{1+(\eta+\Delta)/(\lambda-2\eta-2\Delta)}\right)^2 \tag{25}$$

$$\overset{(23)}{\geq} \tau \cdot \left(\frac{1-(\eta+\Delta)/\lambda}{1+(\eta+\Delta)/\lambda}\right)^2 \cdot \left(\frac{1-(\eta+\Delta)/(\lambda-2\eta-2\Delta)}{1+(\eta+\Delta)/(\lambda-2\eta-2\Delta)}\right)^2 .$$

We can lower bound the RHS of the above equation using Fact A.9.

**Fact A.9.** *For all $\alpha \in (0,1]$ and $\eta,\Delta \in [0,1]$, $\left(\frac{1-\frac{1}{\alpha}\cdot(\eta+\Delta)}{1+\frac{1}{\alpha}\cdot(\eta+\Delta)}\right)^2 \geq \left(1-\frac{4}{\alpha}\cdot(\eta+\Delta)\right).$*

Using Fact A.9 twice we get that

$$
\begin{aligned}
\Omega_{\mathcal{D}}(f) \;\geq\; & \tau \cdot \left(1 - \frac{4(\eta+\Delta)}{\lambda}\right) \cdot \left(1 - \frac{4(\eta+\Delta)}{\lambda - 2\eta - 2\Delta}\right) \\
> \;& \tau \cdot \left(1 - \frac{4(\eta+\Delta)}{\lambda} - \frac{4(\eta+\Delta)}{\lambda - 2\eta - 2\Delta}\right) \quad \text{(Using that } \lambda > 2\eta + 2\Delta \text{ and } \lambda, \eta, \Delta > 0) \\
\geq \;& \tau \cdot \left(1 - \frac{8(\eta+\Delta)}{\lambda - 2\eta - 2\Delta}\right) \quad\quad\quad\quad \text{(Using that } \lambda > 2\eta + 2\Delta \text{ and } \lambda, \eta, \Delta > 0) \\
= \;& \tau \cdot \left(1 - \frac{8(\eta+\Delta)}{\lambda - 2\eta} \cdot \frac{1}{1 - \frac{2}{(\lambda-2\eta)} \cdot \Delta}\right) \\
\geq \;& \tau \cdot \left(1 - \frac{8(\eta+\Delta)}{\lambda - 2\eta} \cdot \left(1 + \frac{4}{(\lambda - 2\eta)} \cdot \Delta\right)\right) \quad \text{(Using that } \tfrac{2}{(\lambda-2\eta)} \cdot \Delta \in [0, \tfrac{1}{2}]) \\
= \;& \tau \cdot \left(1 - \frac{8\eta}{\lambda - 2\eta} - \frac{8\Delta}{\lambda - 2\eta} - \frac{32\eta\Delta}{(\lambda - 2\eta)^2} - \frac{32\Delta^2}{(\lambda - 2\eta)^2}\right) \\
\overset{(9)}{=} \;& \tau \cdot \left(1 - \frac{8\eta}{\lambda - 2\eta} - \frac{\Delta_0}{3} - \frac{\Delta_0}{3} - \frac{\Delta_0}{3}\right).
\end{aligned}
$$

Finally, from the discussion in the proof of Lemma A.6, it follows that for our choice of $N$, the above equation holds with probability at least $1 - 2p\delta_0$. $\qquad\square$

### A.1.4 Step 3: $f^\star$ is feasible for Program (ErrTolerant) with high probability

In this step, we conclude the proof of Theorem 4.3. It remains show that $f^\star$ is feasible for Program (ErrTolerant): The fairness guarantee follows from Lemma A.8, and if $f^\star$ is feasible for Program (ErrTolerant), then the accuracy guarantee follows from Lemma A.4.

**Lemma A.10** (**Structure of the optimal fair classifier**). *If Assumption 1 holds, then For all $\Delta, \delta_0 \in (0, 1)$, given $N \geq \Theta\big(\Delta^{-2} \cdot \big(\mathrm{VC}(\mathcal{F}) \cdot \log\big(\mathrm{VC}(\mathcal{F})/\Delta\big) + \log\big(1/\delta_0\big)\big)\big)$ iid samples $S$ from $\mathcal{D}$, and corresponding perturbed samples $A(S) := \{(x_i, y_i, \widehat{z}_i)\}_{i \in [N]}$, with probability at least $1 - \delta_0$, $f^\star$ is feasible for Program (ErrTolerant).*

*Proof.* Let $\mathcal{G}$ be the event that for all $\ell \in [p]$ and $f \in \mathcal{F}$

$$
\left|\mathrm{Err}_{\widehat{D}}(f) - \mathrm{Err}_{\mathcal{D}}(f)\right| < \eta + \Delta. \tag{26}
$$

Using Lemma A.3 with $g(\widetilde{y}, y, z) := \mathbb{I}\left[\widetilde{y} \neq y\right]$, shows that with probability at least $1 - \delta_0$ Inequality (26) holds for all $f \in \mathcal{F}$. Thus,

$$
\Pr[\mathcal{G}] \geq 1 - \delta_0. \tag{27}
$$

Suppose events $\mathcal{G}$ and $\mathcal{J}$ hold. To show that $f^\star$ is feasible for Program (ErrTolerant), we have to show that

$$
\Omega(f^\star, \widehat{S}) \geq \tau \cdot \left(1 - \frac{4\eta}{\lambda} - \Delta_0\right), \tag{28}
$$

$$
\forall \ell \in [p], \quad \Pr_{\widehat{D}}[\mathcal{E}(f^\star), \mathcal{E}'(f^\star), \widehat{Z} = \ell] \geq \lambda - \eta - \Delta_0. \tag{29}
$$

Since event $\mathcal{J}$ and Assumption 1 hold, we can apply Lemma A.6 to get that $\Omega(f^\star, \widehat{S}) \geq \Omega_{\mathcal{D}}(f^\star) \cdot (1 - \frac{4\eta}{\lambda} - \Delta_0)$. Then, Equation (28) holds since $\Omega_{\mathcal{D}}(f^\star) \geq \tau$. Finally, Equation (29) follows by using Assumption 1 and Equation (15) in the definition of event $\mathcal{J}$. $\qquad\square$

*Proof of Theorem 4.3.* Let $\mathcal{J}$ be the event that the fairness guarantee in Lemma A.8 holds and $\mathcal{G}$ be the event that $f^\star$ is feasible for Program (ErrTolerant). Using the union bound $\mathcal{J}$ and $\mathcal{G}$, we get that

$$
\begin{aligned}
\Pr_{\mathcal{D}}[\mathcal{J} \wedge \mathcal{G}] &\geq 1 - (2p + 1) \cdot \delta_0 \quad\quad \text{(Using Lemma A.8 and Lemma A.10)} \\
&\overset{(9)}{\geq} 1 - \delta.
\end{aligned}
$$

Suppose events $\mathscr{J}$ and $\mathscr{G}$ occur. By Lemma A.8, we know that the following holds $\Omega_{\mathcal{D}}(f) \geq \tau - {}^{8\eta\tau}/_{(\lambda-2\eta)} - \Delta$. Since $\Delta \leq \Delta_0 \leq {}^{8\eta}/_{(\lambda-2\eta)} - {}^{\nu}/_{\tau}$, it follows that

$$\Omega_{\mathcal{D}}(f) \geq \tau - \nu.$$

By Lemma A.10, we know that $f^{\star}$ is feasible for Program (ErrTolerant). Then, from Lemma A.4 it follows that $\mathrm{Err}_{\mathcal{D}}(f_{\mathrm{ET}}) - \mathrm{Err}_{\mathcal{D}}(f^{\star}) \leq 2\eta + \Delta$. Since $\Delta \leq \varepsilon - 2\eta$, it follows that

$$\mathrm{Err}_{\mathcal{D}}(f_{\mathrm{ET}}) - \mathrm{Err}_{\mathcal{D}}(f^{\star}) \leq \varepsilon.$$

$\square$

### A.1.5 Generalization of Theorem 4.3 to the Nasty Sample Noise model

In this section, we generalize Theorem 4.3 to the Nasty sample noise model. Precisely, we show that if Assumption 1 holds with constant $\lambda > 0$ and $\mathcal{F}$ has VC dimension $d \in \mathbb{N}$, then for all perturbation rates $\eta \in (0, {}^{\lambda}/_2)$, fairness thresholds $\tau \in (0, 1]$, bounds on error $\varepsilon > 2\eta$ and constraint violation $\nu > {}^{8\eta\tau}/_{(\lambda-2\eta)}$, and confidence parameters $\delta \in (0, 1)$, given sufficiently many perturbed samples from the $\eta$-Hamming model, with probability at least $1 - \delta$, it holds that

$$\mathrm{Err}_{\mathcal{D}}(f_{\mathrm{ET}}) - \mathrm{Err}_{\mathcal{D}}(f^{\star}) \leq \varepsilon \text{ and } \Omega_{\mathcal{D}}(f_{\mathrm{ET}}) \geq \tau - \nu,$$

where $f_{\mathrm{ET}}$ is an optimal solution of Program (ErrTolerant).

The proof of the above generalization is almost identical to the proof of Theorem 4.3. Instead of repeating the entire proof, we highlight the changes required in the proof of Theorem 4.3.

The generalization requires two changes: (1) Proving an analogue of Lemma A.3 for the Nasty Sample Noise model, (2) generalizing the definition of $s$-stability (Definition A.5) to the Nasty Sample Noise model. These changes are sufficient because all other arguments use the guarantees of Lemma A.3 or $s$-stability, without using any properties of the $\eta$-Hamming model. Updating Definition A.5 only requires changing the perturbation model from $\eta$-Hamming to $\eta$-Nasty Sample Noise; we omit the formal statement. Next, we prove Lemma A.11 which is the required analogue of Lemma A.3.

**Lemma A.11 (Bound on difference in means of bounded functions on $\mathcal{D}$ and on $\widehat{S}$).** *For any bounded function $\ell \colon \{0, 1\} \times \{0, 1\} \times [p] \to [0, 1]$, constants $\Delta, \delta_0 \in (0, 1)$, and adversaries $A$ admissible under the $\eta$-Nasty Sample Noise model, given $N \geq \Theta\big(\Delta^{-2} \cdot \big(\mathrm{VC}(\mathcal{F}) \cdot \log\big({}^{\mathrm{VC}(\mathcal{F})}/_\Delta\big) + \log\big({}^1/_{\delta_0}\big)\big)\big)$ samples $S$ iid from $\mathcal{D}$, and corresponding perturbed samples $A(S) \coloneqq \{(\widehat{x}_i, \widehat{y}_i, \widehat{z}_i)\}_{i \in [N]}$, with probability at least $1 - \delta_0$, it holds that*

$$\forall\, f \in \mathcal{F}, \quad \left| \mathbb{E}_{(\widehat{X}, \widehat{Y}, \widehat{Z}) \sim \widehat{D}} g(f(\widehat{X}, \widehat{Z}), \widehat{Y}, \widehat{Z}) - \mathbb{E}_{(X,Y,Z) \sim \mathcal{D}} [g(f(X, Z), Y, Z)] \right| \leq \Delta + \eta,$$

*where $\widehat{D}$ is the empirical distribution of $A(S)$.*

*Proof.* Let $S \coloneqq \{(x_i, z_i, y_i)\}_{i \in [N]}$. Using the triangle inequality for absolute value, we have

$$\left| \mathbb{E}_{(\widehat{X}, \widehat{Y}, \widehat{Z}) \sim \widehat{D}} [g(f(\widehat{X}, \widehat{Z}), \widehat{Y}, \widehat{Z})] - \mathbb{E}_{(X,Y,Z) \sim \mathcal{D}} [g(f(X, Z), Y, Z)] \right|$$
$$\leq \left| \mathbb{E}_{(X,Y,Z) \sim D} [g(f(X, Z), Y, Z)] - \mathbb{E}_{(X,Y,Z) \sim \mathcal{D}} [g(f(X, Z), Y, Z)] \right|$$
$$+ \left| \mathbb{E}_{(\widehat{X}, \widehat{Y}, \widehat{Z}) \sim \widehat{D}} [g(f(\widehat{X}, \widehat{Z}), \widehat{Y}, \widehat{Z})] - \mathbb{E}_{(X,Y,Z) \sim \mathcal{D}} [g(f(X, Z), Y, Z)] \right|. \quad (30)$$

We can upper bound the first term in the RHS using Lemma A.2; this is identical to the argument under the $\eta$-Hamming model. In particular, we have that with probability at least $1 - \delta$, for all $f \in \mathcal{F}$, it holds that

$$\left| \mathbb{E}_{(X,Y,Z) \sim D} [g(f(X, Z), Y, Z)] - \mathbb{E}_{(X,Y,Z) \sim \mathcal{D}} [g(f(X, Z), Y, Z)] \right| \leq \Delta. \quad (31)$$

(The proof of the upper bound on the second term in the RHS of Equation (30) is slightly different from the proof under the $\eta$-Hamming model.) For all $f \in \mathcal{F}$, it holds that

$$
\left| \mathbb{E}_{(\widehat{X}, \widehat{Y}, \widehat{Z}) \sim \widehat{D}}[g(f(\widehat{X}, \widehat{Z}), \widehat{Y}, \widehat{Z})] - \mathbb{E}_{(X,Y,Z) \sim D}[g(f(X, Z), Y, Z)] \right|
$$

$$
= \quad \frac{1}{N} \left| \sum_{i \in [N]} g(f(\widehat{x}_i, \widehat{z}_i), \widehat{y}_i, \widehat{z}_i) - g(f(x_i, z_i), y_i, z_i) \right|
$$

$$
= \quad \frac{1}{N} \left| \sum_{i \in [N]\,:\, (x_i, y_i, z_i) \neq (\widehat{x}_i, \widehat{y}_i, \widehat{z}_i)} g(f(\widehat{x}_i, \widehat{z}_i), \widehat{y}_i, \widehat{z}_i) - g(f(x_i, z_i), y_i, z_i) \right|
$$

(For all $i \in [N]$, where $(x_i, y_i, z_i) = (\widehat{x}_i, \widehat{y}_i, \widehat{z}_i)$, $g(f(\widehat{x}_i, \widehat{z}_i), \widehat{y}_i, \widehat{z}_i) = g(f(x_i, z_i), y_i, z_i)$.)

$$
\leq \quad \frac{1}{N} \left| \sum_{i \in [N]\,:\, (x_i, y_i, z_i) \neq (\widehat{x}_i, \widehat{y}_i, \widehat{z}_i)} 1 \right| \qquad \text{(Using that } g \text{ is bounded by 0 and 1)}
$$

$$
\leq \quad \eta. \tag{32}
$$

Since with probability at least $1 - \delta$, both Equations (31) and (32) hold for all $f \in \mathcal{F}$, substituting them in Equation (30) gives us the required bound. $\qquad\square$

One can substitute Lemma A.3 by Lemma A.11 and repeat the proof of Theorem 4.3 for the $\eta$-Nasty Sample Noise model.

## A.2 Proof of Theorem 4.4

Theorem 4.4 assumes that $\mathcal{F}$ shatters the set $\{x_A, x_B, x_C\} \times [2] \subseteq \mathcal{X} \times [p]$, the fairness threshold $\tau \in (1/2, 1)$, and statistical rate is the fairness metric; recall that the statistical rate of a classifier $f \in \mathcal{F}$ is

$$
\Omega_{\mathcal{D}}(f) := \frac{\min_{\ell \in [p]} \Pr_{\mathcal{D}}[f = 1 \mid Z = \ell]}{\max_{\ell \in [p]} \Pr_{\mathcal{D}}[f = 1 \mid Z = \ell]}.
$$

Then, given parameters $\tau \in (1/2, 1)$ and $\delta \in [0, 1/2)$, our goal is to show that for any

$$
\varepsilon \in \left[0, \frac{1}{2}\right) \text{ and } \nu \in \left[0, \tau - \frac{1}{2}\right),
$$

$\mathcal{F}$ is not $(\varepsilon, \nu)$-learnable with perturbation rate $\eta$ and confidence $\delta$. We prove a more general result: Given parameters $\tau \in [0, 1)$, $c \in (0, \min\{\tau, 1/2\})$, and $\delta \in [0, 1/2)$, we show that for any

$$
\varepsilon \in [0, c) \text{ and } \nu \in [0, \tau - c),
$$

$\mathcal{F}$ is not $(\varepsilon, \nu)$-learnable with perturbation rate $\eta$ and confidence $\delta$. When $\tau > 1/2$, the original result follows by taking the limit as $c$ approaches $1/2$.

### A.2.1 Proof of Theorem 4.4

*Proof of Theorem 4.4 (assuming Lemmas A.12 to A.14).* Let $\mathcal{L}$ be any learner. Set

$$
\alpha := \min\left\{ \frac{\eta}{2}, 1 - \tau, \tau - c - \nu, c - \varepsilon \right\}, \tag{33}
$$

and confidence parameter $\delta \in (0, 1/2)$ (in Definition 3.2). We construct three distributions $\mathcal{D}_1, \mathcal{D}_2,$ and $\mathcal{D}_3$, parameterized by $\alpha$, that satisfy Lemma A.12. (See Table 2 for details of the distributions).

**Lemma A.12 (Adversary can hide the true distribution).** *For all $\delta_0 \in (0, 1)$, $\alpha \in (0, \eta)$, and $\ell, k \in [3]$, given $N \geq 3 \cdot \ln(1/\delta_0) \cdot (\eta - \alpha)^{-2}$ iid samples, $S$, from $\mathcal{D}_\ell$ there is an adversary $A \in \mathcal{A}(\eta)$ such that with probability $1 - \delta_0$ (over the draw of $S$) the perturbed samples $\widehat{S} := A(S)$ are distributed as iid draws from $\mathcal{D}_k$.*

Suppose $\mathcal{L}$ is given $N$ samples, where

$$
N \geq 3 \cdot \ln 2 \cdot (\eta - \alpha)^{-2}. \tag{34}
$$

Consider three cases, where in the $k$-th case ($k \in [3]$) the samples in $S$ are iid from $\mathcal{D}_k$. By Lemma A.12, in each case there is an $A \in \mathcal{A}(\eta)$, such that with probability at least $1/2$, $\widehat{S} := A(S)$ have the distribution $\mathcal{D}_1$.

Thus, given $\widehat{S}$, with probability at least $1/2$, $\mathcal{L}$ cannot identify the distribution from which $S$ was drawn. As a result, with probability at least $1/2$, $\mathcal{L}$ outputs the same classifier, say $f_{\mathrm{Com}} \in \mathcal{F}$, in all three cases. We show that no $f_{\mathrm{Com}} \in \mathcal{F}$ satisfies the accuracy and fairness guarantee in all three cases.

**Lemma A.13 (No good classifier for all cases).** *There is no classifier $f \in \mathcal{F}$ such that for all $k \in [3]$, $\mathrm{Err}_{\mathcal{D}_k}(f) < c \cdot (1 - \alpha)$ and $\Omega_{\mathcal{D}_k}(f) \geq c + \alpha$.*

**Lemma A.14 (A good classifier for each case).** *For each $k \in [3]$, there is an $h_k \in \mathcal{F}$ such that $\mathrm{Err}_{\mathcal{D}_k}(h_k) < c\alpha/2$ and $\Omega_{\mathcal{D}_k}(h_k) > 1 - \alpha$.*

Note that for each $k \in [3]$, $f_k^\star$ satisfies the fairness constraint because $\tau < 1 - \alpha$ (Equation (33)). Thus, the optimal fair classifier $f_k^\star$ for $\mathcal{D}_k$ subject to having a statistical rate $\tau$ must satisfy

$$\mathrm{Err}_{\mathcal{D}_k}(f_k^\star) < \frac{c\alpha}{2}. \tag{35}$$

(Otherwise, we have a contradiction as $h_k$ satisfies the fairness constraints and has a smaller error than $f_k^\star$.) If $\mathcal{L}$ is a $(\varepsilon, \nu)$-learner, then in the $k$-th case, with probability at least $1 - \delta > 1/2$, $\mathcal{L}$ must output a classifier $f_k$ which satisfies

$$\mathrm{Err}_{\mathcal{D}_k}(f_k) - \mathrm{Err}_{\mathcal{D}_k}(f_k^\star) \leq \varepsilon \text{ and } \tau - \Omega_{\mathcal{D}_k}(f_k) \leq \nu.$$

But in all cases with probability at least $1/2$, $\mathcal{L}$ outputs $f_{\mathrm{Com}}$. Because $1/2 > \delta$, $f_{\mathrm{Com}}$ must satisfy:

$$\text{For all } k \in [3], \quad \mathrm{Err}_{\mathcal{D}_k}(f_{\mathrm{Com}}) - \mathrm{Err}_{\mathcal{D}_k}(f_k^\star) \leq \varepsilon \text{ and } \tau - \Omega_{\mathcal{D}_k}(f_{\mathrm{Com}}) \leq \nu. \tag{36}$$

But from Lemma A.13 we know that for each $k \in [3]$ either

$$\mathrm{Err}_{\mathcal{D}_k}(f_{\mathrm{Com}}) \geq c \cdot (1 - \alpha) \quad \text{or} \quad \Omega_{\mathcal{D}_k}(f_{\mathrm{Com}}) < c + \alpha. \tag{37}$$

**(Case A)** $\mathrm{Err}_{\mathcal{D}_k}(f_{\mathrm{Com}}) \geq c \cdot (1 - \alpha)$**:** In this case, from Equation (36), we have

$$
\begin{aligned}
\varepsilon &\geq c \cdot (1 - \alpha) - \mathrm{Err}_{\mathcal{D}_k}(f_k^\star) \\
&\overset{(35)}{>} c \cdot (1 - \alpha) - \frac{c\alpha}{2} \\
&> c - \alpha \qquad\qquad\qquad\quad \text{(Using that } c < 1/2 \text{ and } \alpha > 0) 
\end{aligned} \tag{38}
$$

But because $\alpha \leq c - \varepsilon$, Equation (38) cannot hold.

**(Case B)** $\Omega_{\mathcal{D}_k}(f_{\mathrm{Com}}) < c + \alpha$**:** In this case, from Equation (36), we have

$$\nu > \tau - c - \alpha. \tag{39}$$

But because $\alpha \leq \tau - c - \nu$, Equation (39) does not hold.

Therefore, we have a contradiction. Hence, $\mathcal{L}$ is not an $(\varepsilon, \nu)$-learner for $\mathcal{F}$. Since the choice of $\mathcal{L}$ was arbitrary, we have shown that there is no learner which $(\varepsilon, \nu)$-learns $\mathcal{F}$. It remains to prove Lemmas A.12 to A.14. $\qquad\square$

### A.2.2 Proof of Lemma A.12

Set $\mathcal{D}_1$, $\mathcal{D}_2$, and $\mathcal{D}_3$ to be unique distributions with marginal distribution specified in Table 2, such that, for any draw $(X, Y, Z) \sim \mathcal{D}_k$ ($k \in [3]$) $Y$ takes the value $\mathbb{I}[X = x_A]$, i.e.,

$$Y = \begin{cases} 1 & \text{if } X = x_A, \\ 0 & \text{otherwise.} \end{cases} \tag{40}$$

In particular, the construction of $\mathcal{D}_1$, $\mathcal{D}_2$, and $\mathcal{D}_3$ ensures that the total variation distance between any pair of distributions is less than $\alpha$. In this section, we prove the following generalization of Lemma A.12.

**Proposition A.15 (Adversary can hide the true distribution).** *For all $\delta, \eta \in (0, 1)$, $\alpha \in (0, \eta)$, and two distributions $\mathcal{P}$ and $\mathcal{Q}$ over $\mathcal{X} \times \{0, 1\} \times [p]$ that satisfy the following conditions:*

*(C1)* $\mathrm{TV}(\mathcal{P}, \mathcal{Q}) = \alpha$,

(a) $\mathcal{D}_1$: $\Pr_{\mathcal{D}_1}[(X,Z) = (r,s)]$ for $(r,s) \in \{x_A, x_B, x_C\} \times [2]$.

|   | $x_A$ | $x_B$ | $x_C$ |
|---|---|---|---|
| 1 | $c(1-\alpha)$ | $(1-c)(1-\alpha)$ | $\alpha/2$ |
| 2 | $c\alpha/2$ | $\alpha(1-c)/2$ | $0$ |

(b) $\mathcal{D}_2$: $\Pr_{\mathcal{D}_2}[(X,Z) = (r,s)]$ for $(r,s) \in \{x_A, x_B, x_C\} \times [2]$.

|   | $x_A$ | $x_B$ | $x_C$ |
|---|---|---|---|
| 1 | $c(1-\alpha)$ | $(1-c)(1-\alpha/2)$ | $c\alpha/2$ |
| 2 | $c\alpha/2$ | $0$ | $\alpha(1-c)/2$ |

(c) $\mathcal{D}_3$: $\Pr_{\mathcal{D}_3}[(X,Z) = (r,s)]$ for $(r,s) \in \{x_A, x_B, x_C\} \times [2]$.

|   | $x_A$ | $x_B$ | $x_C$ |
|---|---|---|---|
| 1 | $c(1-\alpha/2)$ | $(1-c)(1-\alpha)$ | $\alpha(1-c)/2$ |
| 2 | $0$ | $\alpha(1-c)/2$ | $c\alpha/2$ |

Table 2: Marginal distributions of $\mathcal{D}_1$, $\mathcal{D}_2$, $\mathcal{D}_3$ over $\mathcal{X} \times [2]$. Recall that for a sample $(X,Y,Z) \sim \mathcal{D}_k$ ($k \in [3]$), $Y$ takes the value $\mathbb{I}[X = x_A]$.

*(C2)* $\mathcal{P}$ *and* $\mathcal{Q}$ *have the same marginal on* $\mathcal{X}$*, i.e., for all* $T \subseteq \mathcal{X}$*,*

$$\Pr_{(X,Y,Z)\sim\mathcal{P}}[X \in T] = \Pr_{(X,Y,Z)\sim\mathcal{Q}}[X \in T],$$

*(C3) for a random sample* $(X,Y,Z)$ *drawn from* $\mathcal{P}$*, the label* $Y$ *is independent of* $Z$ *conditioned on* $X$*, i.e.,* $Y \perp Z \mid X$*, similarly for a random sample* $(X,Y,Z)$ *drawn from* $\mathcal{Q}$*, the label* $Y$ *is independent of* $Z$ *conditioned on* $X$*.*

*Then, there is an adversary* $A \in \mathcal{A}(\eta)$*, that given* $N$ *iid samples* $S$ *from* $\mathcal{P}$*, where*

$$N \geq 3 \cdot \ln(1/\delta) \cdot (\eta - \alpha)^{-2},$$

*outputs a perturbed samples* $\widehat{S} := A(S)$ *such that with probability* $1 - \delta$ *(over the draw of* $S$*) samples in* $\widehat{S}$ *are distributed as iid draws from* $\mathcal{Q}$*.*

Note that Lemma A.12 follows by substituting $\mathcal{P}$ and $\mathcal{Q}$ by $\mathcal{D}_\ell$ and $\mathcal{D}_k$ respectively.

*Proof of Proposition A.15.* Let $A \in \mathcal{A}(\eta)$ use the following algorithm:

---

1. **For** $r, s \in \mathcal{X}$ **do: Set** $p(r,s) := \min\left\{\frac{\Pr_{(X,Y,Z)\sim\mathcal{Q}}[(X,Z)=(r,s)]}{\Pr_{(X,Y,Z)\sim\mathcal{P}}[(X,Z)=(r,s)]}, 1\right\}$
   *//Since $A$ knows the distributions $\mathcal{P}$ and $\mathcal{Q}$, it can compute $p(r,s)$*

2. **For** $i \in [N]$ **do:**

   (a) **Sample** a point $t_i$ uniformly at random from $[0,1]$
   (b) **If** $t_i \leq p(X_i, Z_i)$ **then: Set** $\widetilde{Z}_i := Z_i$,
   (c) **Otherwise: Set** $\widetilde{Z}_i := 3 - Z_i$     *//If $Z_i = 1$ set $\widetilde{Z}_i = 2$, if $Z_i = 2$ set $\widetilde{Z}_i = 1$*

3. **If** $\sum_{i\in[N]} \mathbb{I}[\widetilde{Z}_i \neq Z_i] < \delta \cdot N$ **then: return** $\{(X_i, \widetilde{Z}_i, Y_i)\}_{i\in[N]}$,

4. **Otherwise: return** $\{(X_i, Z_i, Y_i)\}_{i\in[N]}$

---

(Since $A$ knows the distributions $\mathcal{P}$ and $\mathcal{Q}$, it can compute $p(r,s)$.)

**Lemma A.16.** *If* $N \geq 3 \cdot \ln(1/\delta) \cdot (\eta - \alpha)^{-2}$*, then with probability at least* $1 - \delta$*,* $\sum_{i\in[N]} \mathbb{I}[\widetilde{Z}_i \neq Z_i] < \eta \cdot N$*.*

*Proof.* For all $i \in [N]$, let $C_i \in \{0, 1\}$ be a random variable indicating if $\widetilde{Z}_i \neq Z_i$. Since for each $i \in [N]$ the sample $(X_i, Y_i, Z_i)$ and the point $t_i$ is drawn independently of others, it follows that the random variables $C_i$ are independent of each other. Suppose we can show that $\Pr[C_i] \leq \alpha$. Then, by linearity of expectation, it follows that $\mathbb{E}[\sum_{i \in [N]} C_i] \leq \alpha \cdot N$. Thus, using the Chernoff bound, we get that $\Pr[\sum_{i \in [N]} C_i \leq \eta \cdot N] \geq 1 - \delta$. This completes the proof of Lemma A.16, up to proving $\Pr[C_i = 1] \leq \alpha$. Towards this, observe that

$$
\begin{aligned}
&\Pr[C_i = 1] \\
&= \sum_{r \in \mathcal{X}, s \in [p]} \Pr\left[(X_i, Z_i) = (r, s)\right] \cdot \Pr\left[C_i = 1 \mid (X_i, Z_i) = (r, s)\right] \\
&= \sum_{r \in \mathcal{X}, s \in [p]} \Pr\left[(X_i, Z_i) = (r, s)\right] \cdot \Pr\left[Z_i \neq \widetilde{Z}_i \mid (X_i, Z_i) = (r, s)\right] && \text{(Definition of } C_i) \\
&= \sum_{r \in \mathcal{X}, s \in [p]} \Pr\left[(X_i, Z_i) = (r, s)\right] \cdot (1 - p(r, s)) && \text{(Definition of } p(r, s)) \\
&= \sum_{r \in \mathcal{X}, s \in [p]} \Pr\left[(X_i, Z_i) = (r, s)\right] \cdot \max\left\{1 - \frac{\Pr_{\mathcal{Q}}[(X, Z) = (r, s)]}{\Pr_{\mathcal{P}}[(X, Z) = (r, s)]}, 0\right\} \\
&&& \text{(Definition of } p(r, s)) \\
&= \sum_{r \in \mathcal{X}, s \in [p]} \Pr_{\mathcal{P}}\left[(X, Z) = (r, s)\right] \cdot \max\left\{1 - \frac{\Pr_{\mathcal{Q}}[(X, Z) = (r, s)]}{\Pr_{\mathcal{P}}[(X, Z) = (r, s)]}, 0\right\} \\
&&& \text{(Using that for each } i \in [N], (X_i, Y_i, Z_i) \sim \mathcal{P}) \\
&= \sum_{r \in \mathcal{X}, s \in [p]} \max\left\{\Pr_{\mathcal{P}}\left[(X, Z) = (r, s)\right] - \Pr_{\mathcal{Q}}[(X, Z) = (r, s)], 0\right\} \\
&= \mathrm{TV}(\mathcal{P}, \mathcal{Q}) \\
&= \alpha. && \text{(Using that } \mathrm{TV}(\mathcal{P}, \mathcal{Q}) = \alpha)
\end{aligned}
$$

$\square$

**Lemma A.17.** *Each sample in $\widetilde{S} := \{(X_i, \widetilde{Z}_i, Y_i)\}_{i \in [N]}$ is independent of each other and is distributed according to $\mathcal{Q}$.*

*Proof.* Since for each $i \in [N]$ the sample $(X_i, Y_i, Z_i)$ and the point $t_i$ is drawn independently of others, it follows that the samples $(X_i, \widetilde{Z}_i, Y_i)$ are independent of each other.

To see that $(X_i, \widetilde{Z}_i, Y_i) \sim \mathcal{Q}$, fix any $i \in [N]$, $r \in \mathcal{X}$, and $s \in [2]$. It holds that

$$\Pr[X_i = r, \widetilde{Z}_i = s, Y_i = 1] \overset{(40)}{=} \Pr[Y_i = 1 \mid X_i = r] \cdot \Pr[X_i = r, \widetilde{Z}_i = s], \tag{41}$$

$$\Pr[X_i = r, \widetilde{Z}_i = s, Y_i = 0] \overset{(40)}{=} \Pr[Y_i = 0 \mid X_i = r] \cdot \Pr[X_i = r, \widetilde{Z}_i = s], \tag{42}$$

here we used the fact that $Y_i$ is independent of $Z_i$ (see Equation (40)). Suppose that

$$\Pr[X_i = r, \widetilde{Z}_i = s] = \Pr_{\mathcal{Q}}[(X, Z) = (r, s)]. \tag{43}$$

Then, from Equation (40) we have that

$$\Pr_{\mathcal{Q}}[X = r, \widetilde{Z} = s, Y = 1] \overset{(40)}{=} \Pr[Y_i = 1 \mid X_i = r] \cdot \Pr_{\mathcal{Q}}[(X, Z) = (r, s)], \tag{44}$$

$$\Pr_{\mathcal{Q}}[X = r, \widetilde{Z} = s, Y = 0] \overset{(40)}{=} \Pr[Y_i = 0 \mid X_i = r] \cdot \Pr_{\mathcal{Q}}[(X, Z) = (r, s)]. \tag{45}$$

Further, combining Equations (41) and (42) and Equations (44) and (45), we get for all $y \in \{0, 1\}$

$$\Pr[X_i = r, \widetilde{Z}_i = s, Y_i = y] = \Pr_{\mathcal{Q}}[X = r, \widetilde{Z} = s, Y = y].$$

It remains to prove Equation (43). Before proving it, we recall the following invariant from the statement of this proposition: For all $r \in \mathcal{X}$, it holds that

$$\Pr_{\mathcal{P}}[X = r] = \Pr_{\mathcal{Q}}[X = r]. \tag{46}$$

Consider $\Pr[X_i = r, \widetilde{Z}_i = 1]$ for some $r \in \mathcal{X}$. From the algorithm used by the adversary, we have

$$
\begin{aligned}
\Pr[X_i = r, \widetilde{Z}_i = 1] &= (1 - p(a, 2)) \cdot \Pr[(X_i, Z_i) = (r, 2)] + p(a, 1) \cdot \Pr[(X_i, Z) = (r, 1)] \\
&= (1 - p(a, 2)) \cdot \Pr_{\mathcal{P}}[(X, Z) = (r, 2)] + p(a, 1) \cdot \Pr_{\mathcal{P}}[(X, Z) = (r, 1)]. \\
&\qquad \text{(For all } i \in [N],\ (X_i, Y_i, Z_i) \sim \mathcal{P})\ (47)
\end{aligned}
$$

We consider two cases.

**(Case A)** $\Pr_{\mathcal{Q}}[(X, Z) = (a, 1)] \geq \Pr_{\mathcal{P}}[(X, Z) = (a, 1)]$**:** In this case, we have $p(a, 1) = 1$.

$$
\begin{aligned}
\Pr[X_i = r, \widetilde{Z}_i = 1] &\overset{(47)}{=} (1 - p(a, 2)) \cdot \Pr_{\mathcal{P}}[(X, Z) = (r, 2)] + p(a, 1) \cdot \Pr_{\mathcal{P}}[(X, Z) = (r, 1)] \\
&= \Pr_{\mathcal{P}}[(X, Z) = (r, 2)] + \Pr_{\mathcal{P}}[(X, Z) = (r, 1)] \cdot \left(1 - \frac{\Pr_{\mathcal{Q}}[(X, Z)=(r, 1)]}{\Pr_{\mathcal{P}}[(X, Z)=(r, 1)]}\right) \\
&\qquad\qquad\qquad\qquad\qquad\qquad\qquad\qquad \text{(Definition of } p(r, s)) \\
&= \Pr_{\mathcal{P}}[X = r] - \Pr_{\mathcal{Q}}[(X, Z) = (r, 2)] \\
&\overset{(46)}{=} \Pr_{\mathcal{Q}}[X = r] - \Pr_{\mathcal{Q}}[(X, Z) = (r, 2)] \\
&= \Pr_{\mathcal{Q}}[(X, Z) = (r, 1)]
\end{aligned}
$$

**(Case B)** $\Pr_{\mathcal{Q}}[(X, Z) = (a, 1)] < \Pr_{\mathcal{P}}[(X, Z) = (a, 1)]$**:** In this case, we have $p(a, 1) < 1$.

$$
\begin{aligned}
\Pr[X_i = r, \widetilde{Z}_i = 1] &\overset{(47)}{=} (1 - p(a, 2)) \cdot \Pr_{\mathcal{P}}[(X, Z) = (r, 2)] + p(a, 1) \cdot \Pr_{\mathcal{P}}[(X, Z) = (r, 1)] \\
&= \Pr_{\mathcal{P}}[(X, Z) = (r, 1)] \cdot \frac{\Pr_{\mathcal{Q}}[(X, Z) = (r, 1)]}{\Pr_{\mathcal{P}}[(X, Z) = (r, 1)]} \qquad \text{(Definition of } p(r, s)) \\
&= \Pr_{\mathcal{Q}}[(X, Z) = (r, 1)].
\end{aligned}
$$

In both, cases, we have $\Pr[X_i = r, \widetilde{Z}_i = 1] = \Pr_{\mathcal{Q}}[(X, Z) = (r, 1)]$. By swapping the protected labels, we can show that $\Pr[X_i = r, \widetilde{Z}_i = 2] = \Pr_{\mathcal{Q}}[(X, Z) = (r, 2)]$. This proves Equation (43). $\qquad\square$

From Lemma A.16, with probability at least $1 - \delta$, $\widehat{S} := \{(X_i, \widetilde{Z}_i, Y_i)\}_{i \in [N]}$. By Lemma A.17, the samples $\{(X_i, \widetilde{Z}_i, Y_i)\}_{i \in [N]}$ are iid from $\mathcal{Q}$. Thus, Proposition A.15 follows. $\qquad\square$

### A.2.3   Proof of Lemma A.13

*Proof of Lemma A.13.* Our goal is to show that for every $f \in \mathcal{F}$, there exists a choice $k \in [3]$, such that, $f$ has error at least $c(1 - \alpha)$ or statistical rate at most $c + \alpha$ with respect to $\mathcal{D}_k$. Since $\mathcal{D}_1$, $\mathcal{D}_2$, and $\mathcal{D}_3$ are supported on subsets of $\{x_A, x_B, x_C\} \times [2]$, it suffices to consider the restriction of $\mathcal{F}$ on this domain. There at most $2^6$ classifiers in this restriction. We partition them into three cases.

**(Case A)** $f(x_B, 1) = 1$**:** For any $k \in [3]$, we have

$$
\begin{aligned}
\mathrm{Err}_{\mathcal{D}_k}(f) &= \sum_{r \in \mathcal{X}, s \in [p]} \Pr_{\mathcal{D}_k}[f(X, Z) \neq Y \mid (X, Z) = (r, s)] \cdot \Pr[(X, Z) = (r, s)] \\
&\geq \Pr_{\mathcal{D}_k}[f(X, Z) \neq Y \mid X = x_B, Z = 1] \cdot \Pr[X = x_B, Z = 1] \\
&\overset{(40)}{\geq} \Pr[X = x_B, Z = 1] \qquad\qquad \text{(Using that, in this case, } f(x_B, 1) = 1) \\
&\overset{\text{Table 2}}{\geq} (1 - c) \cdot (1 - \alpha) \\
&> c(1 - \alpha). \qquad\qquad\qquad\qquad\qquad\qquad \text{(Using that } c < 1/2)
\end{aligned}
$$

Thus, in Case A, $f$ has an error larger than $c \cdot (1 - \alpha)$ on each of $\mathcal{D}_1, \mathcal{D}_2$, and $\mathcal{D}_3$.

**(Case B)** $f(x_A, 1) = 0$ **and** $f(x_B, 1) = 0$**:** For any $k \in [3]$, we have

$$
\begin{aligned}
\mathrm{Err}_{\mathcal{D}_k}(f) \quad &= \quad \sum_{r \in \mathcal{X}, s \in [p]} \mathrm{Pr}_{\mathcal{D}_k}\left[ f(X, Z) \neq Y \mid (X, Z) = (r, s) \right] \cdot \mathrm{Pr}\left[ (X, Z) = (r, s) \right] \\
&\geq \quad \mathrm{Pr}_{\mathcal{D}_k}\left[ f(X, Z) \neq Y \mid X = x_A, Z = 1 \right] \cdot \mathrm{Pr}\left[ X = x_A, Z = 1 \right] \\
&\overset{(40)}{\geq} \quad \mathrm{Pr}\left[ X = x_A, Z = 1 \right] \qquad\qquad \text{(Using that, in this case, } f(x_B, 1) = 1) \\
&\overset{\text{Table 2}}{\geq} \quad c(1 - \alpha).
\end{aligned}
$$

Thus, in Case B, $f$ has an error larger than $c \cdot (1 - \alpha)$ on each of $\mathcal{D}_1, \mathcal{D}_2$, and $\mathcal{D}_3$.

**(Case C)** $f(x_A, 1) = 1$ **and** $f(x_B, 1) = 0$**:**

**(Case C.1)** $\sum_{r \in \mathcal{X}} f(r, 2) \geq 2$**:** In this case, $f$ takes a value of 1 on at least two points in the tuple

$$
L = ((x_A, 2), (x_B, 2), (x_C, 2)).
$$

If $f$ takes a value of 0 on a point in $L$, then fix $j \in [3]$ such that $h(L_j) = 0$. Let $k := 4 - j$. Consider the distribution $\mathcal{D}_k$. Notice that by our construction $\mathrm{Pr}_{\mathcal{D}_k}[L_i] = 0$ (see Table 2). Since $L_i$ has measure 0 for $\mathcal{D}_k$, the value of $f$ at this point does not affect its accuracy or statistical rate on $\mathcal{D}_k$. Thus, we can assume that $f(L_i) = 1$. Or in other words, we can assume that

$$
\text{For all } r \in \mathcal{X}, \quad f(r, 2) = 1. \tag{48}
$$

We compute the performance of $f$ on both protected groups $\ell \in [p]$. For $Z = 2$, we have

$$
\begin{aligned}
\mathbb{E}_{\mathcal{D}_k}\left[ f(X, Z) \mid Z = 2 \right] \quad &= \quad \sum_{r \in \mathcal{X}} f(r, 2) \cdot \mathrm{Pr}_{\mathcal{D}_k}\left[ X = r \mid Z = 2 \right] \\
&\overset{(48)}{=} \quad \sum_{r \in \mathcal{X}} 1 \cdot \mathrm{Pr}_{\mathcal{D}_k}\left[ X = r \mid Z = 2 \right] \\
&\overset{\text{Table 2}}{=} \quad 1. \tag{49}
\end{aligned}
$$

For $Z = 1$, we have

$$
\begin{aligned}
\mathbb{E}_{\mathcal{D}_k}\left[ f(X, Z) \mid Z = 1 \right] \quad &= \quad \sum_{r \in \mathcal{X}} f(r, 1) \cdot \mathrm{Pr}_{\mathcal{D}_k}\left[ X = r \mid Z = 1 \right] \\
&= \quad 1 \cdot \mathrm{Pr}_{\mathcal{D}_k}\left[ X = x_A \mid Z = 1 \right] + f(x_C, 1) \cdot \mathrm{Pr}_{\mathcal{D}_k}\left[ X = x_C \mid Z = 1 \right] \\
&\qquad\qquad\qquad\qquad\quad \text{(In this case, } f(x_A, 1) = 1 \text{ and } f(x_B, 1) = 0) \\
&\leq \quad 1 \cdot \mathrm{Pr}_{\mathcal{D}_k}\left[ X = x_A \mid Z = 1 \right] + \mathrm{Pr}_{\mathcal{D}_k}\left[ X = x_C \mid Z = 1 \right] \\
&\qquad\qquad\qquad\qquad\qquad\qquad\qquad \text{(Using that } f(x_C, 1) \leq 1) \\
&\overset{\text{Table 2}}{\leq} \quad \max\left\{ \tfrac{c \cdot (1 - \alpha) + \alpha/2}{1 - \alpha/2}, \tfrac{c \cdot (1 - \alpha) + c\alpha/2}{1 - \alpha/2}, \tfrac{c \cdot (1 - \alpha/2) + \alpha \cdot (1 - c)/2}{1 - \alpha/2} \right\} \\
&= \quad \frac{c \cdot (1 - \alpha) + \alpha/2}{1 - \alpha/2} \qquad\qquad\qquad\qquad\quad \text{(Using } c, \alpha > 0) \\
&< \quad c + \alpha. \qquad\qquad\qquad \text{(Using } c, \alpha > 0 \text{ and } \alpha \leq 1) \tag{50}
\end{aligned}
$$

Since $c < 1/2$ and $\alpha \leq \min\{\eta, 1/2\}$, we have

$$
c + \alpha < 1. \tag{51}
$$

Now, we can compute the statistical rate of $f$ using Equations (49), (50), and (51).

$$
\Omega_{\mathcal{D}_k}(f) = \frac{\min_{\ell \in [p]} \mathbb{E}_{\mathcal{D}_k}\left[ f(X, Z) \mid Z = \ell \right]}{\max_{\ell \in [p]} \mathbb{E}_{\mathcal{D}_k}\left[ f(X, Z) \mid Z = \ell \right]} \overset{(49),(50),(51)}{<} \frac{c + \alpha}{1}.
$$

Thus, in Case C.1, $f$ has a statistical rate smaller than $c + \alpha$ on distribution $\mathcal{D}_k$.

**(Case C.2)** $\sum_{r \in \mathcal{X}} f(r, 2) \leq 1$**:** Thus, $f$ takes a value of 0 on at least two points in the list

$$
L = ((x_A, 2), (x_B, 2), (x_C, 2)).
$$

If $f$ takes a value of 1 on one of the points in $L$, then fix $j \in [3]$ such that $f(L_j) = 0$. Let $k := 4 - k$ Consider the distribution $\mathcal{D}_k$. Notice that by our construction $\Pr_{\mathcal{D}_k}[L_j] = 0$ (see Table 2). Since $L_i$ has measure 0 on $\mathcal{D}_k$, the value of $f$ at this point does not affect its accuracy or statistical rate on $\mathcal{D}_k$. Thus, we can assume that $f(L_j) = 0$. Or in other words, we can assume that

$$\text{For all } r \in \mathcal{X}, \quad f(r, 2) = 0. \tag{52}$$

We would like to compute the statistical rate of $f$. Toward this, we first compute the performance of $f$ on both protected groups. For $Z = 2$, we have

$$\mathbb{E}_{\mathcal{D}_i}\left[f(X, Z) \mid Z = 2\right] = \sum_{r \in \mathcal{X}} f(r, 2) \cdot \Pr\left[X = r \mid Z = 2\right]$$
$$\overset{(52)}{=} 0. \tag{53}$$

For $Z = 1$, we have

$$\begin{aligned}
\mathbb{E}_{\mathcal{D}_k}\left[f(X, Z) \mid Z = 1\right] &= \sum_{r \in \mathcal{X}} f(r, 1) \cdot \Pr_{\mathcal{D}_k}\left[X = r \mid Z = 1\right] \\
&= 1 \cdot \Pr_{\mathcal{D}_k}\left[X = x_A \mid Z = 1\right] + f(x_C, 1) \cdot \Pr_{\mathcal{D}_k}\left[X = x_C \mid Z = 1\right] \\
&\quad \text{(Using that, in this case, } f(x_A, 1) = 1 \text{ and } f(x_B, 1) = 0\text{)} \\
&\geq \Pr_{\mathcal{D}_k}\left[X = x_A \mid Z = 1\right] \\
&\quad \text{(Using } f(x_C, 1) \cdot \Pr\left[X = x_C \mid Z = 1\right] \geq 0\text{)} \\
&\overset{\text{Table 2}}{\geq} \max\left\{\frac{c \cdot (1 - \alpha)}{1 - \alpha/2}, \frac{c(1 - \alpha)}{1 - \alpha/2}, \frac{c \cdot (1 - \alpha/2)}{1 - \alpha/2}\right\} \\
&= \frac{c \cdot (1 - \alpha)}{1 - \alpha/2} \quad \text{(Using that } \alpha, c > 0\text{)} \\
&> c. \quad \text{(Using that } \alpha, c > 0\text{)} \tag{54}
\end{aligned}$$

Now, we can compute the statistical rate of $f$ using Equations (54) and (53).

$$\Omega_{\mathcal{D}_i}(f) = \frac{\min_{\ell \in [p]} \mathbb{E}_{\mathcal{D}_k}\left[f(X, Z) \mid Z = \ell\right]}{\max_{\ell \in [p]} \mathbb{E}_{\mathcal{D}_k}\left[f(X, Z) \mid Z = \ell\right]} \overset{(54),(53),(c>0)}{\leq} 0$$

Thus, in Case C.2, $f$ has a statistical rate 0 on the distribution $\mathcal{D}_k$.

Across all cases, we proved that all $2^6$ classifiers in the restriction of $\mathcal{F}$ to $\{x_A, x_B, x_C\} \times [2]$, either have a error larger than $c \cdot (1 - \alpha)$ or statistical rate smaller than $c + \alpha$ on one of $\mathcal{D}_1, \mathcal{D}_2,$ or $\mathcal{D}_3$. $\quad\square$

### A.2.4 Proof of Lemma A.14

*Proof of Lemma A.14.* For each distribution $\mathcal{D}_1, \mathcal{D}_2,$ and $\mathcal{D}_3$, we will give an classifier $f \in \mathcal{F}$ which satisfies the condition in Lemma A.14.

**(Case A) $\mathcal{D}_1$:** Define $f$ as $f(x, z) := \mathbb{I}[x = x_A]$. Comparing this to Equation (40), we get that

$$\text{Err}_{\mathcal{D}_1}(f) = 0.$$

For $Z = 2$, we have

$$\begin{aligned}
\mathbb{E}_{\mathcal{D}_1}\left[f(X, Z) \mid Z = 2\right] &= \sum_{r \in \mathcal{X}} f(r, 2) \cdot \Pr_{\mathcal{D}_1}\left[X = r \mid Z = 2\right] \\
&= \Pr_{\mathcal{D}_1}\left[X = x_A \mid Z = 2\right] \\
&\overset{\text{Table 2}}{=} c. \tag{55}
\end{aligned}$$

Similarly, for $Z = 1$, we have

$$\begin{aligned}
\mathbb{E}_{\mathcal{D}_1}\left[f(X, Z) \mid Z = 1\right] &= \sum_{r \in \mathcal{X}} f(r, 1) \cdot \Pr_{\mathcal{D}_1}\left[X = r \mid Z = 1\right] \\
&= \Pr_{\mathcal{D}_1}\left[X = x_A \mid Z = 1\right] \\
&\overset{\text{Table 2}}{=} \frac{c(1 - \alpha)}{(1 - \alpha/2)}. \tag{56}
\end{aligned}$$

Thus, we have

$$
\begin{aligned}
\Omega_{\mathcal{D}_1}(h) &= \frac{\min_{\ell \in [p]} \mathbb{E}_{\mathcal{D}_1}\left[f(X,Z) \mid Z = \ell\right]}{\max_{\ell \in [p]} \mathbb{E}_{\mathcal{D}_1}\left[f(X,Z) \mid Z = \ell\right]} \\
&\overset{(55),(56)}{=} \frac{c(1-\alpha)}{(1-\alpha/2)} \cdot \frac{1}{c} && \text{(Using that } c, \alpha > 0) \\
&= 1 - \frac{\alpha}{(2-\alpha)} \\
&\geq 1 - \alpha. && \text{(Using that } \alpha \leq 1)
\end{aligned}
$$

**(Case B)** $\mathcal{D}_2$**:** Define $f$ as $f(x,z) := \mathbb{I}[x = x_A]$. Comparing this to Equation (40), we get that

$$
\mathrm{Err}_{\mathcal{D}_1}(f) = 0.
$$

For $Z = 2$, we have

$$
\begin{aligned}
\mathbb{E}_{\mathcal{D}_2}\left[f(X,Z) \mid Z = 2\right] &= \sum_{r \in \mathcal{X}} f(r,2) \cdot \mathrm{Pr}_{\mathcal{D}_2}\left[X = r \mid Z = 2\right] \\
&= \mathrm{Pr}_{\mathcal{D}_2}\left[X = x_A \mid Z = 2\right] \\
&\overset{\text{Table 2}}{=} c. && (57)
\end{aligned}
$$

Similarly, for $Z = 1$, we have

$$
\begin{aligned}
\mathbb{E}_{\mathcal{D}_2}\left[f(X,Z) \mid Z = 1\right] &= \sum_{r \in \mathcal{X}} f(r,1) \cdot \mathrm{Pr}_{\mathcal{D}_2}\left[X = r \mid Z = 1\right] \\
&= \mathrm{Pr}_{\mathcal{D}_2}\left[X = x_A \mid Z = 1\right] \\
&\overset{\text{Table 2}}{=} \frac{c \cdot (1-\alpha)}{(1-\alpha/2)}. && (58)
\end{aligned}
$$

Thus, we have

$$
\begin{aligned}
\Omega_{\mathcal{D}_1}(h) &= \frac{\min_{\ell \in [p]} \mathbb{E}_{\mathcal{D}_2}\left[f(X,Z) \mid Z = \ell\right]}{\max_{\ell \in [p]} \mathbb{E}_{\mathcal{D}_2}\left[f(X,Z) \mid Z = \ell\right]} \\
&\overset{(57),(58)}{=} \frac{c \cdot (1-\alpha)}{(1-\alpha/2)} \cdot \frac{1}{c} && \text{(Using } c, \alpha > 0) \\
&\geq 1 - \alpha. && \text{(Using that } \alpha \leq 1)
\end{aligned}
$$

**(Case C)** $\mathcal{D}_3$**:** Define $f \in \mathcal{F}$ to be the following classifier

$$
f(x,z) := \begin{cases} 1 & \text{if } (x,z) \in \{(x_A,1),(x_C,2)\}, \\ 0 & \text{otherwise.} \end{cases} \tag{59}
$$

Such an $f \in \mathcal{F}$ exists because $\mathcal{F}$ shatters the set $\{x_A, x_B, x_C\} \times [2]$. We have that

$$
\begin{aligned}
\mathrm{Err}_{\mathcal{D}_3}(f) &= \sum_{r \in \mathcal{X}, s \in [p]} \mathrm{Pr}_{\mathcal{D}_3}[f(r,s) \neq Y \mid (X,Z) = (r,s)] \cdot \mathrm{Pr}_{\mathcal{D}_3}[(X,Z) = (r,s)] \\
&\overset{(40),(59)}{=} \mathrm{Pr}_{\mathcal{D}_3}[(X,Z) = (x_A,2)] + \mathrm{Pr}_{\mathcal{D}_3}[(X,Z) = (x_C,2)] \\
&\overset{\text{Table 2}}{=} \frac{c\alpha}{2}.
\end{aligned}
$$

Further, for $Z = 2$, we have

$$
\begin{aligned}
\mathbb{E}_{\mathcal{D}_3}\left[f(X,Z) \mid Z = 2\right] &= \sum_{r \in \mathcal{X}} f(r,2) \cdot \mathrm{Pr}\left[X = r \mid Z = 2\right] \\
&\overset{(59)}{=} \mathrm{Pr}_{\mathcal{D}_3}\left[X = x_C \mid Z = 2\right] \\
&\overset{\text{Table 2}}{=} c. && (60)
\end{aligned}
$$

Similarly, for $Z = 1$, we have

$$
\begin{aligned}
\mathbb{E}_{\mathcal{D}_3}\left[f(X, Z) \mid Z = 1\right] &= \sum_{r \in \mathcal{X}} f(r, 1) \cdot \Pr\left[X = r \mid Z = 1\right] \\
&\overset{(59)}{=} \Pr_{\mathcal{D}_3}\left[X = x_A \mid Z = 1\right] \\
&\overset{\text{Table 2}}{=} \frac{c \cdot (1 - \alpha/2)}{(1 - \alpha/2)} \\
&= c.
\end{aligned}
\tag{61}
$$

Thus, we can compute $\Omega_{\mathcal{D}_3}(f)$ as follows

$$
\Omega_{\mathcal{D}_3}(f) = \frac{\min_{\ell \in [p]} \mathbb{E}_{\mathcal{D}_3}\left[f(X, Z) \mid Z = \ell\right]}{\max_{\ell \in [p]} \mathbb{E}_{\mathcal{D}_3}\left[f(X, Z) \mid Z = \ell\right]} \overset{(60),(61)}{=} \frac{c}{c} = 1.
$$

Thus, for each $\mathcal{D} \in \{\mathcal{D}_1, \mathcal{D}_2, \mathcal{D}_3\}$, we give a classifier $f \in \mathcal{F}$ such that that has error at most $(c\alpha)/2$ and a statistical rate at least $1 - \alpha$. $\qquad\square$

## A.3  Proof of Theorem 4.5

In Theorem 4.5, the hypothesis class $\mathcal{F}$ that shatters the set $\{x_A, x_B, x_C, x_D, x_E\} \times [2] \subseteq \mathcal{X} \times [p]$, Assumption 1 holds with a constant $\lambda \in (0, 1/4)$, the fairness threshold $\tau = 1$, and statistical rate is the fairness metric; recall that the statistical rate of a classifier $f \in \mathcal{F}$ is

$$
\Omega_{\mathcal{D}}(f) := \frac{\min_{\ell \in [p]} \Pr_{\mathcal{D}}[f = 1 \mid Z = \ell]}{\max_{\ell \in [p]} \Pr_{\mathcal{D}}[f = 1 \mid Z = \ell]}.
$$

Then, given parameters $\eta \in (0, 1]$ and $\delta \in [0, 1/2)$, our goal is to show that for any

$$
\varepsilon < \frac{1}{4} - \frac{2\eta}{5} \text{ and } v < \frac{\eta}{10\lambda} \cdot (1 - 4\lambda) - O\left(\frac{\eta^2}{\lambda^2}\right),
\tag{62}
$$

$\mathcal{F}$ is not $(\varepsilon, \nu)$-learnable with perturbation rate $\eta$ and confidence $\delta$. We prove a more general result: Given parameters $\eta \in (0, 1]$, $\delta \in [0, 1/2)$, for any $c \in (0, \min\{\eta, 2\lambda/9\}]$, we show that for any

$$
0 < \varepsilon < \frac{1}{2} - \lambda - 2c \text{ and } 0 < v < \frac{c(1 - 4\lambda)}{2\lambda} - \frac{3c^2}{4\lambda^2},
\tag{63}
$$

$\mathcal{F}$ is not $(\varepsilon, \nu)$-learnable with perturbation rate $\eta$ and confidence $\delta$. Setting $c = 2\eta/5$ recovers Theorem 4.5.

**Remark A.18.** *The proof of Theorem 4.5 has a similar structure to the proof of Theorem 4.4, but the specific distributions constructed are different from those in the proof of Theorem 4.4. The proof of Theorem 4.5 also borrows Proposition A.15 from the proof of Theorem 4.4; Proposition A.15 is proved in Section A.2.2.*

### A.3.1  Proof of Theorem 4.5

*Proof of Theorem 4.5.* Let $\mathcal{L}$ be any learner. Fix any constant

$$
c \in (0, \min\{\eta, 2\lambda/9\}],
\tag{64}
$$

and confidence parameter $\delta \in (0, 1/2)$ (in Definition 3.2). We construct three distributions $\mathcal{D}_1, \mathcal{D}_2$, and $\mathcal{D}_3$ (parameterized by $c$) that satisfy the requirements of Proposition A.15:

1. The total variation distance between any two distributions is bounded by $\eta$,

2. the distributions have the same marginal on $\mathcal{X}$, and

3. the label $Y$ is independent of the protected attribute $Z$ conditioned on features $X$.

Suppose $\mathcal{L}$ is given $N$ samples, where

$$
N \geq 3 \cdot \ln 2 \cdot (\eta - c)^{-2}.
\tag{65}
$$

Consider three cases, depending on whether the samples in $S$ are iid from $\mathcal{D}_1$, $\mathcal{D}_2$, or $\mathcal{D}_3$.

By Proposition A.15, in each case, there is an adversary $A \in \mathcal{A}(\eta)$, that can ensure that, with probability at least $1/2$, the perturbed samples $\widehat{S} := A(S)$ have the same distribution as iid samples from $\mathcal{D}_1$. Thus, given $\widehat{S}$, $\mathcal{L}$ cannot differentiate between the three cases with probability at least $1/2$. As a result, with probability at least $1/2$, $\mathcal{L}$ outputs the same classifier, say $f_{\text{Com}} \in \mathcal{F}$, in each case. We show that no $f_{\text{Com}} \in \mathcal{F}$ satisfies the accuracy and fairness guarantee in all three cases.

**Lemma A.19 (No good classifier for all cases).** *There is no $f \in \mathcal{F}$, such that, for all $k \in [3]$, $\Pr_{\mathcal{D}_k}[h(X, Z) \neq Y] < 1/2 - \lambda - c/2$ and $\Omega_{\mathcal{D}_k}(f) < 1 - c(1-4\lambda)/(2\lambda) + 3c^2/(4\lambda^2)$.*

**Lemma A.20 (A good classifier for each case).** *For each $k \in [3]$, there is an $f \in \mathcal{F}$ such that $\Pr_{\mathcal{D}_k}[h(X, Z) \neq Y] \leq 3\alpha/2$ and $\Omega_{\mathcal{D}}(h) = 1$.*

Note that for each $k \in [3]$, $h_k$ satisfies the fairness constraint. Thus, an optimal solution $f_k^\star \in \mathcal{F}$ of Program (2) for $\mathcal{D}_k$ and $\tau = 1$ must satisfy

$$\text{Err}_{\mathcal{D}_k}(f_k^\star) < \frac{3c}{2}. \tag{66}$$

(Otherwise, we have a contradiction as $h_k$ satisfies the fairness constraints and has a smaller error than $f_k^\star$.) If $\mathcal{L}$ is a $(\varepsilon, \nu)$-learner, then ibn the $k$-th case, with probability at least $1 - \delta > 1/2$, $\mathcal{L}$ must output a classifier $f_k$ which satisfies

$$\text{Err}_{\mathcal{D}_k}(f_k) - \text{Err}_{\mathcal{D}_k}(f_k^\star) \leq \varepsilon \text{ and } \tau - \Omega_{\mathcal{D}_k}(f_k) \leq \nu.$$

However, in all cases with probability at least $1/2$, $\mathcal{L}$ outputs $f_{\text{Com}}$. Because $1/2 > \delta$, $f_{\text{Com}}$ must satisfy:

$$\text{For all } k \in [3], \quad \text{Err}_{\mathcal{D}_k}(f_{\text{Com}}) - \text{Err}_{\mathcal{D}_k}(f_k^\star) \leq \varepsilon \text{ and } \tau - \Omega_{\mathcal{D}_k}(f_{\text{Com}}) \leq \nu. \tag{67}$$

But from Lemma A.19 we know that for each $k \in [3]$ either

$$\text{Err}_{\mathcal{D}_k}(f_{\text{Com}}) \geq \frac{1}{2} - \lambda - \frac{c}{2} \quad \text{or} \quad \Omega_{\mathcal{D}_k}(f_{\text{Com}}) \leq 1 - \frac{c(1-4\lambda)}{2\lambda} + \frac{3c^2}{4\lambda^2}. \tag{68}$$

**(Case A)** $\text{Err}_{\mathcal{D}_k}(f_{\text{Com}}) \geq 1/2 - \lambda - c/2$**:** In this case, from Equation (67), we have

$$\begin{aligned}
\varepsilon &\geq \frac{1}{2} - \lambda - \frac{c}{2} - \text{Err}_{\mathcal{D}_k}(f_k^\star) \\
&\overset{(66)}{>} \frac{1}{2} - \lambda - \frac{c}{2} - \frac{3c}{2} \\
&= \frac{1}{2} - \lambda - 2c \qquad\qquad \text{(Using that } c < 1/2 \text{ and } \alpha > 0) \quad (69)
\end{aligned}$$

But because $\varepsilon \leq \frac{1}{2} - \lambda - 2c$ (see Equation (63)), Equation (69) cannot hold.

**(Case B)** $\Omega_{\mathcal{D}_k}(f_{\text{Com}}) \leq 1 - c(1-4\lambda)/(2\lambda) + 3c^2/(4\lambda^2)$**:** In this case, from Equation (67), we have

$$\nu \geq \frac{c(1-4\lambda)}{2\lambda} - \frac{3c^2}{4\lambda^2}. \tag{70}$$

But because $\nu < c(1-4\lambda)/(2\lambda) - 3c^2/(4\lambda^2)$ (see Equation (63)), Equation (70) does not hold. Therefore, we have a contradiction. Hence, $\mathcal{L}$ is not an $(\varepsilon, \nu)$-learner for $\mathcal{F}$. Since the choice of $\mathcal{L}$ was arbitrary, we have shown that there is no learner which $(\varepsilon, \nu)$-learns $\mathcal{F}$. It remains to prove the Lemmas A.19 and A.20. $\qquad\square$

### A.3.2 Proof of Lemma A.19

Fix any $c \in (0, \min\{\eta, 2\lambda/9\}]$. Set $\mathcal{D}_1$, $\mathcal{D}_2$, and $\mathcal{D}_3$ to be unique distributions with marginal distribution specified in Table 3, such that, for any draw $(X, Y, Z) \sim \mathcal{D}_k$ ($k \in [3]$) $Y$ takes the value $\mathbb{I}[X \neq x_E]$, i.e.,

$$Y = \begin{cases} 1 & \text{if } X \neq x_E, \\ 0 & \text{otherwise.} \end{cases} \tag{71}$$

In particular, the construction of $\mathcal{D}_1$, $\mathcal{D}_2$, and $\mathcal{D}_3$ ensures that the total variation distance between any pair of distributions is less than $c$.

(a) $\mathcal{D}_1$: $\mathrm{Pr}_{\mathcal{D}_1}[(X,Z)=(r,s)]$ for $(r,s) \in \{x_A, x_B, x_C, x_D, x_E\} \times [2]$.

|   | $x_A$ | $x_B$ | $x_C$ | $x_D$ | $x_E$ |
|---|-------|-------|-------|-------|-------|
| 1 | $0$ | $c/2$ | $c/2$ | $1/2-\lambda-c/2$ | $1/2-\lambda-c/2$ |
| 2 | $c$ | $c/2$ | $c/2$ | $\lambda - c$ | $\lambda - c$ |

(b) $\mathcal{D}_2$: $\mathrm{Pr}_{\mathcal{D}_2}[(X,Z)=(r,s)]$ for $(r,s) \in \{x_A, x_B, x_C, x_D, x_E\} \times [2]$.

|   | $x_A$ | $x_B$ | $x_C$ | $x_D$ | $x_E$ |
|---|-------|-------|-------|-------|-------|
| 1 | $c/2$ | $0$ | $c/2$ | $1/2-\lambda-c/2$ | $1/2-\lambda-c/2$ |
| 2 | $c/2$ | $c$ | $c/2$ | $\lambda - c$ | $\lambda - c$ |

(c) $\mathcal{D}_3$: $\mathrm{Pr}_{\mathcal{D}_3}[(X,Z)=(r,s)]$ for $(r,s) \in \{x_A, x_B, x_C, x_D, x_E\} \times [2]$.

|   | $x_A$ | $x_B$ | $x_C$ | $x_D$ | $x_E$ |
|---|-------|-------|-------|-------|-------|
| 1 | $c/2$ | $c/2$ | $0$ | $1/2-\lambda-c/2$ | $1/2-\lambda-c/2$ |
| 2 | $c/2$ | $c/2$ | $c$ | $\lambda - c$ | $\lambda - c$ |

Table 3: Marginal distributions of $\mathcal{D}_1, \mathcal{D}_2, \mathcal{D}_3$ over $\mathcal{X} \times [2]$. Recall that for a sample $(X, Y, Z) \sim \mathcal{D}_k$ ($k \in [3]$), $Y$ takes the value $\mathbb{I}[X \neq x_E]$.

*Proof of Lemma A.19.* Our goal is to show that for every $f \in \mathcal{F}$, there is a $k \in [3]$, such that, $f$ has error at least $1/2 - \lambda - c/2$ or statistical rate at most $1 + c(1-4\lambda)/(2\lambda) + \frac{3c^2}{(4\lambda^2)}$ with respect to $\mathcal{D}_k$. Since $\mathcal{D}_1, \mathcal{D}_2$, and $\mathcal{D}_3$ are supported on subsets of $\{x_A, x_B, x_C, x_D, x_E\} \times [2]$, it suffices to consider the restriction of $\mathcal{F}$ to $\{x_A, x_B, x_C, x_D, x_E\} \times [2]$. There at most $2^{10}$ classifiers in this restriction. We partition them into four cases.

**(Case A)** $f(x_D, 1) = 0$ **or** $f(x_E, 1) = 1$:

**(Case A.1)** $f(x_D, 1) = 0$: For any $k \in [3]$ it holds that

$$
\begin{aligned}
\mathrm{Err}_{\mathcal{D}_k}(f) &= \sum_{r \in \mathcal{X}, s \in [p]} \mathrm{Pr}_{\mathcal{D}_k}[f(X, Z) \neq Y \mid (X, Z) = (r, s)] \cdot \mathrm{Pr}[(X, Z) = (r, s)] \\
&\geq \mathrm{Pr}_{\mathcal{D}_k}[f(X, Z) \neq Y \mid X = x_D, Z = 1] \cdot \mathrm{Pr}[X = x_D, Z = 1] \\
&\qquad\qquad (\forall\, x \in \mathcal{X}, f(x, 2) \geq 0) \\
&\overset{(71)}{\geq} \mathrm{Pr}[X = x_D, Z = 1] \qquad (\text{Using that, in this case, } f(x_D, 1) = 0) \\
&\overset{\text{Table 3}}{=} 1/2 - \lambda - c/2.
\end{aligned}
$$

Thus, in Case A.1, $f$ has error at least $1/2 - \lambda - c/2$ on each of $\mathcal{D}_1, \mathcal{D}_2$, and $\mathcal{D}_3$.

**(Case A.2)** $f(x_E, 1) = 1$: For any $k \in [3]$ it holds that

$$
\begin{aligned}
\mathrm{Err}_{\mathcal{D}_k}(f) &= \sum_{r \in \mathcal{X}, s \in [p]} \mathrm{Pr}_{\mathcal{D}_k}[f(X, Z) \neq Y \mid (X, Z) = (r, s)] \cdot \mathrm{Pr}[(X, Z) = (r, s)] \\
&\geq \mathrm{Pr}_{\mathcal{D}_k}[f(X, Z) \neq Y \mid X = x_E, Z = 1] \cdot \mathrm{Pr}[X = x_E, Z = 1] \\
&\qquad\qquad (\forall\, x \in \mathcal{X}, f(x, 2) \geq 0) \\
&\overset{(71)}{\geq} \mathrm{Pr}[X = x_E, Z = 1] \qquad (\text{Using that, in this case, } f(x_E, 1) = 1) \\
&\overset{\text{Table 3}}{=} 1/2 - \lambda - c/2.
\end{aligned}
$$

Thus, in Case A.2, $f$ has error at least $1/2 - \lambda - c/2$ on each of $\mathcal{D}_1, \mathcal{D}_2$, and $\mathcal{D}_3$.

In the rest of the cases, assume that $f(x_D, 1) = 1$ and $f(x_E, 1) = 0$.

**(Case B)** $f(x_D, 2) = f(x_E, 2)$:

**(Case B.1)** $f(x_D, 2) = f(x_E, 2) = 0$**:** For any $k \in [3]$ and $Z = 2$ it holds that

$$
\begin{aligned}
\mathbb{E}_{\mathcal{D}_k}\left[f(X, Z) \mid Z = 2\right] \quad &= \quad \sum_{r \in \mathcal{X}} f(r, 2) \cdot \Pr_{\mathcal{D}_k}\left[X = r \mid Z = 2\right] \\
&= \quad \sum_{r \in \mathcal{X} \setminus \{x_D, x_E\}} f(r, 2) \cdot \Pr_{\mathcal{D}_k}\left[X = r \mid Z = 2\right] \\
&\qquad\qquad \text{(Using that, in this case, } f(x_D, 2) = f(x_E, 2) = 0) \\
&\leq \quad \sum_{r \in \mathcal{X} \setminus \{x_D, x_E\}} \Pr_{\mathcal{D}_k}\left[X = r \mid Z = 2\right] \\
&\qquad\qquad\qquad\qquad \text{(Using that } \forall\, r \in \mathcal{X},\ f(r, 2) \leq 1) \\
&\overset{\text{Table 3}}{\leq} \quad \frac{2c}{2\lambda}. \qquad\qquad\qquad\qquad\qquad\qquad\qquad (72)
\end{aligned}
$$

For $Z = 1$, we have

$$
\begin{aligned}
\mathbb{E}_{\mathcal{D}_k}\left[f(X, Z) \mid Z = 1\right] \quad &= \quad \sum_{r \in \mathcal{X}} f(r, 1) \cdot \Pr_{\mathcal{D}_k}\left[X = r \mid Z = 1\right] \\
&\geq \quad 1 \cdot \Pr_{\mathcal{D}_k}\left[X = x_D \mid Z = 1\right] \qquad \text{(From Case A, } f(x_D, 1) = 1) \\
&= \quad \frac{1 - 2\lambda - c}{2(1 - 2\lambda)} \\
&\geq \quad \frac{5/6 - 2\lambda}{2} \qquad\qquad\qquad \text{(Using that } c \leq 1/6 \text{ and } \lambda > 0) \\
&\geq \quad 1/6. \qquad\qquad\qquad\qquad\qquad \text{(Using that } \lambda \leq 1/4) \ (73)
\end{aligned}
$$

Since $c < \lambda/2$, the statistical rate of $f$ is as follows

$$
\Omega_{\mathcal{D}_k}(f) = \frac{\min_{\ell \in [p]} \mathbb{E}_{\mathcal{D}_k}\left[f(X, Z) \mid Z = \ell\right]}{\max_{\ell \in [p]} \mathbb{E}_{\mathcal{D}_k}\left[f(X, Z) \mid Z = \ell\right]} \overset{(72),(73)}{\leq} \frac{6c}{\lambda}.
$$

Thus, in Case B.1, $f$ has a statistical rate at most $\frac{6c}{\lambda}$ on each of $\mathcal{D}_1, \mathcal{D}_2,$ and $\mathcal{D}_3$.

**(Case B.2)** $f(x_D, 2) = f(x_E, 2) = 1$**:** For any $k \in [3]$ and $Z = 2$ it holds that

$$
\begin{aligned}
\mathbb{E}_{\mathcal{D}_k}\left[f(X, Z) \mid Z = 2\right] \quad &= \quad \sum_{r \in \mathcal{X}} f(r, 2) \cdot \Pr_{\mathcal{D}_k}\left[X = r \mid Z = 2\right] \\
&\geq \quad \sum_{r \in \{x_D, x_E\}} f(r, 2) \cdot \Pr_{\mathcal{D}_k}\left[X = r \mid Z = 2\right] \\
&\quad \text{(Using that } f(x_D, 2) = f(x_E, 2) = 1 \text{ and for all } r \in \mathcal{X},\ f(r, 2) \geq 0) \\
&\overset{\text{Table 3}}{=} \quad 1 - \frac{c}{\lambda}. \qquad\qquad\qquad\qquad\qquad\qquad\qquad (74)
\end{aligned}
$$

For $Z = 1$, we have

$$
\begin{aligned}
\mathbb{E}_{\mathcal{D}_k}\left[f(X, Z) \mid Z = 1\right] \quad &= \quad \sum_{r \in \mathcal{X}} f(r, 1) \cdot \Pr_{\mathcal{D}_k}\left[X = r \mid Z = 1\right] \\
&= \quad \sum_{r \in \mathcal{X} \setminus \{x_E\}} f(r, 1) \cdot \Pr_{\mathcal{D}_k}\left[X = r \mid Z = 1\right] \\
&\qquad\qquad\qquad\qquad\qquad \text{(From Case B, } f(x_E, 1) = 0) \\
&\leq \quad \sum_{r \in \mathcal{X} \setminus \{x_E\}} 1 \cdot \Pr_{\mathcal{D}_k}\left[X = r \mid Z = 1\right] \\
&\qquad\qquad\qquad\qquad \text{(Using that } \forall\, r \in \mathcal{X},\ f(r, 2) \leq 1) \\
&\overset{\text{Table 3}}{=} \quad \frac{1}{2} - \frac{c}{2(1 - 2\lambda)}. \qquad\qquad\qquad\qquad\qquad (75)
\end{aligned}
$$

Using this, we can compute an upper bound on the statistical rate of $f$ as follows

$$
\begin{aligned}
\Omega_{\mathcal{D}_k}(f) &= \frac{\min_{\ell \in [p]} \mathbb{E}_{\mathcal{D}_k}\left[f(X, Z) \mid Z = \ell\right]}{\max_{\ell \in [p]} \mathbb{E}_{\mathcal{D}_k}\left[f(X, Z) \mid Z = \ell\right]} \\
&\overset{(75),(74)}{\leq} \frac{\frac{1}{2} - \frac{c}{2(1-2\lambda)}}{1 - \frac{c}{\lambda}} \\
&= \frac{1}{2} \cdot \frac{1 - \frac{c}{(1-2\lambda)}}{1 - \frac{c}{\lambda}} \cdot \frac{1 + \frac{3c}{\lambda}}{1 + \frac{3c}{\lambda}} \\
&= \frac{1}{2} \cdot \frac{1 - \frac{c}{(1-2\lambda)}}{1 + \frac{2c}{\lambda} - \frac{3c^2}{\lambda^2}} \cdot \left(1 + \frac{3c}{\lambda}\right) \\
&\leq \frac{1}{2} + \frac{3c}{2\lambda}. \qquad \text{(Using that for all } 0 < c, \lambda \leq \tfrac{1}{4}, \text{ if } \tfrac{c}{\lambda} \leq \tfrac{1}{3}, \text{ then } \tfrac{c}{1-2\lambda} \geq \tfrac{3c^2}{\lambda^2} - \tfrac{2c}{\lambda})
\end{aligned}
$$

Thus, in Case B.2, $f$ has a statistical rate at most $\frac{1}{2} + \frac{3c}{2\lambda}$ on each of $\mathcal{D}_1, \mathcal{D}_2$, and $\mathcal{D}_3$.

In the rest of the cases, we assume that $f(x_D, 2) \neq f(x_E, 2)$.

**(Case C)** $\sum_{r \in \{x_A, x_B, x_C\}} f(r, 2) \geq 2$**:** In Case C, $f$ outputs 0 on at most one point in the tuple

$$
L = ((x_A, 2), (x_B, 2), (x_C, 2)).
$$

If $f$ takes a value of 0 on a point, say $L_j$, in $L$, then fix $k \in [3]$, such that $\mathcal{D}_k(L_j) = c/2$. (Such a distribution always exists by construction). Otherwise, fix any $k \in [3]$. For $Z = 2$, we have

$$
\begin{aligned}
\mathbb{E}_{\mathcal{D}_k}\left[f(X, Z) \mid Z = 2\right] &= \sum_{r \in \mathcal{X}} f(r, 2) \cdot \Pr_{\mathcal{D}_k}\left[X = r \mid Z = 2\right] \\
&\geq \frac{\lambda + c}{2\lambda}. \qquad \text{(By our choice of } k) \quad (76)
\end{aligned}
$$

For $Z = 1$, we have

$$
\begin{aligned}
&\mathbb{E}_{\mathcal{D}_k}\left[f(X, Z) \mid Z = 1\right] \\
&= \sum_{r \in \mathcal{X}} f(r, 1) \cdot \Pr_{\mathcal{D}_k}\left[X = r \mid Z = 1\right] \\
&= \Pr_{\mathcal{D}_k}\left[X = x_D \mid Z = 1\right] + \sum_{r \in \{x_A, x_B, x_C\}} f(r, 1) \cdot \Pr_{\mathcal{D}_k}\left[X = r \mid Z = 1\right] \\
&\qquad\qquad \text{(From previous cases, } f(x_D, 1) = 1 \text{ and } f(x_E, 1) = 0) \\
&\overset{\text{Table 3}}{\leq} \frac{(1 - 2\lambda + c)}{2(1 - 2\lambda)}. \qquad\qquad\qquad\qquad\qquad\qquad\qquad\qquad (77)
\end{aligned}
$$

We can compute the statistical rate of $f$ using Equations (76) and (77)

$$
\begin{aligned}
\Omega_{\mathcal{D}_k}(f) &= \frac{\min_{\ell \in [p]} \mathbb{E}_{\mathcal{D}_k}\left[f(X, Z) \mid Z = \ell\right]}{\max_{\ell \in [p]} \mathbb{E}_{\mathcal{D}_k}\left[f(X, Z) \mid Z = \ell\right]} \\
&\overset{(77),(76)}{\leq} \frac{1 + \frac{c}{1-2\lambda}}{1 + \frac{c}{2\lambda}} \\
&\leq \left(1 + \frac{c}{1 - 2\lambda}\right) \cdot \left(1 - \frac{c}{2\lambda} + \frac{c^2}{4\lambda^2}\right) \\
&\qquad\qquad\qquad \text{(Using that for all } x \geq 0, \; \tfrac{1}{1+x} \leq 1 - x + x^2) \\
&= 1 + \frac{c}{1 - 2\lambda} - \frac{c}{2\lambda} + \frac{c^2}{4\lambda^2} \cdot \left(1 + \frac{c}{1 - 2\lambda}\right) - \frac{c^2}{2\lambda(1 - 2\lambda)} \\
&\leq 1 - \frac{c(1 - 4\lambda)}{2\lambda(1 - 2\lambda)} + \frac{3c^2}{4\lambda^2} \qquad\qquad \text{(Using that } 0 < \lambda \leq \tfrac{1}{4} \text{ and } c > 0) \\
&\leq 1 - \frac{c}{2\lambda} \cdot (1 - 4\lambda) + \frac{3c^2}{4\lambda^2}. \qquad\qquad \text{(Using that } \lambda \geq 0)
\end{aligned}
$$

Thus, in Case C, $f$ has a statistical rate at most $1 - \frac{c}{2\lambda} \cdot (1 - 4\lambda) + \frac{3c^2}{4\lambda^2}$ on the chosen distribution $\mathcal{D}_k$.

**(Case D)** $\sum_{r \in \{x_A, x_B, x_C\}} f(r, 2) \leq 1$**:** In Case D, $f$ outputs of 1 on at most one point in the tuple

$$L = ((x_A, 2), (x_B, 2), (x_C, 2)).$$

If $f$ takes a value of 1 on a point, say $L_j$, in $L$, then fix $k \in [3]$, such that $\mathcal{D}_k(L_j) = c/2$. (Such a distribution always exists by construction). Otherwise, fix any $k \in [3]$. For $Z = 2$, we have

$$\mathbb{E}_{\mathcal{D}_k} \left[ f(X, Z) \mid Z = 2 \right] = \sum_{r \in \mathcal{X}} f(r, 2) \cdot \Pr_{\mathcal{D}_k} \left[ X = r \mid Z = 2 \right] \leq \frac{\lambda - c}{2\lambda}.$$

$$\text{(By our choice of } k) \quad (78)$$

For $Z = 1$, we have

$$\mathbb{E}_{\mathcal{D}_k} \left[ f(X, Z) \mid Z = 1 \right]$$
$$= \sum_{r \in \mathcal{X}} f(r, 1) \cdot \Pr_{\mathcal{D}_k} \left[ X = r \mid Z = 1 \right]$$
$$= \Pr_{\mathcal{D}_k} \left[ X = x_D \mid Z = 1 \right] + \sum_{r \in \{x_A, x_B, x_C\}} f(r, 1) \cdot \Pr_{\mathcal{D}_k} \left[ X = r \mid Z = 1 \right]$$
$$\text{(From previous cases, } f(x_D, 1) = 1 \text{ and } f(x_E, 1) = 0)$$
$$\overset{\text{Table 3}}{\geq} \frac{(1 - 2\lambda - c)}{2(1 - 2\lambda)}. \quad (79)$$

We can compute the statistical rate of $f$ using Equations (78) and (79)

$$
\begin{aligned}
\Omega_{\mathcal{D}_k}(f) &= \frac{\min_{\ell \in [p]} \mathbb{E}_{\mathcal{D}_k} \left[ f(X, Z) \mid Z = \ell \right]}{\max_{\ell \in [p]} \mathbb{E}_{\mathcal{D}_k} \left[ f(X, Z) \mid Z = \ell \right]} \\
&\overset{(79),(78)}{\leq} \frac{1 - \frac{c}{2\lambda}}{1 - \frac{c}{1 - 2\lambda}} \\
&\leq \left( 1 - \frac{c}{2\lambda} \right) \cdot \left( 1 + \frac{c}{1 - 2\lambda} + \frac{2c^2}{(1 - 2\lambda)^2} \right) \\
&\qquad\qquad \text{(Using that for all } x \leq \tfrac{1}{2}, \frac{1}{1-x} \leq 1 + x + 2x^2) \\
&= 1 + \frac{c}{1 - 2\lambda} - \frac{c}{2\lambda} + \frac{2c^2}{(1 - 2\lambda)^2} \cdot \left( 1 - \frac{c}{2\lambda} \right) - \frac{c^3}{2\lambda(1 - 2\lambda)^2} \\
&\leq 1 - \frac{c(1 - 4\lambda)}{2\lambda(1 - 2\lambda)} + 8c^2 \qquad\qquad \text{(Using that } 0 < \lambda \leq \tfrac{1}{4} \text{ and } c > 0) \\
&\leq 1 - \frac{c}{2\lambda} \cdot (1 - 4\lambda) + \frac{3c^2}{4\lambda^2}. \qquad\qquad \text{(Using that } 0 \leq \lambda \leq \tfrac{1}{4})
\end{aligned}
$$

Thus, in this case, $f$ has a statistical rate at most $1 - \frac{c}{2\lambda} \cdot (1 - 4\lambda) + \frac{3c^2}{4\lambda^2}$ on the chosen distribution $\mathcal{D}_k$. $\qquad\square$

### A.3.3  Proof of Lemma A.20

*Proof of Lemma A.20.* For each distribution $\mathcal{D}_1$, $\mathcal{D}_2$, and $\mathcal{D}_3$, we will give classifiers $f_1, f_2, f_3 \in \mathcal{F}$, such that, for each $k \in [3]$, $f_k$ has error at most $3\alpha/2$ and a statistical rate 1 with respect to $\mathcal{D}_k$. The idea is to choose a classifier $f_k$ such that, conditioned on a value of $Z$, they label exactly $1/2$-fraction of the samples as positive on $\mathcal{D}_k$. Thus, they have a statistical rate of 1. Then, by the construction it follows that these classifiers also have a small error. We give the classifier $f_1$ for distribution $\mathcal{D}_1$. Classifiers $f_2$ and $f_3$ follow by symmetry.

Define $f_1 \in \mathcal{F}$ to be the following classifier

$$f_1(x, z) := \begin{cases} 1 & \text{if } x \in \{x_B, x_D\}, \\ 1 & \text{if } x = x_C \text{ and } z = 2, \\ 0 & \text{otherwise.} \end{cases}$$

Using Equation (71) and Table 3, we get that

$$\text{Err}_{\mathcal{D}_1}(f_1) = \frac{3c}{2}.$$

For $Z = 2$, we have

$$
\begin{aligned}
\mathbb{E}_{\mathcal{D}_1}\left[f_1(X, Z) \mid Z = 2\right] &= \sum_{r \in \mathcal{X}} f_1(r, 2) \cdot \text{Pr}_{\mathcal{D}_1}\left[X = r \mid Z = 2\right] \\
&= \text{Pr}_{\mathcal{D}_1}\left[X \in \{x_B, x_C, x_D\} \mid Z = 2\right] \\
&\overset{\text{Table 3}}{=} \frac{1}{2}.
\end{aligned}
\tag{80}
$$

For $Z = 1$, we have

$$
\begin{aligned}
\mathbb{E}_{\mathcal{D}_1}\left[f(X, Z) \mid Z = 1\right] &= \sum_{r \in \mathcal{X}} f_1(r, 1) \cdot \text{Pr}_{\mathcal{D}_1}\left[X = r \mid Z = 1\right] \\
&= \text{Pr}_{\mathcal{D}_1}\left[X \in \{x_B, x_D\} \mid Z = 1\right] \\
&\overset{\text{Table 3}}{=} \frac{1}{2}.
\end{aligned}
\tag{81}
$$

Thus, we have

$$\Omega_{\mathcal{D}_1}(h) = \frac{\min_{\ell \in [p]} \mathbb{E}_{\mathcal{D}_1}\left[f(X, Z) \mid Z = \ell\right]}{\max_{\ell \in [p]} \mathbb{E}_{\mathcal{D}_1}\left[f(X, Z) \mid Z = \ell\right]} \overset{(80),(81)}{=} 1.$$

$\square$

### A.4 Impossibility result omitted from Section 4

**Theorem A.21** (**Fairness guarantee of Theorem 4.3 is optimal up to constant factors**). *For all perturbation rates $\eta \in (0, 1/2]$, constants $\lambda \in (0, 1/4)$, and confidence parameters $\delta \in [0, 1/2)$, if the fairness metric is statistical rate and $\tau = 1$, then even with the knowledge that Assumption 1 holds with a (known) constant $\lambda$, for any $\varepsilon < \eta$ and $v < \tau$ it is impossible to $(\varepsilon, \nu)$-learn any hypothesis class $\mathcal{F} \subseteq \{0, 1\}^{\mathcal{X} \times [p]}$ that shatters a set of 6 points of the form $\{x_A, x_B, x_C\} \times [2] \subseteq \mathcal{X} \times [p]$ for some distinct $x_A, x_B, x_C \in \mathcal{X}$.*

Thus, Theorem A.21 shows that for any $\varepsilon < \eta$, no algorithm can $(\varepsilon, \nu)$-learn any hypothesis classes $\mathcal{F}$ satisfying mild assumptions; this shows that the accuracy guarantee in Theorem 4.3 is optimal up to a constant factor.

#### A.4.1 Proof of Theorem A.21

Theorem A.21's assumes that $\mathcal{F}$ shatters a set $\{x_A, x_B, x_C\} \times [2] \subseteq \mathcal{X} \times [p]$, Assumption 1 holds with a constant $\lambda \in (0, 1/4)$, $\tau = 1$, and the statistical rate is the fairness metric; recall that the statistical rate of $f \in \mathcal{F}$ with respect to distribution $\mathcal{D}$ is

$$\Omega_{\mathcal{D}}(f) := \frac{\min_{\ell \in [p]} \text{Pr}_{\mathcal{D}}[f = 1 \mid Z = \ell]}{\max_{\ell \in [p]} \text{Pr}_{\mathcal{D}}[f = 1 \mid Z = \ell]}.$$

Then, given parameters $\eta \in (0, 1/2]$, $\lambda \in (0, 1/4)$, and $\delta \in [0, 1/2)$ our goal is to prove that for any

$$0 < \varepsilon < \eta \text{ and } \nu \in [0, 1],$$

$\mathcal{F}$ is not $(\varepsilon, \nu)$-learnable with perturbation rate $\eta$ and confidence $\delta$.

Our proof is inspired by the approach of [39, Theorem 1] and [11, Theorem 1]. It has a similar structure to the proof of Theorem 4.4; but uses a different construction in which label $Y$ depends on $Z$ conditioned on $X$. (For a reader who has read the proof of Theorem 4.4, this means that condition (C3) in Proposition A.15 does not hold; and we have to give a different algorithm for the adversary to "hide" the true distribution.)

*Proof of Theorem A.21.* Let $\mathcal{L}$ be any learner. We construct two distributions $\mathcal{P}$ and $\mathcal{Q}$ (parameterized by $\eta$) such that Lemma A.22 holds (see Figures 2 and 3 for a description of $\mathcal{P}$ and $\mathcal{Q}$).

|   | $x_A$ | $x_B$ | $x_C$ |
|---|-------|-------|-------|
| 1 | $\eta/2$ | $1/2 - \eta$ | $\eta/2$ |
| 2 | $\eta/2$ | $1/2 - \eta$ | $\eta/2$ |

Figure 2: Marginal distributions of $\mathcal{P}$ and $\mathcal{Q}$ over $\mathcal{X} \times [2]$. For $(r, s) \in \{x_A, x_B, x_C\} \times [2]$, the corresponding cell denotes $\Pr_\mathcal{P}[(X, Z) = (r, s)] = \Pr_\mathcal{Q}[(X, Z) = (r, s)]$.

**Lemma A.22 (Adversary can hide the true distribution).** *For both $\mathcal{D} \in \{\mathcal{P}, \mathcal{Q}\}$, given $N \in \mathbb{N}$ iid samples $S$ from $\mathcal{D}$, if $\eta \cdot N$ is integral, then there is a distribution $\mathcal{D}_{\mathrm{Mix}}$ over $\mathcal{X} \times \{0, 1\} \times [p]$ and an adversary $A \in \mathcal{A}(\eta)$ such that with probability at least $1/2$ (over the draw of $S$) samples in $\widehat{S} := A(S)$ are distributed as iid draws from $\mathcal{D}_{\mathrm{Mix}}$.*

Suppose $\mathcal{L}$ is given $N \in \mathbb{N}$ samples $S$; where $N \cdot \eta$ is integral. Consider two cases: In the first case, $S$ is iid from $\mathcal{P}$ and in the second case, $S$ is iid from $\mathcal{Q}$. By Lemma A.22, in each case there is an $A \in \mathcal{A}(\eta)$ such that, with probability at least $1/2$, the perturbed samples $\widehat{S} := A(S)$ are have the distribution $\mathcal{D}_{\mathrm{Mix}}$.

Thus, given $\widehat{S}$, with probability at least $1/2$, the learner $\mathcal{L}$ cannot identify the distribution from which $S$ was drawn. As a result, with probability at least $1/2$, $\mathcal{L}$ must to output a common classifier, say $f_{\mathrm{Com}} \in \mathcal{F}$, which satisfies the accuracy and fairness guarantee for both $\mathcal{P}$ and $\mathcal{Q}$. We show that no $f_{\mathrm{Com}} \in \mathcal{F}$ satisfies the accuracy and fairness guarantee for both $\mathcal{P}$ and $\mathcal{Q}$.

**Lemma A.23 (No good classifier for both $\mathcal{P}$ and $\mathcal{Q}$).** *There is no $f \in \mathcal{F}$, such that*

$$\mathrm{Err}_\mathcal{P}[f] < \eta \text{ and } \mathrm{Err}_\mathcal{Q}[f] < \eta.$$

**Lemma A.24 (A good classifier for each $\mathcal{P}$ and $\mathcal{Q}$).** *For each $\mathcal{D} \in \{\mathcal{P}, \mathcal{Q}\}$, there is $f_\mathcal{D}^\star \in \mathcal{F}$ such that*

$$\mathrm{Err}_\mathcal{D}[f_\mathcal{D}^\star] = 0 \text{ and } \Omega_\mathcal{D}(f_\mathcal{D}) = 1.$$

Note that $f_\mathcal{P}^\star$ (respectively $f_\mathcal{Q}^\star$) has perfect accuracy and satisfies the fairness constraints with respect to $\mathcal{P}$ (respectively $\mathcal{Q}$). If $\mathcal{L}$ is a $(\varepsilon, \nu)$-learner, then the first case, with probability at least $1 - \delta > 1/2$, $\mathcal{L}$ must output a classifier $f_1$ which satisfies

$$\mathrm{Err}_\mathcal{P}(f_1) - \mathrm{Err}_\mathcal{P}(f_\mathcal{P}^\star) \le \varepsilon \text{ and } \tau - \Omega_\mathcal{P}(f_1) \le \nu.$$

And in the second case, with probability at least $1 - \delta$, $\mathcal{L}$ must output a classifier $f_2$ which satisfies

$$\mathrm{Err}_\mathcal{Q}(f_2) - \mathrm{Err}_\mathcal{Q}(f_\mathcal{Q}^\star) \le \varepsilon \text{ and } \tau - \Omega_\mathcal{Q}(f_2) \le \nu.$$

However, in both cases, with probability at least $1/2$, $\mathcal{L}$ outputs $f_{\mathrm{Com}}$. Because $1 - \delta > 1/2$, $f_{\mathrm{Com}}$ must satisfy

$$\mathrm{Err}_\mathcal{P}(f_{\mathrm{Com}}) - \mathrm{Err}_\mathcal{P}(f_\mathcal{P}^\star) \le \varepsilon \text{ and } \mathrm{Err}_\mathcal{Q}(f_{\mathrm{Com}}) - \mathrm{Err}_\mathcal{Q}(f_\mathcal{Q}^\star) \le \varepsilon \tag{82}$$

$$\tau - \Omega_\mathcal{P}(f_{\mathrm{Com}}) \le \nu \text{ and } \tau - \Omega_\mathcal{Q}(f_{\mathrm{Com}}) \le \nu \tag{83}$$

But since $\varepsilon < \eta$, Equation (82) contradicts Lemma A.22. Thus, $\mathcal{L}$ is not an $(\varepsilon, \nu)$-learner for $\mathcal{F}$. Since the choice of $\mathcal{L}$ was arbitrary, we have shown that there is no learner which $(\varepsilon, \nu)$-learns $\mathcal{F}$.

It remains to prove the Lemma A.22, Lemma A.23, and Lemma A.24. $\qquad\square$

### A.4.2 Proof of Lemma A.22

Set $\mathcal{P}$ and $\mathcal{Q}$ to be the unique distributions satisfying the following properties:

(P1) $\mathcal{P}$ and $\mathcal{Q}$ have the same marginal distribution on $\mathcal{X} \times [p]$; their marginal distribution on $\mathcal{X} \times [p]$ is given in Figure 2.

(P2) for a sample $(X, Y, Z) \sim \mathcal{P}$, Figure 3(a) denotes the distribution of $Y$ conditioned on $X$ and $Z$, and for a sample $(X, Y, Z) \sim \mathcal{Q}$, Figure 3(b) denotes the distribution of $Y$ conditioned on $X$ and $Z$.

|     | $x_A$ | $x_B$ | $x_C$ |
| --- | --- | --- | --- |
| 1   | 1   | 1   | 0   |
| 2   | 0   | 1   | 1   |

(a) *Distribution* $\mathcal{P}$: For $(r, s) \in \{x_A, x_B, x_C\} \times [2]$, the corresponding cell denotes the value of $Y$ conditioned on $X$ and $Z$, when sample $(X, Y, Z) \sim \mathcal{P}$.

|     | $x_A$ | $x_B$ | $x_C$ |
| --- | --- | --- | --- |
| 1   | 0   | 1   | 1   |
| 2   | 1   | 1   | 0   |

(b) *Distribution* $\mathcal{Q}$: For $(r, s) \in \{x_A, x_B, x_C\} \times [2]$, the corresponding cell denotes the value of $Y$ conditioned on $X$ and $Z$, when sample $(X, Y, Z) \sim \mathcal{Q}$.

Figure 3: Figure 3(a) denotes the conditional distribution of $Y$ when $(X, Y, Z) \sim \mathcal{P}$. Figure 3(b) denotes the conditional distribution of $Y$ when $(X, Y, Z) \sim \mathcal{Q}$.

From Figures 2 and 3, it follows that, conditional on $X = x_B$ the samples $(X, Y, Z) \sim \mathcal{P}$ and $(X, Y, Z) \sim \mathcal{Q}$ follow the same distribution. Then, the goal of the adversary $A$ (in Lemma A.22) is to specifically perturb the samples with $X \neq x_B$ so that the perturbed sample $(X, Y, \widehat{Z})$ follows the same distribution irrespective of whether the original sample $(X, Y, Z)$ is drawn from $\mathcal{P}$ or $\mathcal{Q}$.

*Proof of Lemma A.22.* Let $S := \{(X_i, Z_i, Y_i)\}_{i \in [N]}$. Let $A \in \mathcal{A}(\eta)$ execute the following algorithm:

---

1. **For** $i \in [N]$ **do:**

   (a) **Sample** $t_i$ uniformly at random from $[0, 1]$
   (b) **If** $X_i = x_B$ **or** $t_i \leq 1/2$ **then: Set** $\widetilde{Z}_i := Z_i$,
   (c) **Otherwise: Set** $\widetilde{Z}_i := 3 - Z_i$ \qquad //If $Z_i = 1$ set $\widetilde{Z}_i = 2$, if $Z_i = 2$ set $\widetilde{Z}_i = 1$

2. **If** $\sum_{i \in [N]} \mathbb{I}[\widetilde{Z}_i \neq Z_i] \leq \eta \cdot N$ **then: return** $\{(X_i, \widetilde{Z}_i, Y_i)\}_{i \in [N]}$,

3. **Otherwise: return** $\{(X_i, Z_i, Y_i)\}_{i \in [N]}$

---

Intuitively, $A$ perturbs the samples with $X \neq x_B$, such that whether $S$ is drawn from $\mathcal{P}$ or $\mathcal{Q}$, the following invariant holds: For any $s \in [2]$,
$$\Pr[Y = 0 \mid X \neq x_B, \widetilde{Z} = s] = \Pr[Y = 1 \mid X \neq x_B, \widetilde{Z} = s] = 1/2.$$
We will show that combining this with the facts that $\mathcal{P}$ and $\mathcal{Q}$ have the same marginal distribution on $\mathcal{X} \times [p]$, and that conditioned on $X = x_B$ samples from $\mathcal{P}$ and $\mathcal{Q}$ have the same distribution suffices to prove Lemma A.22.

The check in Step 2 ensures that $A$ never perturbs more than $\eta \cdot N$ samples; thus, $A$ is admissible in the $\eta$-Hamming model. Lemma A.25 shows that the check in Step 2 is true with probability at least $1/2$.

**Lemma A.25.** *If $\eta \cdot N$ is integral, then with probability at least $1/2$, $\sum_{i \in [N]} \mathbb{I}[\widetilde{Z}_i \neq Z_i] < \eta \cdot N$.*

*Proof.* For all $i \in [N]$, let $C_i \in \{0, 1\}$ be a random variable indicating if $\widetilde{Z}_i \neq Z_i$. It suffices to show that with probability at least $1/2$, $\sum_{i \in [N]} C_i \leq \eta \cdot N$. Towards this, we first compute $\Pr[C_i = 1]$ as follows

$$\begin{aligned}
\Pr[C_i = 1] &= \Pr[Z_i \neq \widetilde{Z}_i] & \text{(Definition of } C_i) \\
&= \Pr[X_i \neq x_B, t_i > 1/2] & \text{(Using Step 1(b) and Step 1(c))} \\
&= \Pr[X_i \neq x_B] \cdot \Pr[t_i > 1/2] & ((X_i, Y_i, Z_i) \text{ and } t_i \text{ are independent}) \\
&= 2\eta \cdot \frac{1}{2}. & \text{(Using } t_i \text{ is uniformly at random from } [0, 1] \text{ and Figure 2)}
\end{aligned}$$

Since for each $i$, the sample $(X_i, Y_i, Z_i)$ and the point $t_i$ is drawn independently of each other, other samples, and other points, it follows that the random variables $\{C_i\}_i$ are independent. Hence, $\sum_{i \in [N]} C_i$ follows a binomial distribution with mean $\eta \cdot N$. Since $\eta \cdot N$ is integral, we have that the median of the distribution of $\sum_{i \in [N]} C_i$ is $\eta \cdot N$. Thus, we get that $\Pr[\sum_{i \in [N]} C_i \leq \eta \cdot N] = 1/2$. $\qquad\square$

Thus, Lemma A.25 shows that with probability at least $1/2$, $\widehat{S} := \{(X_i, \widetilde{Z}_i, Y_i)\}_{i \in [N]}$.

**Lemma A.26.** *There is a distribution $\mathcal{D}_{\mathrm{Mix}}$, such that, for all $\mathcal{D} \in \{\mathcal{P}, \mathcal{Q}\}$, given $S$ iid from $\mathcal{D}$, the samples $\widetilde{S} := \{(X_i, \widetilde{Z}_i, Y_i)\}_{i \in [N]}$ (computed in the algorithm of A) are independent of each other and is distributed according $\mathcal{D}_{\mathrm{Mix}}$.*

*Proof.* Since for each $i \in [N]$ the sample $(X_i, Y_i, Z_i)$ and the point $t_i$ is drawn independently of others, it follows that the samples $(X_i, \widetilde{Z}_i, Y_i)$ are independent of each other.

Next, we show that $\widetilde{S}$ follows the same distribution whether $S$ is iid from $\mathcal{P}$ or from $\mathcal{Q}$. The distribution $\mathcal{D}_{\mathrm{Mix}}$ will be implicit in the proof. Since samples in $S$ are iid and the algorithm of $A$ does not depend on $i$, it follows that all samples in $\widetilde{S}$ follow the same distribution. Thus, it suffices to consider any one sample in $\widetilde{S}$. Fix any $i \in [N]$. Consider the $i$-th sample $(X_i, \widetilde{Z}_i, Y_i)$. For the rest of the proof we drop the subscript $i$ in $(X_i, \widetilde{Z}_i, Y_i)$, $(X_i, Z_i, Y_i)$, and $t_i$.

Fix any $y \in \{0, 1\}$ and $s \in [2]$. Then, it holds that

$$
\begin{aligned}
\Pr[X = x_A, Y = y, \widetilde{Z} = s] = &\quad \Pr[X = x_A, Y = y, \widetilde{Z} = s \mid t \leq 1/2] \cdot \Pr[t \leq 1/2] \\
&+ \Pr[X = x_A, Y = y, \widetilde{Z} = s \mid t > 1/2] \cdot \Pr[t > 1/2] \\
= &\quad \Pr[X = x_A, Y = y, Z = s \mid t \leq 1/2] \cdot \Pr[t \leq 1/2] \\
&+ \Pr[X = x_A, Y = y, Z = 3 - s \mid t > 1/2] \cdot \Pr[t > 1/2] \\
= &\quad \Pr[X = x_A, Y = y, Z = s] \cdot 1/2 \\
&+ \Pr[X = x_A, Y = y, Z = 3 - s] \cdot 1/2 \\
= &\quad \Pr[X = x_A, Y = y] \cdot 1/2. \qquad \text{(Using that } Z \in \{0, 1\})
\end{aligned}
$$

Now, from Figure 2 it follows that $\Pr[X = x_A, Y = y] = \eta/2$ both when $(X, Y, Z) \sim \mathcal{P}$ or when $(X, Y, Z) \sim \mathcal{Q}$. Thus, we get that

$$
\Pr[X = x_A, Y = y, \widetilde{Z} = s] = \frac{\eta}{4}.
$$

Replacing $x_A$ by $x_C$ in the above argument, we get that

$$
\Pr[X = x_C, Y = y, \widetilde{Z} = s] = \frac{\eta}{4}.
$$

Finally, since $A$ does not perturb samples with $X = x_B$ (see Step 1(b)), from Figures 2 and 3 it follows that

$$
\begin{aligned}
\Pr[X = x_B, Y = y, \widetilde{Z} = s] &= \Pr[X = x_B, Y = y, Z = s] \\
&= \mathbb{I}[y = 1] \cdot \left(\frac{1}{2} - \eta\right).
\end{aligned}
$$

Since the choice of $i \in [N]$, $y \in \{0, 1\}$, and $s \in [2]$ was arbitrary. We get that all samples in $\widetilde{S}$ follow the same distribution whether $S$ is iid from $\mathcal{P}$ or from $\mathcal{Q}$. $\qquad\square$

Lemma A.22 follows because with probability at least $1/2$, $\widehat{S}$ is $\{(X_i, \widetilde{Z}_i, Y_i)\}_{i \in [N]}$ (by Lemma A.25) and the samples $\{(X_i, \widetilde{Z}_i, Y_i)\}_{i \in [N]}$ are independent and distributed according to $\mathcal{D}_{\mathrm{Mix}}$ (by Lemma A.17). $\qquad\square$

### A.4.3 Proof of Lemma A.23

*Proof of Lemma A.23.* Fix any $f \in \mathcal{F}$.

$$\mathrm{Err}_{\mathcal{P}}(f) + \mathrm{Err}_{\mathcal{Q}}(f)$$

$$= \Pr_{(X,Y,Z)\sim\mathcal{P}}[f(X,Z) \neq Y] + \Pr_{(X,Y,Z)\sim\mathcal{Q}}[f(X,Z) \neq Y] \quad \text{(Using the definition of Err)}$$

$$= \sum_{r\in\mathcal{X},s\in[2]} \Pr_{(X,Y,Z)\sim\mathcal{P}}[X=r, Z=s] \cdot \Pr_{(X,Y,Z)\sim\mathcal{P}}[f(X,Z) \neq Y \mid X=r, Z=s]$$

$$+ \Pr_{(X,Y,Z)\sim\mathcal{Q}}[X=r, Z=s] \cdot \Pr_{(X,Y,Z)\sim\mathcal{Q}}[f(X,Z) \neq Y \mid X=r, Z=s]$$

$$\geq \sum_{r\in\{x_A,x_C\},s\in[2]} \frac{\eta}{2} \cdot \Pr_{(X,Y,Z)\sim\mathcal{P}}[f(X,Z) \neq Y \mid X=r, Z=s]$$

$$+ \frac{\eta}{2} \cdot \Pr_{(X,Y,Z)\sim\mathcal{Q}}[f(X,Z) \neq Y \mid X=r, Z=s]$$

$$= \sum_{r\in\{x_A,x_C\},s\in[2]} \frac{\eta}{2} \cdot \Pr_{(X,Y,Z)\sim\mathcal{P}}[f(X,Z) \neq 0 \mid X=r, Z=s]$$

$$+ \frac{\eta}{2} \cdot \Pr_{(X,Y,Z)\sim\mathcal{Q}}[f(X,Z) \neq 1 \mid X=r, Z=s]$$

$$\text{(Using Property (P2) of } \mathcal{P} \text{ and } \mathcal{Q}; \text{ see Figure 3)}$$

$$= \sum_{r\in\{x_A,x_C\},s\in[2]} \frac{\eta}{2} \cdot \Pr_{(X,Y,Z)\sim\mathcal{P}}[f(X,Z) \neq 0 \mid X=r, Z=s]$$

$$+ \frac{\eta}{2} \cdot \Pr_{(X,Y,Z)\sim\mathcal{P}}[f(X,Z) \neq 1 \mid X=r, Z=s]$$

$$\text{(Using Property (P1) of } \mathcal{P} \text{ and } \mathcal{Q}; \text{ see Figure 2)}$$

$$= \sum_{r\in\{x_A,x_C\},s\in[2]} \frac{\eta}{2}$$

$$= 2\eta.$$

Since $\mathrm{Err}_{\mathcal{P}}(f), \mathrm{Err}_{\mathcal{Q}}(f) \geq 0$, by the Pigeonhole principle either $\mathrm{Err}_{\mathcal{P}}(f) \geq \eta$ or $\mathrm{Err}_{\mathcal{Q}}(f) \geq \eta$. $\square$

### A.4.4 Proof of Lemma A.24

*Proof of Lemma A.24.* Consider a sample $(X,Y,Z) \sim \mathcal{P}$ Notice that, for some $r \in \mathcal{X}$ and $s \in [2]$, conditioned on $X = r$ and $Z = s$, the label $Y$ is uniquely identified by Figure 3(a). Let $f_{\mathcal{P}}^\star$ be the classifier than given $(r,s)$ predicts the value in the corresponding cell of Figure 3(a). Clearly, $\mathrm{Err}_{\mathcal{P}}(f_{\mathcal{P}}^\star) = 0$. Further, by our assumption that $\mathcal{F}$ shatters the set $\{x_A, x_B, x_C\} \times [2] \subseteq \mathcal{X} \times [p]$, it follows that $f_{\mathcal{P}}^\star \in \mathcal{F}$.

The construction of $f_{\mathcal{Q}}^\star$ follows symmetrically by using Figure 3(b). $\square$

### A.5 Additional remarks about the $\eta$-Hamming model and theoretical results

In Example A.27, we show that the ratio of the fairness of a classifier $f \in \mathcal{F}$ with respect to the perturbed samples $\widehat{S}$ and with respect to the unperturbed samples $S$ can be 0.

**Example A.27 (Ratio of fairness of a classifier on perturbed and unperturbed samples).** *Let $S := \{(x_i, y_i, z_i)\}_{i\in[N]}$ denote the $N$ unperturbed samples. Suppose that the fairness metric $\Omega$ is statistical rate and $S$ has an equal number of samples from each protected group (i.e., for all $\ell \in [p]$, $\sum_{i\in[N]} \mathbb{I}[z_i = \ell] = N/p$.) Consider a classifier $f \in \mathcal{F}$ that has exactly $\eta \cdot N$ positive predictions on each protected group $\ell \in [p]$, i.e., for all $\ell \in [p]$, $|\{i \in [N] \mid f(x_i, z_i) = 1 \text{ and } z_i = \ell\}| = \eta \cdot N$. This implies that $\Omega(f, S) = 1$. Fix any protected group $\ell \in [p]$. An adversary $A \in \mathcal{A}(\eta)$, can perturb the protected attributes of all $\eta \cdot N$ samples in the set $\{i \in [N] \mid f(x_i, z_i) = 1 \text{ and } z_i = \ell\}$. In this case, $\Pr_{\widehat{S}}[f = 1 \mid Z = \ell] = 0$. This implies that $\Omega(f, \widehat{S}) = 0$. Thus, in this example, $\frac{\Omega(f,\widehat{S})}{\Omega(f,S)} = 0$.*

In Remark A.28 we give two example hypothesis classes that satisfy the assumptions in Theorems 4.4 and 4.5. Theorem 4.5 assumes that there exist five distinct points $x_A, x_B, x_C, x_D, x_E \in \mathcal{X}$ such that $\mathcal{F}$ shatters the set of points $P := \{x_A, x_B, x_C, x_D, x_E\} \times [2] \subseteq \mathcal{X}$. Theorem 4.4 makes the weaker assumption that $\mathcal{F}$ shatters the set $\{x_A, x_B, x_C\} \times [2] \subseteq \mathcal{X}$.

**Remark A.28.**

1. **Decision trees.** *Suppose $\mathcal{X} := \mathbb{R}$. Then, the following hypothesis class of two-layer decision trees shatters $P$: On the first layer, the decision tree splits the root node into five nodes by thresholding $X \in \mathcal{X}$. On the second layer, it further splits each node in the first layer into two leaves depending on whether $Z = 1$ or $Z = 2$. The resulting tree has 10 leaves. This hypothesis class shatters $P$: One can choose the thresholds in the first layer so that each of $x_A$, $x_B$, $x_C$, $x_D$, and $x_E$ belong to different nodes on the first layer. Then, the $2^{10}$ hypothesis generated by assigning different outcomes to the leaves shatter $P$.*

2. **SVM with kernels.** *Suppose $\mathcal{X} := \mathbb{R}^5$, and $x_A = e_1$, $x_B = e_2$, $x_C = e_3$, $x_D = e_4$, $x_E = e_5$, where $e_i$ is the $i$-th standard basis in $\mathbb{R}^5$. Consider the hypothesis class of SVM classifiers on the feature space $\mathcal{X}^2$; where given a sample $(x, z) \in \mathcal{X} \times [p]$, we map it to $\mathcal{X}^2$ using the map $\psi \colon \mathcal{X} \times [p] \to \mathcal{X}^2$ defined as follows: $\psi(x, z) := (x, z \cdot x) \in \mathcal{X}^2$. One can verify that the hypothesis class of SVM classifiers in this feature space shatter the set $P$ (more precisely, it shatters the image of $P$ under map $\psi$).*

**Remark A.29 (Test errors).** *In the paper, we focus on the setting where given perturbed samples as input, the learner's goal is to find a classifier that satisfies a given set of fairness constraints with the highest accuracy, where both accuracy and fairness are measured with respect to* true distribution *$\mathcal{D}$. In some applications, for example, when protected attributes are self-reported after deployment, the test samples can also have perturbations. Given a number $\Delta > 0$, a sufficient number of test samples $T$, and an $\eta$-Hamming adversary $A$, one can show that $\mathrm{Err}_{\widehat{D}}(f_{\mathrm{ET}}) - \mathrm{Err}_{\mathcal{D}}(f^\star) \leq 3\eta + \Delta$ and $\Omega(f_{\mathrm{ET}}, A(T)) \geq \tau - {}^{12\eta\tau}/(\lambda-2\eta) - \Delta$, where $A(T)$ denotes the perturbed test samples and $\widehat{D}$ is the empirical distribution of $A(T)$.*

# B Extensions of theoretical results

## B.1 Theoretical results with multiple protected attributes and fairness metrics

In this section, we extend Program (ErrTolerant) to the general case with $m \in \mathbb{N}$ fairness constraints $\Omega^{(1)}, \Omega^{(2)}, \ldots, \Omega^{(k)}$ with respect to $m$ (not necessarily distinct) protected attributes $Z^{(1)}, Z^{(2)}, \ldots, Z^{(m)}$.

**Definition B.1 (General Error-tolerant program).** *Given a perturbation rate $\eta \in [0, 1]$, constants $\lambda, \Delta \in (0, 1]$, and for each $r \in [m]$, given a fairness constraint $\Omega^{(r)}$, corresponding events $\mathcal{E}^{(r)}$ and $(\mathcal{E}')^{(r)}$ (as in Definition 3.1), and protected attribute $Z^{(r)} \in [p_r]$, we define the general error-tolerant program for perturbed samples $S$, with empirical distribution is $\widehat{D}$, as*

$$\min_{f \in \mathcal{F}} \quad \mathrm{Err}_{\widehat{D}}(f), \tag{General-ErrTolerant} \quad (84)$$

$$\text{s.t.,} \quad \forall\, r \in [m], \qquad \Omega^{(r)}(f, \widehat{S}) \geq \tau \cdot \left( \frac{1 - (\eta + \Delta)/\lambda}{1 + (\eta + \Delta)/\lambda} \right)^2, \tag{85}$$

$$\forall\, r \in [m] \text{ and } \ell \in [p_r], \quad \mathrm{Pr}_{\widehat{D}}\left[ \mathcal{E}^{(r)}(f), (\mathcal{E}')^{(r)}(f), \widehat{Z} = \ell \right] \geq \lambda - \eta - \Delta. \tag{86}$$

Let $f^\star \in \mathcal{F}$ have the lowest error subject to satisfying all fairness constraints with respect to $\mathcal{D}$:

$$f^\star := \mathrm{argmin}_{f \in \mathcal{F}} \mathrm{Err}_{\mathcal{D}}(f) \quad \text{s.t.,} \quad \text{for all } r \in [m], \quad \Omega_{\mathcal{D}}^{(r)}(f) \geq \tau.$$

We need the following generalization of Assumption 1:

**Assumption 2.** *There is a known constant $\lambda > 0$ such that*

$$\min_{r \in [m]} \min_{\ell \in [p_r]} \mathrm{Pr}_{\mathcal{D}}[\mathcal{E}^{(r)}(f^\star), (\mathcal{E}')^{(r)}(f^\star), Z = \ell] \geq \lambda.$$

**Theorem B.2 (Extending Theorem 4.3 to multiple protected attributes and fairness constraints).** *Suppose Assumption 2 holds with constant $\lambda > 0$ and $\mathcal{F}$ has VC dimension $d \in \mathbb{N}$. Then, for all perturbation rates $\eta \in (0, \lambda/2)$, fairness thresholds $\tau \in (0, 1]$, bounds on error $\varepsilon > 2\eta$ and constraint violation $\nu > {}^{8\eta\tau}/(\lambda-2\eta)$, and confidence parameters $\delta \in (0, 1)$ with probability at least $1 - \delta$, the optimal solution $f_{\mathrm{ET}} \in \mathcal{F}$ of Program (General-ErrTolerant) with parameters $\eta$, $\lambda$, and $\Delta := O\left(\varepsilon - 2\eta, \nu - {}^{8\eta\tau}/(\lambda-2\eta), \lambda - 2\eta\right)$, and $N = \mathrm{poly}(d, {}^{1}/\Delta, \log(\delta^{-1} \cdot \sum_{i \in [m]} p_i))$ perturbed samples from the $\eta$-Hamming model satisfies*

$$\mathrm{Err}_{\mathcal{D}}(f_{\mathrm{ET}}) - \mathrm{Err}_{\mathcal{D}}(f^\star) \leq \varepsilon, \tag{87}$$

$$\text{for all } r \in [m], \quad \Omega_{\mathcal{D}}^{(r)}(f_{\mathrm{ET}}) \geq \tau - \nu. \tag{88}$$

The proof of Theorem B.2 is similar to the proof of Theorem 4.3. Instead of repeating the entire proof, we highlight the differences between the two proofs.

*Proof.* Set

$$N := \Theta\left(\frac{1}{\Delta^2 \cdot (\lambda - 2\eta)^4} \cdot \left(d \log\left(\frac{d}{\Delta^2 \cdot (\lambda - 2\eta)^4}\right) + \log\left(\delta^{-1} \cdot \sum_{r \in [m]} p_r\right)\right)\right).$$

Fix any fairness constraint $r \in [m]$. Applying Lemma 4.8 to the $r$-th fairness constraint, we get that any $f \in \mathcal{F}$ feasible for Program (General-ErrTolerant) is $(\frac{1-(\eta+\Delta)/(\lambda-2\eta)}{1+(\eta+\Delta)/(\lambda-2\eta)})^2$-stable with respect to the fairness constraint $\Omega^{(r)}$. This satisfies the first condition in Lemma A.8. The second condition holds because any $f \in \mathcal{F}$ feasible for Program (General-ErrTolerant) satisfies the fairness constraint in Equation (85). This allows us to use Lemma A.8; we get that with probability at least $1 - \delta \cdot \frac{p_r}{\sum_i p_i}$, any $f \in \mathcal{F}$ feasible for Program (General-ErrTolerant) satisfies that

$$\Omega_{\mathcal{D}}^{(r)}(f_{\mathrm{ET}}) \geq \tau - \nu.$$

Using the union bound over $r \in [m]$, implies that Equation (88) holds with probability at least $1 - \delta$.

If we can show that $f^\star$ is feasible for Program (General-ErrTolerant), then Lemma A.4 implies Equation (87). Using Assumption 2 and Lemma 4.6, it follows that $f^\star$ satisfies the lower bounds in Equation (86) with probability at least $1 - \frac{\delta}{\sum_i p_i}$. This, along with Lemma 4.8, implies that $f^\star$ is $(\frac{1-(\eta+\Delta)/\lambda}{1+(\eta+\Delta)/\lambda})^2$-stable for all fairness metrics. Then, using Lemma A.10, it follows that $f^\star$ satisfies the constraints for a particular $r \in [m]$ with probability at least $1 - \delta \cdot \frac{p_r}{\sum_i p_i}$. Taking the union bound over all $r \in [m]$, it follows that $f^\star$ is feasible for Program (General-ErrTolerant) with probability at least $1 - \delta$. □

**Remark B.3.** *Like Program (ErrTolerant), in general, Program (General-ErrTolerant) is also a nonconvex optimization program. But, for any arbitrarily small $\alpha > 0$, the techniques from [13] can be used to find an $f \in \mathcal{F}$ that has the optimal objective value for Program (General-ErrTolerant) and that additively violates its fairness constraint (85) by at most $\alpha$ by solving a set of $O\left((\lambda\alpha)^{-m}\right)$ convex programs (see Section C for details on an analogous argument for Program (ErrTolerant)).*

## B.2 Theoretical results for Program (ErrTolerant+)

In this section, we show that Program (ErrTolerant+) offers a better fairness guarantee than Program (ErrTolerant) (up to a constant) if the classifiers in $\mathcal{F}$ do not use the protected attributes for prediction. In particular, we prove Theorem B.4.

**Theorem B.4 (Guarantees for Program (ErrTolerant+)).** *Suppose for each $\ell \in [p]$*

$$\lambda_\ell := \Pr_{\mathcal{D}}\left[\mathcal{E}(f^\star), \mathcal{E}'(f^\star), Z = \ell\right] \quad and \quad \gamma_\ell := \Pr_{\mathcal{D}}[\mathcal{E}'(f^\star), Z = \ell],$$

*$\mathcal{F}$ has VC dimension $d \in \mathbb{N}$, and classifiers in $\mathcal{F}$ do not use the protected attributes for prediction. Let $s$ be the optimal value of Program (7) and $\lambda := \min_{\ell \in [p]} \lambda_\ell$. Then, for all perturbation rates $\eta \in (0, \lambda/2)$, fairness thresholds $\tau \in (0, 1]$, bounds on error $\varepsilon > 2\eta$ and constraint violation*

$$\nu > \tau \cdot \left(1 - s + \frac{4\eta}{(\lambda - 2\eta)}\right),$$

*and confidence parameters $\delta \in (0, 1)$ with probability at least $1 - \delta$, the optimal solution $f_{\mathrm{ET}} \in \mathcal{F}$ of Program (ErrTolerant) with parameters $\eta$, $\lambda$,*

$$\Delta := O\left(\min\left\{\varepsilon - 2\eta, \nu - \tau \cdot \left(1 - s + \frac{4\eta}{(\lambda - 2\eta)}\right), \lambda - 2\eta\right\}\right),$$

*and $N := \mathrm{poly}(d, 1/\Delta, \log(p/\delta))$ perturbed samples from an $\eta$-Hamming adversary satisfies*

$$\mathrm{Err}_{\mathcal{D}}(f_{\mathrm{ET}}) - \mathrm{Err}_{\mathcal{D}}(f^\star) \leq \varepsilon \quad and \quad \Omega_{\mathcal{D}}(f_{\mathrm{ET}}) \geq \tau - \nu. \tag{89}$$

### B.2.1 Proof of Theorem B.4

The proof of Theorem B.4 is similar to the proof of Theorem 4.3. Instead of repeating the entire proof, we highlight the differences from the proof of Theorem 4.3.

*Proof of Theorem B.4.* The proof of Theorem 4.3 has three main steps:

1. Step 1 proves that if $f^\star$ is feasible for Program (ErrTolerant), then the $f_{\mathrm{ET}}$ has accuracy $2\eta$-close to the accuracy of $f^\star$ (Lemma A.4).

2. Step 2 proves that any $f \in \mathcal{F}$ feasible for Program (ErrTolerant) satisfies the fairness guarantee in Theorem 4.3 with high probability (Lemma A.8). The main substep in the proof of Lemma A.8 is to show that any $f \in \mathcal{F}$ feasible for Program (ErrTolerant) is $(\frac{1-(\eta+\Delta)/\lambda}{1+(\eta+\Delta)/\lambda})^2$-stable.

3. Step 3 proves that, with high probability, $f^\star$ is feasible for Program (ErrTolerant) (Lemma A.10). Combining this with Step 1 shows that $f_{\mathrm{ET}}$ satisfies the accuracy guarantee in Theorem 4.3 with high probability.

The proof of Step 1 (Lemma A.4) only depends on the perturbation model and the objective of Program (ErrTolerant); thus, it also generalizes to Program (ErrTolerant+). The proof of Step 2 (Lemma A.8), follows because any $f \in \mathcal{F}$ feasible for Program (ErrTolerant) satisfies the following inequality:

$$\Pr_{\widehat{D}}\left[\mathcal{E}(f), \mathcal{E}'(f), \widehat{Z} = \ell\right] \geq \lambda - \eta - \Delta. \tag{90}$$

Since for all $\ell \in [p]$, $\lambda_\ell \geq \lambda$, any $f \in \mathcal{F}$ feasible for Program (ErrTolerant+) also satisfies Equation (90). Thus, Lemma A.8 also holds for Program (ErrTolerant+). This shows that any $f \in \mathcal{F}$ feasible for Program (ErrTolerant+) satisfies

$$\Omega_{\mathcal{D}}(f_{\mathrm{ET}}) \geq \tau - \frac{8\eta\tau}{\lambda - 2\eta}.$$

However, Theorem B.4 has a tighter fairness guarantee.[5] The tighter guarantee follows by lower bounding $\Omega(f, \widehat{S})$ by $\tau \cdot s$ in Equation (25) in the proof of Lemma A.8.

The main difference in the proofs of Theorem 4.3 and Theorem B.4 is in Step 3. Using Assumption 1 and Lemma A.3, we can show that $f^\star$ satisfies: $\Pr_{\widehat{D}}\left[\mathcal{E}(f), \mathcal{E}'(f), \widehat{Z} = \ell\right] \geq \lambda_\ell - \eta - \Delta$. It remains to show that $f^\star$ satisfies the fairness constraint: $\Omega(f, \widehat{S}) \geq \tau \cdot s$.

In the proof of Lemma A.10 we show a weaker result: $f^\star$ satisfies the fairness constraint: $\Omega(f, \widehat{S}) \geq \tau \cdot (\frac{1-(\eta+\Delta)/\lambda}{1+(\eta+\Delta)/\lambda})^2$. This follows because $f^\star$ is $(\frac{1-(\eta+\Delta)/\lambda}{1+(\eta+\Delta)/\lambda})^2$-stable. Instead, here, we prove that with high probability $f^\star$ satisfies the following inequality

$$\frac{\Omega(f^\star, \widehat{S})}{\Omega_{\mathcal{D}}(f^\star)} \geq s. \tag{91}$$

(Equation (91) does not imply to $s$-stability because $s$-stability also requires an upper bound of $1/s$.) This suffices to show that $f^\star$ is feasible for Program (ErrTolerant+) with high probability; by our discussion so far, it also proves Theorem B.4. $\qquad\square$

It remains to prove Equation (91). Formally, we prove Lemma B.5.

**Lemma B.5.** *Let $s$ be the optimal value of Program* (7)*, then with probability at least $1 - \delta_0$*

$$\frac{\Omega(f^\star, \widehat{S})}{\Omega_{\mathcal{D}}(f^\star)} \geq s, \tag{92}$$

*where $f^\star$ is an optimal solution of Program* (2) *and $\delta_0$ is as defined in Equation* (9)*.*

---

[5] Because for all $\ell \in [p]$, $\lambda_\ell \geq \lambda$ and $\gamma_\ell \geq \lambda$, it can be shown that $s \geq (\frac{1-(\eta+\Delta)/\lambda}{1+(\eta+\Delta)/\lambda})^2$. Thus, the guarantee in Theorem B.4 is stronger than the guarantee in Theorem 4.3.

*Proof.* Let the unperturbed samples be $S := \{(x_i, y_i, z_i)\}_{i \in [N]}$ and the perturbs samples be $\widehat{S} := \{(x_i, y_i, \widehat{z}_i)\}_{i \in [N]}$. We will prove that $\Omega(f, \widehat{S})/\Omega(f, S) \geq s$. Then the result follows as give $N = \text{poly}(d, 1/\Delta, \log(p/\delta_0))$ iid samples from $\mathcal{D}$, it holds that with probability at lest $1 - \delta_0$, $\Omega(f, S)/\Omega_{\mathcal{D}}(f) \geq 1 - \Delta$.

Let $\mathcal{E}$ and $\mathcal{E}'$ be the events defining the fairness metric $\Omega$ (Definition 3.1). For each $\ell \in [p]$, we define

$$\eta_\ell^1 := \text{Pr}_{\widehat{D}}[\mathcal{E}(f^\star(X)), \mathcal{E}'(f^\star(X)), \widehat{Z} = \ell] - \text{Pr}_D[\mathcal{E}(f^\star(X)), \mathcal{E}'(f^\star(X)), Z = \ell],$$

$$\eta_\ell^2 := \text{Pr}_{\widehat{D}}[\neg\mathcal{E}(f^\star(X)), \mathcal{E}'(f^\star(X)), \widehat{Z} = \ell] - \text{Pr}_D[\neg\mathcal{E}(f^\star(X)), \mathcal{E}'(f^\star(X)), Z = \ell].$$

Substituting the values of $\lambda_\ell$ and $\gamma_\ell$, we get that[6]

$$\eta_\ell^1 := \text{Pr}_{\widehat{D}}[\mathcal{E}(f^\star(X)), \mathcal{E}'(f^\star(X)), \widehat{Z} = \ell] - \lambda_\ell, \tag{93}$$

$$\eta_\ell^2 := \text{Pr}_{\widehat{D}}[\neg\mathcal{E}(f^\star(X)), \mathcal{E}'(f^\star(X)), \widehat{Z} = \ell] - (\gamma_\ell - \lambda_\ell). \tag{94}$$

Intuitively, $\eta_\ell^1$ is the number of samples with $\mathbb{I}[\mathcal{E}(f^\star(X)), \mathcal{E}'(f^\star(X))] = 1$ added to protected group $Z = \ell$ and $\eta_\ell^2$ is the number of samples with $\mathbb{I}[\neg\mathcal{E}(f^\star(X)), \mathcal{E}'(f^\star(X))] = 1$ added to protected group $Z = \ell$. (Note that the event $\mathcal{E}(f^\star(X))$ and $\neg\mathcal{E}(f^\star(X))$ are disjoint.)

The values $\{\eta_\ell^1, \eta_\ell^2\}_\ell$ satisfy several conditions: Because the total number of samples added or removed from all protected groups is 0, it holds that $\sum_{\ell \in [p]} \eta_\ell^1 + \eta_\ell^2 = 0$. Moreover, because $f$ does not use $Z$ for prediction, perturbing $Z$ does not change the value of $\mathbb{I}[\mathcal{E}(f^\star(X)), \mathcal{E}'(f^\star(X))]$, and hence, it holds that

$$\sum\nolimits_{\ell \in [p]} \eta_\ell^1 = 0 \quad \text{and} \quad \sum\nolimits_{\ell \in [p]} \eta_\ell^2 = 0. \tag{95}$$

Next, because any $\eta$-Hamming adversary perturbs at most $\eta$-fraction of the samples, we have that

$$\sum\nolimits_{\ell \in [p]} |\eta_\ell^1| + |\eta_\ell^2| \leq 2\eta. \tag{96}$$

Finally, as the probability in the RHS of Equation (94) is nonnegative, it follows that for all $\ell \in [p]$

$$\eta_\ell^2 \geq -(\gamma_\ell - \lambda_\ell). \tag{97}$$

Now we are ready to prove the result

$$\Omega(f^\star, \widehat{S}) := \min_{\ell, k \in [p]} \frac{\text{Pr}_{\widehat{D}}[\mathcal{E}'(f^\star(X)), \widehat{Z} = \ell]}{\text{Pr}_{\widehat{D}}[\mathcal{E}(f^\star(X)), \mathcal{E}'(f^\star(X)), \widehat{Z} = \ell]} \cdot \frac{\text{Pr}_{\widehat{D}}[\mathcal{E}(f^\star(X)), \mathcal{E}'(f^\star(X)), \widehat{Z} = k]}{\text{Pr}_{\widehat{D}}[\mathcal{E}'(f^\star(X)), \widehat{Z} = k]}. \tag{98}$$

Fix any $\ell, k \in [p]$. From Equations (93) and (94) we have that

$$\frac{\text{Pr}_{\widehat{D}}[\mathcal{E}'(f^\star(X)), \widehat{Z} = \ell]}{\text{Pr}_{\widehat{D}}[\mathcal{E}(f^\star(X)), \mathcal{E}'(f^\star(X)), \widehat{Z} = \ell]} \frac{\text{Pr}_{\widehat{D}}[\mathcal{E}(f^\star(X)), \mathcal{E}'(f^\star(X)), \widehat{Z} = k]}{\text{Pr}_{\widehat{D}}[\mathcal{E}'(f^\star(X)), \widehat{Z} = k]} = \frac{\lambda_\ell + \eta_\ell^1}{\gamma_\ell + \eta_\ell^1 + \eta_\ell^2} \cdot \frac{\gamma_k + \eta_k^1 + \eta_k^2}{\lambda_k + \eta_k^1}.$$

Our goal is to lower bound the LHS of the above equation because it implies a lower bound on $\Omega(f^\star, \widehat{S})$ by Equation (98). Towards this, given vectors $\eta^1, \eta^2 \in \mathbb{R}^p$, define the following objective:

$$\text{Obj}(\eta^1, \eta^2) := \frac{\lambda_\ell + \eta_\ell^1}{\gamma_\ell + \eta_\ell^1 + \eta_\ell^2} \cdot \frac{\gamma_k + \eta_k^1 + \eta_k^2}{\lambda_k + \eta_k^1}.$$

We would like to lower bound $\text{Obj}(\eta^1, \eta^2)$ subject to Equations (95) to (97), i.e., we would like to lower bound the value of the following program

$$\text{Obj}(\eta^1, \eta^2), \tag{99}$$

$$\text{s.t.,} \quad \text{eqs. (95), (96) and (97) hold}$$

We claim that for all $i \notin \{\ell, k\}$ it is optimal to set $\eta_i^1 = 0$ in Program (99). To see this, note that given any solution $\eta^1, \eta^2 \in \mathbb{R}^p$ to Program (99), with $\alpha := \sum_{i \neq \ell, k} \eta_i^1$, we can construct another solution $\delta^1, \delta^2 \in \mathbb{R}^p$ that has a better objective while satisfying $\delta_i^1 = 0$ for all $i \notin \{\ell, k\}$: For all

---

[6]Here, we implicitly use the fact that $f \in \mathcal{F}$ does not use the protected attribute $Z$ for prediction.

$i \notin \{\ell, k\}$, set $\delta_i^1 := 0$. Next, if $\alpha > 0$, set $\delta_k^1 := \eta_k^1 + \alpha$, and otherwise, set $\delta_\ell^1 := \eta_\ell^1 + \alpha$. Finally, let all other variables remain unchanged. (Under Equation (97), $(\delta^1, \delta^2)$ has a smaller objective than $(\eta^1, \eta^2)$.) Thus, we get that

$$
\begin{array}{lll}
\min\limits_{\eta^1, \eta^2 \in \mathbb{R}^p} \mathrm{Obj}(\eta^1, \eta^2), & = & \min\limits_{\eta^1, \eta^2 \in \mathbb{R}^p} \mathrm{Obj}(\eta^1, \eta^2), \\
\text{s.t.,} \quad \text{eqs. (95), (96) and (97) hold} & & \text{s.t.,} \quad \text{eqs. (95), (96) and (97) hold} \\
& & \text{for all } i \notin \{\ell, k\}, \eta_i^1 = 0
\end{array} \tag{100}
$$

Next, dropping Equation (97) from the constraint, which only improves the objective, we get that:

$$
\begin{array}{lll}
\min\limits_{\eta^1, \eta^2 \in \mathbb{R}^p} \mathrm{Obj}(\eta^1, \eta^2), & \geq & \min\limits_{\eta^1, \eta^2 \in \mathbb{R}^p} \mathrm{Obj}(\eta^1, \eta^2), \\
\text{s.t.,} \quad \text{eqs. (95), (96) and (97) hold} & & \text{s.t.,} \quad \text{eqs. (95) and (96) hold} \\
\text{for all } i \notin \{\ell, k\}, \eta_i^1 = 0 & & \text{for all } i \notin \{\ell, k\}, \eta_i^1 = 0.
\end{array} \tag{101}
$$

We claim that for all $i \notin \{\ell, k\}$ it is optimal to set $\eta_i^2 := 0$ in the program in the RHS of Equation (101). This follows by a similar construction used to prove Equation (100). To see this, note that given any solution $\eta^1, \eta^2 \in \mathbb{R}^p$ to the program in the RHS of Equation (101), with $\alpha := \sum_{i \neq \ell, k} \eta_i^2$, there is another solution $\delta^1, \delta^2 \in \mathbb{R}^p$ that has a better objective value while satisfying $\delta_i^2 = 0$ for all $i \notin \{\ell, k\}$: For all $i \notin \{\ell, k\}$, set $\delta_i^1 := 0$. Next, if $\alpha > 0$, set $\delta_\ell^2 := \eta_\ell^2 + \alpha$, otherwise set $\delta_k^2 := \eta_k^2 + \alpha$. Finally, let all other variables remain unchanged. $((\delta^1, \delta^2)$ always has a smaller objective than $(\eta^1, \eta^2)$.) Thus, we have

$$
\begin{array}{lll}
\min\limits_{\eta^1, \eta^2 \in \mathbb{R}^p} \mathrm{Obj}(\eta^1, \eta^2), & = & \min\limits_{\eta^1, \eta^2 \in \mathbb{R}^p} \mathrm{Obj}(\eta^1, \eta^2), \\
\text{s.t.,} \quad \text{eqs. (95) and (96) hold} & & \text{s.t.,} \quad \text{eqs. (95) and (96) hold} \\
\text{for all } i \notin \{\ell, k\}, \eta_i^1 = 0 & & \text{for all } i \notin \{\ell, k\}, \eta_i^1 = 0, \eta_i^2 = 0.
\end{array} \tag{102}
$$

Rewriting the program in the RHS of Equation (102), by dropping the always 0 variables, we get

$$
\begin{array}{lll}
\min\limits_{\eta^1, \eta^2 \in \mathbb{R}^p} \mathrm{Obj}(\eta^1, \eta^2), & = & \min\limits_{\eta_\ell^1, \eta_k^1, \eta_\ell^2, \eta_k^2 \in \mathbb{R}} \dfrac{\lambda_\ell + \eta_\ell^1}{\gamma_\ell + \eta_\ell^1 + \eta_\ell^2} \cdot \dfrac{\gamma_k + \eta_k^1 + \eta_k^2}{\lambda_k + \eta_k^1}, \\
\text{s.t.,} \quad \text{eqs. (95) and (96) hold} & & \text{s.t.,} \quad \eta_\ell^1 = -\eta_k^1 \text{ and } \eta_\ell^2 = -\eta_k^2 \\
\text{for all } i \notin \{\ell, k\}, \eta_i^1 = 0, \eta_i^2 = 0 & & |\eta_\ell^1| + |\eta_\ell^2| \leq \eta
\end{array} \tag{103}
$$

Rewriting the program in the RHS of Equation (103), we get

$$
\begin{array}{lll}
\min\limits_{\eta_\ell^1, \eta_k^1, \eta_\ell^2, \eta_k^2 \in \mathbb{R}} \dfrac{\lambda_\ell + \eta_\ell^1}{\gamma_\ell + \eta_\ell^1 + \eta_\ell^2} \cdot \dfrac{\gamma_k + \eta_k^1 + \eta_k^2}{\lambda_k + \eta_k^1}, & = & \min\limits_{\eta_\ell, \eta_k \in \mathbb{R}} \dfrac{\lambda_\ell - \eta_\ell}{\gamma_\ell - \eta_\ell + \eta_k} \cdot \dfrac{\gamma_k + \eta_\ell - \eta_k}{\lambda_k + \eta_\ell}, \\
\text{s.t.,} \quad \eta_\ell^1 = -\eta_k^1 \text{ and } \eta_\ell^2 = -\eta_k^2, & & \text{s.t.,} \quad |\eta_\ell^1| + |\eta_\ell^2| \leq \eta \\
|\eta_\ell^1| + |\eta_\ell^2| \leq \eta
\end{array} \tag{104}
$$

In the program in the RHS of Equation (104), it is optimal to set $\eta_\ell, \eta_k \geq 0$; this simplifies the constraint $|\eta_\ell^1| + |\eta_\ell^2| \leq \eta$ to $\eta_\ell^1 + \eta_\ell^2 \leq \eta$. The sequence of equations, Equations (100) to (104), implies that

$$
\begin{aligned}
\Omega(f^\star, \widehat{S}) & \geq \min_{\ell, k \in [p]} \min_{\eta_\ell, \eta_k \geq 0} \frac{\lambda_\ell - \eta_\ell}{\gamma_\ell - \eta_\ell + \eta_k} \cdot \frac{\gamma_k + \eta_\ell - \eta_k}{\lambda_k + \eta_\ell}, \quad \text{s.t.,} \quad \eta_\ell^1 + \eta_\ell^2 \leq \eta \\
& = \min_{\ell, k \in [p]} \frac{\lambda_\ell \cdot \gamma_k}{\gamma_\ell \cdot \lambda_k} \cdot \min_{\eta_\ell, \eta_k \geq 0} \frac{1 - \eta_\ell/\lambda_\ell}{1 - (\eta_\ell - \eta_k)/\gamma_\ell} \cdot \frac{1 + (\eta_\ell - \eta_k)/\gamma_k}{1 + \eta_\ell/\lambda_k}, \quad \text{s.t.,} \quad \eta_\ell^1 + \eta_\ell^2 \leq \eta \\
& = \Omega(f^\star, S) \cdot \min_{\ell, k \in [p]} \min_{\eta_\ell, \eta_k \geq 0} \frac{1 - \eta_\ell/\lambda_\ell}{1 - (\eta_\ell - \eta_k)/\gamma_\ell} \cdot \frac{1 + (\eta_\ell - \eta_k)/\gamma_k}{1 + \eta_\ell/\lambda_k}, \quad \text{s.t.,} \quad \eta_\ell^1 + \eta_\ell^2 \leq \eta \\
& \hspace{10cm} \text{(By the definition of } \lambda_\ell \text{ and } \gamma_\ell) \\
& = \Omega(f^\star, S) \cdot s. \hspace{8cm} \text{(By the definition of } s)
\end{aligned}
$$

$\qquad\square$

# C   Reduction from Program (ErrTolerant) to a set of convex programs

In general, Program (ErrTolerant) is a nonconvex optimization program. But, we can reduce Program (ErrTolerant) to a set of convex programs. Formally, for any arbitrarily small $\alpha > 0$, we can find an $f \in \mathcal{F}$ that has the optimal objective value for Program (ErrTolerant) and that additively violates its fairness constraint (Equation (4)) by at most $\alpha$, by solving a set of $O(1/(\lambda\alpha))$ convex programs. In this section, we present this reduction. It largely follows from [13], but is included for completeness.

Recall that given a fairness metric $\Omega$ and corresponding events $\mathcal{E}$ and $\mathcal{E}'$ (as in Definition 3.1), perturbed samples $\widehat{S}$, whose empirical distribution is $\widehat{D}$, a perturbation rate $\eta \in [0, 1]$, and constants $\lambda, \Delta \in (0, 1]$, Program (ErrTolerant) is the following program:

$$\min_{f \in \mathcal{F}} \quad \mathrm{Err}_{\widehat{D}}(f), \tag{105}$$

$$\text{s.t.,} \quad \Omega(f, \widehat{S}) \geq \tau \cdot \left( \frac{1 - (\eta + \Delta)/\lambda}{1 + (\eta + \Delta)/\lambda} \right)^2,$$

$$\forall \ell \in [p], \ \mathrm{Pr}_{\widehat{D}}\left[ \mathcal{E}(f), \mathcal{E}'(f), \widehat{Z} = \ell \right] \geq \lambda - \eta - \Delta.$$

Equivalently defining scalars $\widehat{\tau} := \left( \frac{1 - (\eta + \Delta)/\lambda}{1 + (\eta + \Delta)/\lambda} \right)^2$ and $\widehat{\lambda} := \lambda - \eta - \Delta$, our goal is to solve

$$\min_{f \in \mathcal{F}} \quad \mathrm{Err}_{\widehat{D}}(f), \tag{106}$$

$$\text{s.t.,} \quad \Omega(f, \widehat{S}) \geq \widehat{\tau}, \tag{107}$$

$$\forall \ell \in [p], \ \mathrm{Pr}_{\widehat{D}}\left[ \mathcal{E}(f), \mathcal{E}'(f), \widehat{Z} = \ell \right] \geq \widehat{\lambda}. \tag{108}$$

**Remark C.1.** *All references to the results in [13] are to its arXiv version.*

**Remark C.2.** *In this section, all probabilities and expectations are with respect to the draw of perturbed samples $(X, Y, \widehat{Z})$. Given $\widehat{D}$, the empirical distribution over $\widehat{S}$, we use $\mathrm{Pr}_{\widehat{D}}[\cdot]$ to denote $\mathrm{Pr}_{(X,Y,\widehat{Z}) \sim \widehat{D}}[\cdot]$ and $\mathbb{E}_{\widehat{D}}[\cdot]$ to denote $\mathbb{E}_{(X,Y,\widehat{Z}) \sim \widehat{D}}[\cdot]$.*

## C.1   Performance metrics in Definition 3.1 are a special case of the metrics in [13]

To use the results in [13], we need to show that Definition 3.1 is a special case of [13, Definition 2.3].

**Lemma C.3.** *Suppose $\mathcal{F} := \{0, 1\}^{\mathcal{X} \times [p]}$. For all events $\mathcal{E}$ and $\mathcal{E}'$, that can depend on $f$, the corresponding metric $q$ is a "performance function" as defined in [13, Definition 2.3].*

*Proof.* Observe that

$$q_\ell(f) := \mathrm{Pr}_{\widehat{D}}[\mathcal{E}(f) \mid \mathcal{E}'(f), \widehat{Z} = \ell] = \frac{\mathrm{Pr}_{\widehat{D}}[\mathcal{E}(f), \mathcal{E}'(f) \mid \widehat{Z} = \ell]}{\mathrm{Pr}_{\widehat{D}}[\mathcal{E}'(f) \mid \widehat{Z} = \ell]}.$$

To simplify the notation, below, we use $f$ to denote $f(X, \widehat{Z})$. We can rewrite the denominator as:

$$\begin{aligned}
\mathrm{Pr}_{\widehat{D}}[\mathcal{E}'(f) \mid \widehat{Z} = \ell] &= \mathrm{Pr}_{\widehat{D}}[\mathcal{E}'(f) \mid \widehat{Z} = \ell] \\
&= \mathrm{Pr}_{\widehat{D}}[\mathcal{E}'(f) \mid f = 0, \widehat{Z} = \ell] \cdot \mathrm{Pr}_{\widehat{D}}[f = 0 \mid \widehat{Z} = \ell] \\
&\quad + \mathrm{Pr}_{\widehat{D}}[\mathcal{E}'(f) \mid f = 1, \widehat{Z} = \ell] \cdot \mathrm{Pr}_{\widehat{D}}[f = 1 \mid \widehat{Z} = \ell] \\
&= c_0 \cdot \mathrm{Pr}_{\widehat{D}}[f = 0 \mid \widehat{Z} = \ell] + c_1 \cdot \mathrm{Pr}_{\widehat{D}}[f = 1 \mid \widehat{Z} = \ell],
\end{aligned}$$

where we defined

$$c_0 := \mathrm{Pr}_{\widehat{D}}[\mathcal{E}'(f) \mid f = 0, \widehat{Z} = \ell],$$
$$c_1 := \mathrm{Pr}_{\widehat{D}}[\mathcal{E}'(f) \mid f = 1, \widehat{Z} = \ell].$$

Let $\alpha_0 := c_0$ and $\alpha_1 := c_1 - c_0$. Then, we have

$$\begin{aligned}
\mathrm{Pr}_{\widehat{D}}[\mathcal{E}'(f) \mid \widehat{Z} = \ell] &= c_0 \cdot \left( 1 - \mathrm{Pr}_{\widehat{D}}[f = 1 \mid \widehat{Z} = \ell] \right) + c_1 \cdot \mathrm{Pr}_{\widehat{D}}[f = 1 \mid \widehat{Z} = \ell] \\
&= \alpha_0 + \alpha_1 \cdot \mathrm{Pr}_{\widehat{D}}[f = 1 \mid \widehat{Z} = \ell] \qquad \text{(Using } \alpha_0 := c_0 \text{ and } \alpha_1 := c_1 - c_0\text{)} \\
&= \alpha_0 + \alpha_1 \cdot \mathrm{Pr}_{\widehat{D}}[f(X, \widehat{Z}) = 1 \mid \widehat{Z} = \ell]. \tag{109}
\end{aligned}$$

Where the last equality follows due to our notation that the event $f = 1$ denotes $f(X, \widehat{Z}) = 1$ for random draws $(X, Y, \widehat{Z}) \sim \widehat{D}$. Next, by replacing $\mathcal{E}'(f)$ by $\mathcal{E}(f) \wedge \mathcal{E}'(f)$, we get that

$$\mathrm{Pr}_{\widehat{D}}[\mathcal{E}(f), \mathcal{E}'(f) \mid \widehat{Z} = \ell] = \beta_0 + \beta_1 \cdot \mathrm{Pr}_{\widehat{D}}[f(X, \widehat{Z}) = 1 \mid \widehat{Z} = \ell]. \tag{110}$$

for some $0 \leq \beta_0 \leq 1$ and $-1 \leq \beta_1 \leq 1$. Comparing Equations (109) and (110) with [13, Definition 2.3], it follows that $q_\ell(f)$ is a special case of the performance functions in [13, Definition 2.3]. $\quad\square$

## C.2 Reduction from Program (ErrTolerant) to a set of convex programs

Before stating the result, we need some additional notation.

**Definition C.4.** *Given a fairness metric $\Omega$, the corresponding performance metric $q_\ell$ (from Definition 3.1), desired fairness threshold $\tau \in (0, 1]$, approximation parameter $\alpha \in (0, 1]$, and lower and upper bounds $L, U \in [0, 1]$, define the sets $K(\tau, \alpha), P(L, U) \subseteq \mathcal{F}$ as*

$$K(\tau, \alpha) := \left\{ f \in \mathcal{F} \colon \min_{\ell \in [p]} q_\ell(f) \geq \tau \cdot \max_{\ell \in [p]} q_\ell(f) - \alpha \right\},$$
$$P(L, U) := \{ f \in \mathcal{F} \colon \text{for all } \ell \in [p],\ L \leq q_\ell(f) \leq U \}.$$

Note that $K(\tau, 0)$ (i.e., setting $\alpha = 0$) is the set of classifiers that satisfy the fairness constraint $\Omega(f) \geq \tau$ exactly. $K(\tau, \alpha)$ $(\alpha > 0)$ is the set of classifiers that satisfy a relaxation of this constraint. Formally, for any $\alpha > 0$, the set of classifiers $\alpha$-feasible for Program (106) are all $f$ in $K(\tau, \alpha)$ that also satisfy Equation (108). Under Assumption 1, any $\alpha$-feasible classifier $f \in \mathcal{F}$ additively violates the fairness constraint in Program (106) by at most $\alpha/\lambda$. To see this, suppose $f \in \mathcal{F}$ is $\alpha$-stable, then

$$\min_{\ell \in [p]} q_\ell(f) \geq \tau \cdot \max_{\ell \in [p]} q_\ell(f) - \alpha$$
$$= \left( \tau - \frac{\alpha}{\max_{\ell \in [p]} q_\ell(f)} \right) \cdot \max_{\ell \in [p]} q_\ell(f)$$
$$\geq \left( \tau - \frac{\alpha}{\lambda} \right) \cdot \max_{\ell \in [p]} q_\ell(f), \qquad \text{(Using that } \max_{\ell \in [p]} q_\ell(f) \geq \lambda\text{)}$$

and hence,

$$\Omega_{\widehat{D}}(f) = \frac{\min_{\ell \in [p]} q_\ell(f)}{\min_{\ell \in [p]} q_\ell(f)} \geq \tau - \frac{\alpha}{\lambda}. \tag{111}$$

Using the above notation, we can write Program (106) as follows:

$$\min_{f \in \mathcal{F}} \quad \text{Err}_{\widehat{D}}(f),$$
$$\text{s.t.,} \quad f \in K(\widehat{\tau}, 0),$$
$$\forall\, \ell \in [p],\ \Pr_{\widehat{D}}\left[ \mathcal{E}(f), \mathcal{E}'(f), \widehat{Z} = \ell \right] \geq \widehat{\lambda}.$$

Here, $K(\widehat{\tau}, 0)$ can be a nonconvex set. But as [13] show, it can be approximated as a union of convex sets. In particular, they approximate $K(\widehat{\tau}, 0)$ as the union $\bigcup_{j=1}^{J} P(L_j, U_j)$ for some $J \in \mathbb{N}$ and vectors $L, U \in [0, 1]^J$. (One can prove that for all $L, U \in [0, 1]$, $P(L, U)$ is a convex set [13].)

**Theorem C.5** (**Implicit in [13, Theorem 3.1]**). *Given constants $\tau, \alpha \in (0, 1]$, let $J := \lceil \tau/\alpha \rceil$, and for all $j \in [J]$, let $L_j := (j-1)\alpha$ and $U_j := (j\alpha)/\tau$. For all fairness metrics $\Omega$ and corresponding performance metric $q_\ell$ (as defined in Definition 3.1) it holds that*

$$K(\tau, 0) \subseteq \bigcup_{j=1}^{J} P(L_j, U_j) \subseteq K(\tau, \alpha).$$

Theorem C.5 allows one to reduce the problem of finding an $\alpha$-feasible classifier to solving a set of $\lceil \tau/\alpha \rceil$ convex programs of the following form: For some $L, U \in [0, 1]$

$$\min_{f \in \mathcal{F}} \quad \text{Err}_{\widehat{D}}(f), \tag{112}$$
$$\text{s.t.,} \quad f \in P(L, U), \tag{113}$$
$$\forall\, \ell \in [p],\ \Pr_{\widehat{D}}\left[ \mathcal{E}(f), \mathcal{E}'(f), \widehat{Z} = \ell \right] \geq \widehat{\lambda}. \tag{114}$$

**Theorem C.6.** *Given constants $\tau, \alpha \in (0, 1]$, let $J := \lceil \tau/\alpha \rceil$, and for all $j \in [J]$, let $L_j := (j-1)\alpha$ and $U_j := (j\alpha)/\tau$. Further, let $f_j$ be the optimal solution of Equation (112) with $L := L_j$ and $U := U_j$. Then, $f_\alpha := \arg\min_{f_j} \text{Err}(f_j, \widehat{S})$ has the optimal accuracy for Program (106) and is $\alpha$-feasible for Program (106).*

If Assumption 1 holds, then using Theorem C.6 and Equation (111), we can to find an $f \in \mathcal{F}$ that has the optimal objective value for Program (ErrTolerant) and that additively violates its fairness constraint (4) by at most $\alpha$ by solving a set of $\lceil \widehat{\tau}/(\lambda\alpha) \rceil$ convex programs.

*Proof of Theorem C.6.*

**A. Fairness guarantee:** Since $f_\alpha$ is an optimal solution of Program (112) with $L := L_j$ and $U := U_j$ for some $r \in [J]$, $f_\alpha \in P(L_r, U_r)$, and hence, $f \in \bigcup_{j=1}^{J} P(L_j, U_j)$. Therefore, by Theorem C.5, $f_\alpha \in K(\tau, \alpha)$. Since $f_\alpha \in K(\tau, \alpha)$ and $f_\alpha$ satisfies Equation (114), $f_\alpha$ is an $\alpha$-feasible solution for Program (106).

**B. Accuracy guarantee:** Let $f_{\mathrm{ET}} \in \mathcal{F}$ be the optimal solution of Program (106). Since $f_{\mathrm{ET}} \in K(\tau, 0)$ and by Theorem C.5 the inclusion $\bigcup_{j=1}^{J} P(L_j, U_j) \supseteq K(\tau, 0)$ holds, $f_{\mathrm{ET}} \in \bigcup_{j=1}^{J} P(L_j, U_j)$. Further, since $f_\alpha$ minimizes $\mathrm{Err}(\cdot, \widehat{S})$ over $\bigcup_{j=1}^{J} P(L_j, U_j)$ and by Theorem C.5 the containment $\bigcup_{j=1}^{J} P(L_j, U_j) \supseteq K(\tau, 0)$ holds, it follows that $\mathrm{Err}(f_\alpha, \widehat{S}) \leq \mathrm{Err}(f_{\mathrm{ET}}, \widehat{S})$. $\square$

# D    Further comparison to related work

## D.1    Other related work

**Fair classification without perturbations.**    A large body of work studies fair classification. Here, several works frame fair classification as a constrained optimization program and develop algorithms to solve these programs [65, 63, 62, 48, 27, 1, 13]. A different approach is to alter the decision boundary of a given classifier to improve its fairness [26, 32, 28, 51, 61, 24] (possibly with different alterations for different protected groups). Furthermore, some works preprocesses the training data to "correct" for its bias [37, 47, 38, 64, 25, 42]. However, these works require the protected attributes in the training samples to be known exactly, whereas in this paper we study the setting where fraction of the protected attributes are arbitrarily corrupted.

**Missing protected attributes.**    Some works have studied fair classification in the absence of protected attributes–using auxiliary data. For example, [30] use other variables as proxies for protected attributes and [19] augment their dataset with "related data" (that includes protect attributes) to control fairness. In the absence of auxiliary data, [33] use distributionally robust optimization to minimize the maximum empirical risk across the protected groups, and [43] use a neural network to identify "potential" protected groups. However, these approaches do not offer provable guarantees on accuracy (with respect to $f^\star$). In contrast, our approach uses perturbed protected attributes and comes with provable guarantees on fairness and accuracy (with respect to $f^\star$).

**Stochastic perturbations in labels.**    [10, 58] study fair classification with perturbations in the labels: [10] consider a model where perturbations arise due to bias in the training samples. They show that, under some models of bias, adding fairness constraints can improve the accuracy of the classifier on the unbiased data. [58] consider a model where the labels in each protected group are perturbed to a different value independently with a known (group-dependent) probability; they give a framework for a non-binary protected attribute that provably outputs a classifier with near-optimal accuracy that nearly satisfies the fair constraint with respect to equalized odds, true-positive rate, or false-positive rate fairness constraints. In contrast, we focus on adversarial perturbations in the features, and our framework can be extended to adversarial perturbations in both features and labels (see Section A.1.5). Finally, our framework works for a large class of linear-fractional fairness metrics (which include true-positive rate and false-positive rate fairness constraints, and can ensure equalized odds fairness).

## D.2    Performance of prior frameworks under the $\eta$-Hamming model

In this section, we present examples showing that prior frameworks for fair classification can have low accuracy and fairness compared to our framework under the $\eta$-Hamming model.

### D.2.1    [11]'s framework can output classifiers with low statistical rate

In this section, for any $\delta \in (0, 1/4)$, we give an example (Example D.1) where with high probability [11]'s framework outputs a classifier $f_{\mathrm{OPT}}$ that has perfect accuracy and 0 statistical rate. On the same example, an optimal solution $f_{\mathrm{ET}}$ of Program (ErrTolerant) has accuracy $1 - \delta$ and perfect statistical rate 1.

**Example D.1.** *Fix $\mathcal{X}$ to be any set with at least two distinct points, say $x_A$ and $x_B$. Let $\mathcal{F}$ be any hypothesis class that shatters the set $\{x_A, x_B\} \times [2] \subseteq \mathcal{X} \times [p]$. Define the distribution $\mathcal{D}$ as follows*

$$\Pr_{\mathcal{D}}[X = x, Y = y, Z = z] := \begin{cases} 1/3 - \delta/3 & \text{if } x = x_A, y = 1, z = 1, \\ 1/3 - \delta/3 & \text{if } x = x_B, y = 0, z = 1, \\ \delta & \text{if } x = x_A, y = 0, z = 2, \\ 1/3 - \delta/3 & \text{if } x = x_B, y = 0, z = 2, \\ 0 & \text{otherwise,} \end{cases}$$

*where $\delta$ is some constant smaller than $1/4$. Note that for a sample $(X, Y, Z) \sim \mathcal{D}$, conditioned on $X = x$ and $Z = z$, $Y$ takes the value $\mathbb{I}[x = x_A, z = 1]$. Thus, the classifier $f_{\mathrm{OPT}}(x, z) := \mathbb{I}[x = x_A, z = 1]$ has 0 predictive error. One can verify that $f_{\mathrm{OPT}}$ has a statistical rate of 0 with respect to $\mathcal{D}$. Since $\mathcal{F}$ shatters $\{x_A, x_B\} \times [2]$, $\mathcal{F}$ contains $f_{\mathrm{OPT}}(x, z)$. Further, any other classifier in $\mathcal{F}$ has an error at least $\delta$ with respect to $\mathcal{D}$.*

*[11]'s framework outputs the classifier with the minimum empirical risk on the given samples. Suppose the perturbation rate is $\eta := 0$. Then, given a sufficient number of samples from $\mathcal{D}$, with high probability, $f_{\mathrm{OPT}} \in \mathcal{F}$ has the minimum empirical error, and hence, is output by [11]'s framework; $f_{\mathrm{OPT}}$ satisfies*

$$\mathrm{Err}_{\mathcal{D}}(f_{\mathrm{OPT}}) = 0 \quad \text{and} \quad \Omega_{\mathcal{D}}(f_{\mathrm{OPT}}) = 0,$$

*where $\Omega$ is the statistical rate fairness metric.*

*Next, we show that on this example, Program (ErrTolerant) outputs a classifier with a large statistical rate. Set the fairness threshold to be any value $\tau < 1$. Fix any $\lambda \leq \delta$ (this ensures that Assumption 1 is satisfied). Fix any $\Delta > 0$. Finally, as mentioned, $\eta := 0$.*

*One can verify $f_{\mathrm{ET}}(x, z) := \mathbb{I}[x = x_A]$ has error $\mathrm{Err}_{\mathcal{D}}(f_{\mathrm{ET}}) = \delta$ and a statistical rate of 1 with respect to $\mathcal{D}$. In contrast, any other classifier with statistical rate at least $1 - 2\delta$ has error at least $1/3 - \delta/3 > \delta$ with respect to $\mathcal{D}$. (O1) Using this, one can show that, given a sufficient number of iid samples from $\mathcal{D}$, with high probability, any other classifier feasible for Equation (4) in Program (ErrTolerant) has an error larger than the error of $f_{\mathrm{ET}}$ (on the given samples). (O2) Further, because $\lambda \leq \delta$, one can verify that given a sufficient number of iid samples from $\mathcal{D}$, with high probability, $f_{\mathrm{ET}}$ satisfies Equation (5) in Program (ErrTolerant); thus, with high probability, $f_{\mathrm{ET}}$ is feasible for Program (ErrTolerant).*

*Combining observations (O1) and (O2), we get that: Given a sufficient number of iid samples from $\mathcal{D}$, with high probability, $f_{\mathrm{ET}}$ is the optimal solution Program (ErrTolerant) with parameters $\tau = 1 - \delta$, $\lambda = \delta$, $\eta = 0$, and $\Delta > 0$; $f_{\mathrm{ET}}$ satisfies*

$$\mathrm{Err}_{\mathcal{D}}(f_{\mathrm{ET}}) = \delta \quad \text{and} \quad \Omega_{\mathcal{D}}(f_{\mathrm{ET}}) = 1.$$

### D.2.2 [44]'s and [14]'s frameworks can output classifiers with low accuracy

In this section, for any $\eta \in (0, 1/2)$, we give an example where with high probability [44]'s and [14]'s frameworks output classifiers $f_L$ and $f_C$ (respectively) whose error is at least $1/4$ higher than the error of $f^\star$ (Theorem D.3); where $f^\star$ is an optimal solution to Program (2). On the same example, an optimal solution of Program (ErrTolerant) has error within $2\eta$ of the error of $f^\star$ and violates the fairness constraint by at most $O(\eta)$.

[44] and [14] take parameters $\delta_L, \tau \in [0, 1]$ as input; these parameters control the desired fairness, where decreasing $\delta_L$ or increasing $\tau$ increases the desired fairness. [14] also takes the constant $\lambda$ from Assumption 1 as input. In addition, both [44] and [14] require group specific perturbation rates as input: for each pair $\ell, k \in [p]$, they require $P\ell k := \Pr_D[\widehat{Z} = k \mid Z = \ell]$.

Let $P \in [0, 1]^{p \times p}$ denote the resulting matrix. To give a meaningful estimate of $P$ with adversarial noise, we define the following restriction of the Hamming adversary.

**Definition D.2** (*$P$-restricted Hamming adversary*). *Given a matrix $P \in [0, 1]^{p \times p}$ and $N \in \mathbb{N}$ samples $\{(x_i, y_i, z_i)\}_{i \in [N]}$, for each $\ell \in [p]$, let $G_\ell := \{i \in [N] \mid z_i = \ell\}$ be the set of samples with protected attribute $\ell$. For each $\ell, k \in [p]$, the $P$-restricted Hamming adversary $A_{RH}$ chooses $P\ell k \cdot |G_\ell|$ samples $i \in [N]$ from $G_\ell$, and perturbs their protected attribute $z_i$ from $\ell$ to $\widehat{z}_i = k$.[7]*

---

[7] We assume that $P\ell k \cdot N_\ell$ is integral for all $\ell, k \in [p]$. This can be ensured by slightly increasing $P\ell k$ or $N$.

The modifier "$P$-restricted" refers to the restriction placed by the matrix $P$ on the adversary. Let $\mathcal{A}_{RH}(P)$ be the set of all $P$-restricted Hamming adversaries. Then one can show that $\mathcal{A}_{RH}(P) \subseteq \mathcal{A}(\eta)$ for any $\eta \geq \max_{\ell \in [p]} \sum_{k \in [p]} P\ell k$.

**Theorem D.3.** *Suppose that there are two protected groups ($p := 2$) and $\mathcal{X}$ contains at least two distinct points. Then, there is a family of hypothesis classes $\mathcal{F}$ such that for all fairness thresholds $\tau \in (0, 1]$ and perturbation rates $\eta \in (0, 1/2)$, there is*

1. *a distribution $\mathcal{D}$ over $\mathcal{X} \times \{0, 1\} \times [2]$ that satisfies Assumption 1 with $\lambda := \tau/4$,*

2. *a matrix $P \in [0, 1]^{2 \times 2}$ such that $\mathcal{A}_{RH}(P) \subseteq \mathcal{A}(\eta)$, and*

3. *an adversary $A_{RH} \in \mathcal{A}_{RH}(P)$ that perturbs at most $\eta$-fraction of the samples,*

*such that, if the fairness metric is statistical rate, then for a draw of $N \in \mathbb{N}$ iid samples $S$ from $\mathcal{D}$, with probability at least $1 - e^{-\Omega(\eta^2 \tau^2 N)}$ (over the draw of $S$), it holds that the optimal classifiers*

1. *$f_C \in \mathcal{F}$ of [14]'s program with parameters $P$, $\lambda$, and $\tau$ and samples $A_{RH}(S)$,*

2. *$f_L \in \mathcal{F}$ of [44]'s program with parameters $P$ and $\delta_L := 1/2 - \tau/2$ and samples $A_{RH}(S)$, and[8]*

3. *$f_{ET} \in \mathcal{F}$ of Program (ErrTolerant) with parameters $\eta$, $\lambda$, and $\tau$ and samples $A_{RH}(S)$*

*have errors*

$$\mathrm{Err}_{\mathcal{D}}(f_C) - \mathrm{Err}_{\mathcal{D}}(f^\star) \geq \frac{1}{4}, \tag{115}$$

$$\mathrm{Err}_{\mathcal{D}}(f_L) - \mathrm{Err}_{\mathcal{D}}(f^\star) \geq \frac{1}{4}, \tag{116}$$

$$\mathrm{Err}_{\mathcal{D}}(f_{ET}) - \mathrm{Err}_{\mathcal{D}}(f^\star) \leq 2\eta. \tag{117}$$

*Further, $f_{ET}$ has statistical rates at least $\tau - O(\eta/\tau)$ with respect to $\mathcal{D}$, i.e., $\Omega_{\mathcal{D}}(f_{ET}) \geq \tau - O(\eta/\tau)$.*

**Proof for Theorem D.3.** Let $S := \{(x_i, y_i, z_i)\}_{i \in [N]}$ denote $N$ iid samples from $\mathcal{D}$.

**Setting $P, A$, and $\mathcal{D}$.** We let $P := \begin{bmatrix} 1-\eta_1 & \eta_1 \\ \eta_2 & 1-\eta_2 \end{bmatrix}$, where $\eta_1 := 0$ and $\eta_2 := \eta$. Since $\eta = \max_{\ell \in [p]} \sum_{k \in [p]} P\ell k$, we can verify that $\mathcal{A}_{RH}(P) \subseteq \mathcal{A}(\eta)$. We fix $A \in \mathcal{A}_{RH}(P)$ to be the following algorithm.

---

**Input.** A perturbation rate $\eta > 0$, matrix $P := \begin{bmatrix} 1-\eta_1 & \eta_1 \\ \eta_2 & 1-\eta_2 \end{bmatrix}$, where $\eta_1, \eta_2 \in [0, 1]$, and samples $S := \{(x_i, y_i, z_i)\}_{i \in [N]}$

**Output.** Samples $\widehat{S}$

---

    1. **For $\ell \in [2]$ do:**
        (a) **Set** $N_\ell := \eta_\ell \cdot \sum_{i \in [N]} \mathbb{I}[z_i = \ell]$
        (b) **Set** $G_A := \{i \in [N] : z_i = \ell, x_i = x_A\}$ and $G_B := \{i \in [N] : z_i = \ell, x_i = x_B\}$
        (c) **Initialize** $C = \emptyset$                             *// Corrupted samples*
        (d) Pick any $\min\{N_\ell, |G_B|\}$ items from $G_B$ and add them to $C$
        (e) Pick any $N_\ell - \min\{N_\ell, |R_B|\}$ items from $G_A$ and add them to $C$
        (f) **For $i \in C$ do: Set** $\widehat{z}_i = 3 - \ell$       *// If $z_i = 1$ the $\widehat{z}_i = 2$, and if $z_i = 2$ then $\widehat{z}_i = 1$*
        (g) **For $i \in (G_A \cup G_B) \backslash C$ do: Set** $\widehat{z}_i = \ell$
    2. **return** $\widehat{S} := \{(x_i, y_i, \widehat{z}_i)\}_{i \in [N]}$

---

One can verify that $A$ perturbs exactly $P\ell k \cdot |G_\ell|$ samples with protected attribute $\ell$ to protected attribute $k$. Hence, $A$ is a $P$-restricted Hamming adversary. Further, as $\eta_1 + \eta_2 = \eta$, it also follows that $A$ perturbs at most $\eta$-fraction of samples, and hence, is an $\eta$-Hamming adversary.

---

    [8]In this example, $(1/2) - (\tau/2)$ is the minimum value of $\delta_L$ needed to ensure that $f^\star$, an optimal solution of Program (2), is feasible for [44]'s program with $\eta = 0$.

Fix $\mathcal{X}$ to be any set with at least two distinct points, say $x_A$ and $x_B$. Let $\mathcal{F}$ be any hypothesis class that shatters the set $\{x_A, x_B\} \times [2]$. We define the distribution $\mathcal{D}$ as follows

$$
\Pr_{\mathcal{D}}[X = x, Y = y, Z = z] := \begin{cases} \tau/4 & \text{if } x = x_A, y = 1, z = 1, \\ 1/4 & \text{if } x = x_A, y = 1, z = 2, \\ 1/2 - \tau/4 & \text{if } x = x_B, y = 0, z = 1, \\ 1/4 & \text{if } x = x_B, y = 0, z = 2, \\ 0 & \text{otherwise.} \end{cases} \tag{118}
$$

Note that for a sample $(X, Y, Z) \sim \mathcal{D}$, conditioned on $X$, the value of $Y$ is $\mathbb{I}[X = x_A]$. Thus, the classifier $f^\star(x, z) := \mathbb{I}[X = x_A]$ has 0 predictive error.

We use Lemma D.4 in the proof of Theorem D.3.

**Lemma D.4 (Estimates of statistic on perturbed samples).** *For all $\delta \in (0, 1)$, with probability at least $1 - e^{-\Omega(\delta^2 \cdot N)}$ (over the draw of S), the following bounds hold*

$$
\left| \Pr_{\widehat{D}}[f^\star = 1, Z = 1] - \Pr_{\mathcal{D}}[f^\star = 1, Z = 1] \right| \le \delta, \tag{119}
$$

$$
\left| \Pr_{\widehat{D}}[f^\star = 1, Z = 2] - \Pr_{\mathcal{D}}[f^\star = 1, Z = 2] \right| \le \delta, \tag{120}
$$

$$
\left| \Pr_{\widehat{D}}[Z = 1] - (\Pr_{\mathcal{D}}[Z = 1] + \eta \cdot \Pr_{\mathcal{D}}[Z = 2]) \right| \le \delta, \tag{121}
$$

$$
\left| \Pr_{\widehat{D}}[Z = 2] - \Pr_{\mathcal{D}}[Z = 2] \cdot (1 - \eta) \right| \le \delta. \tag{122}
$$

*Equivalently substituting the statistics on $\mathcal{D}$ in Equations (119) to (122), we get*

$$
\left| \Pr_{\widehat{D}}[f^\star = 1, Z = 1] - \frac{\tau}{4} \right| \le \delta, \tag{123}
$$

$$
\left| \Pr_{\widehat{D}}[f^\star = 1, Z = 2] - \frac{1}{4} \right| \le \delta, \tag{124}
$$

$$
\left| \Pr_{\widehat{D}}[Z = 1] - \frac{1 + \eta}{2} \right| \le \delta, \tag{125}
$$

$$
\left| \Pr_{\widehat{D}}[Z = 2] - \frac{1 - \eta}{2} \right| \le \delta. \tag{126}
$$

The proof of Lemma D.4 follows by analyzing the algorithm of $A$ and using the Chernoff bound. The proof of Lemma D.4 appears at the end of this section.

*Proof of Theorem D.3.* Since the distribution $\mathcal{D}$ is supported on 4 points, namely $\{x_A, x_B\} \times [2]$ (see Equation (118)), we only need to consider hypothesis in the restriction of $\mathcal{F}$ on the set $\{x_A, x_B\} \times [2]$; this restriction has $2^4$ hypothesis.

The first observation is that $\mathbb{I}[X = x_A]$ is an optimal solution for Program (2) and satisfies. $\mathrm{Err}_{\mathcal{D}}(f^\star) = 0$ and $\Omega_{\mathcal{D}}(f^\star) = 1$, where $\Omega$ is the statistical rate fairness metric. This is because, $\mathbb{I}[X = x_A]$ satisfies the constraints of Program (2) and has perfect accuracy.

The second observation is that $f^\star$ is not feasible for [14] and [44]'s programs.

$f^\star$ **is not feasible for [14]'s program.** [14] express their constraints (for statistical rate) in terms of vectors $u(f), w \in [0, 1]^2$ (where $u(f)$ depends on $f \in \mathcal{F}$). They define $u(f)$ and $w$ as follows

$$
u(f) := (P^T)^{-1} \cdot \begin{bmatrix} \Pr_{\widehat{D}}[f = 1, Z = 1] \\ \Pr_{\widehat{D}}[f = 1, Z = 2] \end{bmatrix}, \tag{127}
$$

$$
w := (P^T)^{-1} \cdot \begin{bmatrix} \Pr_{\widehat{D}}[Z = 1] \\ \Pr_{\widehat{D}}[Z = 2] \end{bmatrix}. \tag{128}
$$

[14] impose the following constraint

$$
\frac{\min_{\ell \in [p]} u(f)_\ell / w_\ell}{\max_{\ell \in [p]} u(f)_\ell / w_\ell} \ge \tau. \tag{129}
$$

In our example, $(P^T)^{-1} := \frac{1}{1-\eta} \cdot \begin{bmatrix} 1-\eta & -\eta \\ 0 & 1 \end{bmatrix}$. Set

$$\delta := \frac{\eta \cdot \tau}{64}.$$

Substituting the value of $(P^T)^{-1}$ in Equations (127) and (128), and then using Lemma D.4, we get that with probability at least $1 - e^{-\Omega(\delta^2 \cdot N)}$, $u(f^\star)$ and $w$ satisfy the following bounds

$$\left| u(f^\star)_1 - \left( \frac{\tau}{4} - \frac{\eta}{4(1-\eta)} \right) \right| \leq \frac{\delta}{1-\eta}, \tag{130}$$

$$\left| u(f^\star)_2 - \frac{1}{4(1-\eta)} \right| \leq \delta, \tag{131}$$

$$\left| w_1 - \frac{1}{2} \right| \leq \frac{\delta}{1-\eta}, \tag{132}$$

$$\left| w_2 - \frac{1}{2} \right| \leq \delta. \tag{133}$$

Suppose the Equations (130) to (133) hold. Toward computing the constraint in Equation (129), we compute bounds for $u(f^\star)_1/w_1$ and $u(f^\star)_2/w_2$.

$$\frac{u(f^\star)_1}{w_1} \leq \frac{\frac{\tau}{4} - \frac{\eta}{4(1-\eta)} - \frac{\delta}{1-\eta}}{\frac{1}{2} + \frac{\delta}{1-\eta}} \qquad \text{(Using Equations (130) and (132))}$$

$$\leq \frac{\frac{\tau}{4} - \frac{\eta}{8(1-\eta)}}{\frac{1}{2} \cdot \left(1 - \frac{2\delta}{1-\eta}\right)} \qquad \text{(Using that } \delta \leq \eta/8)$$

$$= \frac{\tau}{2} \cdot \frac{1 - \frac{\eta}{2\tau(1-\eta)}}{1 - \frac{2\delta}{1-\eta}}$$

$$< \frac{\tau}{2}, \qquad \text{(Using that } \delta \leq \eta/(4\tau)) \tag{134}$$

$$\frac{u(f^\star)_2}{w_2} \geq \frac{\frac{1}{4(1-\eta)} - \delta}{\frac{1}{2} + \delta} \qquad \text{(Using Equations (131) and (133))}$$

$$= \frac{1}{2(1-\eta)} \cdot \frac{1 - 4\delta(1-\eta)}{1 + 2\delta}$$

$$\geq \frac{1}{2(1-\eta)} \cdot (1 - 4\delta(1-\eta)) \cdot (1 - 2\delta) \qquad \text{(Using that for all } x \in \mathbb{R}, \frac{1}{1+x} \geq 1 - x.)$$

$$\geq \frac{1}{2(1-\eta)} \cdot (1 - 6\delta) \qquad \text{(Using that } \delta, \eta > 0)$$

$$> \frac{1}{2}. \qquad \text{(Using that } \delta < \eta/6) \tag{135}$$

Substituting Equations (134) and (135) in Equation (129), we get that

$$\frac{\min_{\ell \in [p]} u(f)_\ell/w_\ell}{\max_{\ell \in [p]} u(f)_\ell/w_\ell} \leq \frac{u(f)_1/w_1}{u(f)_2/w_2} \overset{(134),(135)}{<} \tau.$$

Thus, $f^\star$ is not feasible for [14]'s optimization program.

**$f^\star$ is not feasible for [44]'s program.** For any $f \in \mathcal{F}$, [44] impose the constraint

$$\left| \Pr_{\widehat{D}}[f = 1 \mid Z = 1] - \Pr_{\widehat{D}}[f = 1 \mid Z = 2] \right| \leq \delta_L \cdot (1 - \alpha - \beta), \tag{136}$$

where $\alpha, \beta \in [0,1]$ are some function of $\eta_1$ and $\eta_2$. In particular, it holds that if $\eta_1 > 0$ (respectively $\eta_2 > 0$) then $\alpha > 0$ (respectively $\beta > 0$), otherwise $\alpha = 0$ (respectively $\beta = 0$) In our example, $\eta_1 = 0$ and $\eta_2 = \eta > 0$. Thus, $\alpha = 0$ and $\beta > 0$. Recall that $\delta_L := \frac{1}{2} - \frac{\tau}{2}$. To show that $f^\star$ does not

satisfy Equation (136), we bound $\Pr_{\widehat{D}}[f = 1 \mid Z = 1]$ and $\Pr_{\widehat{D}}[f = 1 \mid Z = 2]$.

$$\Pr_{\widehat{D}}[f = 1 \mid Z = 1] \leq \frac{\frac{\tau}{4} + \delta}{\frac{1+\eta}{2} - \delta} \qquad \text{(Using Equations (123) and (125))}$$

$$= \frac{\tau}{2(1+\eta)} \cdot \frac{1 + \frac{4\delta}{\tau}}{1 - \frac{2\delta}{1+\eta}}$$

$$\leq \frac{\tau}{2(1+\eta)} \cdot (1 + \frac{\eta}{16}) \cdot \left(1 + \frac{4\delta}{1+\eta}\right)$$
$$\text{(Using that } \delta := \frac{\eta\tau}{64} \text{ and } \frac{4\delta}{1+\eta} \in [0, 1/2])$$

$$\leq \frac{\tau}{2(1+\eta)} \cdot \left(1 + \frac{\eta}{8}\right)^2 \qquad \text{(Using that } \delta \leq \frac{\eta}{4})$$

$$< \frac{\tau}{2}, \qquad\qquad \text{(Using that } \eta \leq 1) \quad (137)$$

$$\Pr_{\widehat{D}}[f = 1 \mid Z = 2] \geq \frac{\frac{1}{4} - \delta}{\frac{1-\eta}{2} + \delta} \qquad \text{(Using Equations (124) and (126))}$$

$$= \frac{1}{2(1-\eta)} \cdot \frac{1 - 4\delta}{1 + \frac{2\delta}{1+\eta}}$$

$$\geq \frac{1}{2(1-\eta)} \cdot \left(1 - \frac{\eta}{8}\right) \cdot \left(1 - \frac{2\delta}{1+\eta}\right)$$
$$\text{(Using that } 4\delta \leq \eta/8 \text{ and for all } x \in \mathbb{R}, (1+x)^{-1} \geq 1 - x)$$

$$\geq \frac{1}{2(1-\eta)} \cdot \left(1 - \frac{\eta}{8}\right)^2 \qquad \text{(Using that } 2\delta \leq \eta/8)$$

$$\geq \frac{1 - \frac{\eta}{4}}{2(1-\eta)}$$

$$> \frac{1}{2}. \qquad\qquad \text{(Using that } \eta > 0) \quad (138)$$

Thus, combining Equations (137) and (138) and the fact that $\beta > 0$ and $\alpha = 0$, it follows that $f^\star$ is not feasible for Equation (136).

**[44]'s and [14]'s frameworks output a classifier with large error.** Since $f^\star$ is not feasible for [44]'s and [14]'s programs, they must output some other classifier $f_{\text{Alt}} \in \mathcal{F}$. Toward a contradiction, suppose that $\text{Err}_{\mathcal{D}}(f_{\text{Alt}}) < 1/4$. Consider the set $U := \{x_A, x_B\} \times [2] \setminus \{(x_A, 1)\}$. Each point in the $U$ has probability mass at least $1/4$. Thus, if $f_{\text{Alt}}$ has different outcome than $f^\star$ on the set $U$, then $\text{Err}_{\mathcal{D}}(f_{\text{Alt}}) \geq 1/4$. So we must have $f_{\text{Alt}}(r, s) = f^\star(r, s)$ for all $(r, s) \in U$. Because $f_{\text{Alt}}$ is different than $f^\star$, its outcome must differ from $f^\star$ on at least one point in the support of $\mathcal{D}$. The only remaining point is $(x_A, 1)$. Thus, we must have $f_{\text{Alt}}(x_A, 1) = 0$. However, in this case, one can show that $\Pr_{\widehat{D}}[f_{\text{Alt}} = 1, Z = 1] = 0$ and $\Pr_{\widehat{D}}[f_{\text{Alt}} = 1, Z = 1] > 0$. Substituting this in Equations (129) and (136) we get, that $f_{\text{Alt}}$ is not feasible for [14]'s and [44]'s optimization programs. This is a contradiction since we assumed that [14] and [44] output $f_{\text{Alt}}$. This proves Equations (115) and (116).

Finally, Equations (117) and the bound on $\Omega_{\mathcal{D}}(f_{ET})$ follows from Theorem 4.3 because $f^\star$ satisfies Assumption 1 with constant $\lambda = \tau/4$. $\qquad\square$

**Remark D.5** (**Choice of** $P$). *In Theorem D.3, we fix* $P := \begin{bmatrix} 1 & 0 \\ \eta & 1-\eta \end{bmatrix}$. *However, we can show that Theorem D.3 holds for* $P := \begin{bmatrix} 1-\eta_1 & \eta_1 \\ \eta_2 & 1-\eta_2 \end{bmatrix}$ *where* $0 \leq \eta_1 < \eta_2 < \eta$. *The only change is that the high probability guarantee changes from* $1 - e^{-\Omega(\min\{\tau,\eta\}\cdot N)}$ *to* $1 - e^{-\Omega(\min\{\tau,|\eta_2 - \eta_1|\}\cdot N)}$. *Note that the distribution $\mathcal{D}$ does not change.*

**Proof of Lemma D.4.**

*Proof of Lemma D.4.* We give a proof of Equations (119) and (121). The proofs of Equations (120) and (122) follow by replacing $Z = 1$ by $Z = 2$ in the following argument.

Since $A$ only flips samples with feature $x_B$ and $f^\star(x_B, z) = 0$ for all $z \in [2]$, we have that

$$\Pr_{\widehat{D}}[f^\star = 1, Z = 1] = \Pr_D[f^\star = 1, Z = 1] \tag{139}$$

Using the Chernoff bound, it follows that the next inequality holds with probability at least $1 - 2e^{-\frac{16}{3\tau^2} \cdot \delta^2 N}$

$$|\Pr_D[f^\star = 1, Z = 1] - \Pr_{\mathcal{D}}[f^\star = 1, Z = 1]| \le \delta \tag{140}$$

Equation (119) follows from Equations (139) and (140) by using the triangle inequality for the absolute value function. Since $A$ flips $\eta$-fraction of the samples with $Z = 2$ to $Z = 1$, we have that

$$\Pr_{\widehat{D}}[Z = 1] = \Pr_D[Z = 1] + \eta \cdot \Pr_D[Z = 2], \tag{141}$$

$$\Pr_{\widehat{D}}[Z = 2] = \Pr_D[Z = 2] \cdot (1 - \eta). \tag{142}$$

Using the Chernoff bound, it follows that the next inequality holds with probability at least $1 - 4e^{-\frac{16}{3} \cdot \delta^2 N}$

$$|\Pr_D[Z = 1] - \Pr_{\mathcal{D}}[Z = 1]| \le \delta \tag{143}$$

Equation (121) follows from Equations (141) and (143) by using the triangle inequality for the absolute value function. $\qquad\square$

### D.2.3 [59]'s distributionally robust framework can output classifiers with low accuracy

In this section, for any $\eta \in (0, 1/4)$, we give an example where with high probability [59]'s distributionally robust optimization (DRO) framework outputs a classifier $f_{\mathrm{DRO}} \in \mathcal{F}$ whose error is at least $1/2 - \eta/2$ (Theorem D.6) On the same example, an optimal solution of Program (ErrTolerant) has error at most $2\eta$ and additively violates the fairness constraint by at most $O(\eta)$.

[59], in their distributionally robust approach, assume that for each protected group its feature and label distributions in the true data and the perturbed data are a known total variation distance away from each other. Formally, given a distribution $\mathcal{P}$, for each $\ell \in [p]$, let $\mathcal{P}_\ell$ be the distribution of features and labels in group $\ell$ when the data is drawn from $\mathcal{P}$, i.e., $\mathcal{P}_\ell$ is the distribution of $(X, Y) \mid Z = \ell$ when $(X, Y, Z) \sim \mathcal{P}$. Let $\mathcal{D}$ be the true distribution of samples and let $\widehat{D}$ be the (empirical) distribution of perturbed samples. Define a vector $\gamma \in [0, 1]^p$ as follows: For all $\ell \in [p]$

$$\gamma_\ell := \mathrm{TV}(\mathcal{D}_\ell, \widehat{D}_\ell). \tag{144}$$

[59] assume that an upper bound on $\gamma$ is known. One can show that in presence of an $\eta$-Hamming adversary a tight upper bound on $\gamma_\ell$ is $\frac{\eta}{\Pr[Z=\ell]}$; It is achieved by the adversary that, given $N$ samples, perturbs the protected attribute of $\eta \cdot N$ samples with protected attribute $Z = \ell$.

Let $\mathfrak{D}(\gamma)$ be the set of all distributions $\mathcal{P}$ which satisfy that:

$$\text{for all } \ell \in [p], \quad \mathrm{TV}(\widehat{D}_\ell, \mathcal{P}_\ell) \le \gamma_\ell. \tag{145}$$

Then, [59]'s output a classifier $f_{\mathrm{DRO}} \in \mathcal{F}$ which has the highest accuracy on $\widehat{D}$ such that it satisfies their fairness constraints for all distributions $\mathcal{P} \in \mathfrak{D}(\gamma)$; for statistical rate, they solve

$$\min_{f \in \mathcal{F}} \ \mathrm{Err}_{\widehat{D}}(f), \quad \text{s.t.,} \quad \text{for all } \mathcal{P} \in \mathfrak{D}(\gamma), \quad \mathbb{E}_{\mathcal{P}}[f = 1 \mid Z = \ell] = \mathbb{E}_{\mathcal{P}}[f = 1]. \tag{146}$$

**Theorem D.6.** *Suppose that there are two protected groups ($p := 2$) and $\mathcal{X}$ contains at least three distinct points. Then, there is a family of hypothesis classes $\mathcal{F}$ such that for all perturbation rates $\eta \in (0, 1/4)$, there is an adversary $A \in \mathcal{A}(\eta)$ and a distribution $\mathcal{D}$ over $\mathcal{X} \times \{0, 1\} \times [2]$ that satisfies*

1. *Assumption 1 with $\lambda := 1/4$, and*

2. *$\Pr_{\mathcal{D}}[Z = 1] = \Pr_{\mathcal{D}}[Z = 2] = 1/2$,*

*such that, for a draw of $N \in \mathbb{N}$ iid samples $S$ from $\mathcal{D}$, with probability at least $1 - O(e^{-N})$ (over the draw of $S$), it holds that the optimal solution $f_{\mathrm{DRO}} \in \mathcal{F}$ of Program (146) with parameter $\gamma = (2\eta, 2\eta)$[9] and samples $A(S)$ has error*

$$\mathrm{Err}_{\mathcal{D}}(f_{\mathrm{DRO}}) \ge \frac{1}{2} - \eta, \tag{147}$$

---

[9]Following the fact mentioned earlier, for each $\ell \in [2]$, we set $\gamma_\ell := \frac{\eta}{\Pr[Z=\ell]}$.

*and the optimal solution $f_{\mathrm{ET}} \in \mathcal{F}$ of Program (ErrTolerant) with parameters $\eta, \lambda$, and $\tau = 1$ and samples $A(S)$ has error*

$$\mathrm{Err}_{\mathcal{D}}(f_{\mathrm{ET}}) \leq 2\eta. \tag{148}$$

*While $\Omega_{\mathcal{D}}(f_{\mathrm{DRO}}) = 1$ and $\Omega_{\mathcal{D}}(f_{\mathrm{ET}}) \geq 1 - O(\eta)$.*

**Proofs for Section D.2.3.**

**Setting $P$, $\mathcal{D}$, and $A$.** Fix $\mathcal{X}$ to be any set with at least three distinct points, say $x_A, x_B$ and $x_C$. Let $\mathcal{F}$ be any hypothesis class that shatters the set $\{x_A, x_B, x_C\} \times [2] \subseteq \mathcal{X} \times [p]$. Define $\mathcal{D}$ as the unique distribution such that for a draw $(X, Y, Z) \sim \mathcal{P}$, $Y$ takes the value $\mathbb{I}[X \neq x_C]$ and that has the following marginal distribution:

|   | $x_A$ | $x_B$ | $x_C$ |
|---|-------|-------|-------|
| 1 | $\eta$ | $1/4$ | $1/4 - \eta$ |
| 2 | $0$ | $1/4$ | $1/4$ |

Where for each $(r, s) \in \{x_A, x_B, x_C\} \times [2]$, the corresponding cell denotes $\mathrm{Pr}_{(X,Y,Z) \sim \mathcal{P}}[(X, Z) = (r, s)]$. Because $Y$ takes the value $\mathbb{I}[X \neq x_C]$, the classifier $f^\star(x, z) := \mathbb{I}[X \neq x_C]$ has 0 predictive error. We fix $A \in \mathcal{A}(\eta)$ to be the adversary that does not perturb any samples. (This suffices to prove the claim in Theorem D.6, but one can also other adversaries in $\mathcal{A}(\eta)$.)

**Supporting lemmas.** We use the following lemmas in the proof of Theorem D.6.

**Lemma D.7.** *The classifier $f^\star(X, Z) = \mathbb{I}[X = x_B]$ is an optimal solution for Program (2) with $\tau = 1$: $\mathrm{Err}_{\mathcal{D}}(f^\star) = \eta$ and $\Omega_{\mathcal{D}}(f^\star) = 1$, where $\Omega$ is the statistical rate fairness metric.*

*Proof.* One can verify that the classifier with the perfect accuracy, $f_{\mathrm{OPT}} := \mathbb{I}[X \neq x_C]$, has a statistical rate strictly smaller than 1. So it is not feasible for Program (2) for $\tau = 1$. Any feasible classifier $f \in \mathcal{F}$ must differ from $f_{\mathrm{OPT}}$ on some point in the support of $\mathcal{D}$. Since (by construction) all points in the support of $\mathcal{D}$ have a probability mass at least $\eta$, it follows that any $f \in \mathcal{F}$ feasible Program (2) must have an error at least $\eta$. Now the result follows since $f^\star := \mathbb{I}[X = x_A]$ is feasible for Program (2) and has the optimal error, $\mathrm{Err}_{\mathcal{D}}(f^\star) = \eta$. $\qquad\square$

**Lemma D.8.** *Consider the distribution $\mathcal{P}$, such that, for a draw $(X, Y, Z) \sim \mathcal{P}$, $Y := \mathbb{I}[X \neq x_C]$ and that has the following marginal distribution:*

|   | $x_A$ | $x_B$ | $x_C$ |
|---|-------|-------|-------|
| 1 | $\eta$ | $1/4 - \eta$ | $1/4$ |
| 2 | $0$ | $1/4$ | $1/4$ |

*Where for each $(r, s) \in \{x_A, x_B, x_C\} \times [2]$, the corresponding cell denotes $\mathrm{Pr}_{(X,Y,Z) \sim \mathcal{P}}[(X, Z) = (r, s)]$. Given a draw of $N$ iid samples from $\mathcal{D}$, with probability at least $1 - e^{-\Omega(N)}$, it holds that $\mathcal{P} \in \mathfrak{D}(\gamma)$ and $\mathcal{D} \in \mathfrak{D}(\gamma)$.*

*Proof.* Given a sufficient number of iid samples $S$ from $\mathcal{D}$, one can show that with high probability, the empirical distribution $D$ of $S$ satisfies that: $\mathrm{TV}(D_1, \mathcal{D}_1) \leq \eta$ and $\mathrm{TV}(D_2, \mathcal{D}_2) \leq \eta$. Since $\gamma := (2\eta, 2\eta)$, this implies that with high probability $\mathcal{D} \in \mathfrak{D}(\gamma)$. Further, the construction in Lemma D.8 ensures that $\mathrm{TV}(\mathcal{P}_1, \mathcal{D}_1) = \eta$ and $\mathrm{TV}(\mathcal{P}_2, \mathcal{D}_2) = \eta$. By using the triangle inequality of the total variation distance, it follows that with high probability, $\mathrm{TV}(\mathcal{P}_1, D_1) \leq 2\eta$ and $\mathrm{TV}(\mathcal{P}_2, D_2) \leq 2\eta$. Since $\gamma := (2\eta, 2\eta)$, it follows that with high probability $\mathcal{P} \in \mathfrak{D}(\gamma)$. $\qquad\square$

**Lemma D.9.** *Any $f \in \mathcal{F}$ that satisfies the following equalities*

$$\mathbb{E}_{\mathcal{D}}[f = 1 \mid Z = \ell] = \mathbb{E}_{\mathcal{D}}[f = 1], \tag{149}$$

$$\mathbb{E}_{\mathcal{P}}[f = 1 \mid Z = \ell] = \mathbb{E}_{\mathcal{P}}[f = 1], \tag{150}$$

*where $\mathcal{D}$ is true distribution and $\mathcal{P}$ is the distribution from Lemma D.8 must have an error*

$$\mathrm{Err}_{\mathcal{D}}(f) \geq \frac{1}{2} - \eta.$$

Using Lemmas D.7 to D.9, Theorem D.6 follows as a corollary.

*Proof of Theorem D.6.* From Lemma D.8 know that with probability at least $1 - e^{-\Omega(N)}$, $\mathcal{P} \in \mathfrak{D}$ and $\mathcal{D} \in \mathfrak{D}(\gamma)$. Suppose that this event happens. Assume that $f_{\mathrm{DRO}} \in \mathcal{F}$ is the optimal solution of Program (146). Since $f_{\mathrm{DRO}}$ is feasible for Program (146), it must satisfy that

$$\mathbb{E}_{\mathcal{D}}[f_{\mathrm{DRO}} = 1 \mid Z = \ell] = \mathbb{E}_{\mathcal{P}}[f_{\mathrm{DRO}} = 1],$$
$$\mathbb{E}_{\mathcal{P}}[f_{\mathrm{DRO}} = 1 \mid Z = \ell] = \mathbb{E}_{\mathcal{P}}[f_{\mathrm{DRO}} = 1],$$

Then Lemma D.9 tells us that $\mathrm{Err}_{\mathcal{D}}(f_{\mathrm{DRO}}) \geq \frac{1}{2} - \eta$.

Finally, one can verify that when statistical rate is the fairness metric, $f^\star$ (from Lemma D.7) satisfies Assumption 1 with $\lambda = \frac{1}{2}$. Thus, Equation (148) and the inequality $\Omega_{\mathcal{D}}(f_{\mathrm{ET}}) \geq 1 - O(\eta)$ follow from Theorem 4.3. $\qquad\square$

*Proof of Lemma D.9.* Since that both $\mathcal{D}$ and $\mathcal{P}$ are supported on subsets of $\{x_A, x_B, x_C\} \times [2]$, it suffices to consider the restriction of $\mathcal{F}$ on this domain. Consider any classifier $f \in \mathcal{F}$ and define the following variables

$$f_{A1} := f(x_A, 1), \quad f_{B1} := f(x_B, 1), \quad f_{C1} := f(x_C, 1)$$
$$f_{A2} := f(x_A, 2), \quad f_{B2} := f(x_B, 2), \quad f_{C2} := f(x_C, 2),$$

denoting the predictions of $f$ on $\{x_A, x_B, x_C\} \times [2]$. Since $f$ satisfies Equation (149), we must have

$$\begin{aligned}
2 \cdot \left( \eta f_{A1} + \frac{1}{4} f_{A2} + \left( \frac{1}{4} - \eta \right) f_{A3} \right) &= \mathbb{E}_{\mathcal{D}}[f = 1 \mid Z = \ell] \\
&= \mathbb{E}_{\mathcal{D}}[f = 1] \\
&= \eta f_{A1} + \frac{1}{4} f_{A2} + \left( \frac{1}{4} - \eta \right) f_{A3} + \frac{1}{4} f_{B2} + \frac{1}{4} f_{B3}. \quad (151)
\end{aligned}$$

Similarly, since $f$ satisfies Equation (150), we must have

$$\begin{aligned}
2 \cdot \left( \eta f_{A1} + \left( \frac{1}{4} - \eta \right) f_{A2} + \frac{1}{4} f_{A3} \right) &= \mathbb{E}_{\mathcal{P}}[f = 1 \mid Z = \ell] \\
&= \mathbb{E}_{\mathcal{P}}[f = 1] \\
&= \eta f_{A1} + \left( \frac{1}{4} - \eta \right) f_{A2} + \frac{1}{4} f_{A3} + \frac{1}{4} f_{B2} + \frac{1}{4} f_{B3}. \quad (152)
\end{aligned}$$

Combining Equations (151) and (152), we get

$$\begin{aligned}
\eta f_{A1} + \left( \frac{1}{4} - \eta \right) f_{A2} + \frac{1}{4} f_{A3} &= \frac{1}{4} f_{B2} + \frac{1}{4} f_{B3} \\
&= \eta f_{A1} + \frac{1}{4} f_{A2} + \left( \frac{1}{4} - \eta \right) f_{A3}. \quad (153)
\end{aligned}$$

On canceling the like terms in the LHS and RHS, and using that $\eta > 0$, we get

$$f_{A2} = f_{A3}.$$

We consider two cases.

**(Case A)** $f_{A2} = f_{A3} = 1$**:** Substituting $f_{A2} = f_{A3} = 1$ in Equation (153), we get

$$\eta f_{A1} + \frac{1}{2} - \eta = \frac{1}{4} f_{B2} + \frac{1}{4} f_{B3}.$$

Here, the RHS can only take values $\{0, 1/4, 1/2\}$ and the LHS can only take values $\{1/2 - \eta, 1/2\}$. Thus, the unique solution is $f_{A1} = f_{B2} = f_{B3} = 1$. One can verify that the unique resulting classifier has error $\mathrm{Err}_{\mathcal{D}}(f) = 1/2 - \eta$.

**(Case B)** $f_{A2} = f_{A3} = 0$**:** Substituting $f_{A2} = f_{A3} = 0$ in Equation (153), we get

$$\eta f_{A1} = \frac{1}{4} f_{B2} + \frac{1}{4} f_{B3}.$$

Here, the RHS can only take values $\{0, {}^1/_4, {}^1/_2\}$ and the LHS can only take values $\{0, \eta\}$. Thus, the unique solution is $f_{A1} = f_{B2} = f_{B3} = 0$. One can verify that the unique resulting classifier has error $\mathrm{Err}_{\mathcal{D}}(f) = {}^1/_2$.

Thus, any $f \in \mathcal{F}$ satisfying Equations (149) and (150), must have an error at least ${}^1/_2 - \eta$. $\qquad\square$

# E  Implementation details and additional empirical results

In this section, we give an implementation of our optimization framework using the logistic loss function (Section E.1.1), list the all hyper-parameters used for our approach (Section E.1.1) and baselines (Section E.1.4), and present some additional empirical results (Section E.2).

**Code.**  The code for all the simulations is available at `https://github.com/AnayMehrotra/Fair-classification-with-adversarial-perturbations`.

## E.1  Implementation details

### E.1.1  Implementation of our framework using logistic regression

As an illustration, we implement our optimization framework (Program (ErrTolerant+)) using the logistic regression framework. For some $\theta \in \mathbb{R}^d$, let $f_\theta$ be the linear-classifier that given an input $x \in \mathbb{R}^d$ predicts $\mathbb{I}\left[1/\left(1 + e^{-\langle x, \theta \rangle}\right) \geq 0.5\right]$ (or equivalently that predicts $\mathbb{I}[\langle x, \theta \rangle \geq 0]$). Then, the logistic regression framework considers the following hypothesis class: $\mathcal{F}_{\mathrm{LR}} := \left\{f_\theta \mid \theta \in \mathbb{R}^d\right\}$; see [54] for more details.

Several baselines (e.g., **CHKV** and **LMZV**) do not use protected attributes for prediction. For a fair comparison, in this implementation we do not use protected attributes for prediction. Let the domain of the features $\mathcal{X}$ satisfy $\mathcal{X} \subseteq \mathbb{R}^t$ for some $t \in \mathbb{N}$. In this case, we have $d := t$, where $d$ is the dimension of $\theta$ (which parameterizes the hypothesis class $\mathcal{F}_{\mathrm{LR}}$). (To use protected attributes for prediction, one can set $d := t + 1$ and append the protected attribute to the features.)

Recall that Program (ErrTolerant+) takes the following values as input: the perturbation rate $\eta \in [0, 1]$, fairness threshold $\tau \in [0, 1]$, and for each $\ell \in [p]$, $\lambda_\ell \in [0, 1]$ and $\gamma_\ell \in [0, 1]$. Given these, we solve the following problem and initialize $s$ to be its solution

$$\min_{\eta_1, \eta_2, \ldots, \eta_p \geq 0} \min_{\ell, k \in [p]} \frac{1 - \eta_\ell/\lambda_\ell}{1 + (\eta_k - \eta_\ell)/\gamma_\ell} \cdot \frac{1 + (\eta_\ell - \eta_k)/\gamma_k}{1 + \eta_\ell/\lambda_k}, \quad \text{s.t.,} \quad \sum_{\ell \in [p]} \eta_\ell \leq \eta + \Delta. \quad (154)$$

We solve Program (154) once to initialize $s$, then the same value $s$ is used for all runs of Program (ErrTolerant+). Let $\mathcal{E}(f)$ and $\mathcal{E}'(f)$ denote the events defining the relevant linear-fractional fairness metric (Definition 3.1). Let $S := \{(x_i, y_i, z_i)\}_{i \in [N]}$ be the perturbed samples. Then we solve the following constrained optimization program

$$\min_{\theta \in \mathbb{R}^d} \quad \frac{1}{N} \cdot \sum_{i=1}^N y_i \cdot \log f_\theta(x_i) + (1 - y_i) \cdot \log\left(1 - f_\theta(x_i)\right) \qquad \text{(ErrTolerant+)} \quad (155)$$

$$\text{s.t.,} \qquad \min_{\ell, k \in [p]} \frac{q_\ell(f)}{q_k(f)} \geq \tau \cdot s$$

$$\forall \, \ell \in [p], \quad \frac{1}{N} \sum_{i \in [N]} \mathbb{I}[\mathcal{E}(f(x_i)), \mathcal{E}'(f(x_i)), Z = \ell] \geq \lambda_\ell - \eta - \Delta,$$

where for each $\ell \in [p]$,

$$q_\ell(f) := \frac{\sum_{i \in [N]: z_i = \ell} \mathbb{I}[\mathcal{E}(f(x_i)), \mathcal{E}'(f(x_i))]}{\sum_{i \in [N]: z_i = \ell} \mathbb{I}[\mathcal{E}'(f(x_i))]}.$$

In particular, for statistical rate, for each $i \in [N]$,

$$\mathcal{E}(f(x_i)) := \mathbb{I}[f(x_i) = 1] \text{ and } \mathcal{E}'(f(x_i)) := 1,$$

and for false-positives rate, for each $i \in [N]$,

$$\mathcal{E}(f(x_i)) := \mathbb{I}[f(x_i) = 1] \text{ and } \mathcal{E}'(f(x_i)) := \mathbb{I}[y_i = 0].$$

By substituting the appropriate $\mathcal{E}$ and $\mathcal{E}'$, one can extend this implementation to any linear-fractional fairness metric (Definition 3.1).

**Hyper-parameters.** As a heuristic, given $\mathcal{E}$ and $\mathcal{E}'$, in our simulations for each $\ell \in [p]$, we set $\gamma_\ell = \lambda_\ell := \Pr_{(X,Y,\widehat{Z}) \sim \widehat{D}}[\mathcal{E}'(Y), \widehat{Z} = \ell]$, where $\widehat{D}$ is the empirical distribution of $\widehat{S}$ and $Y$ is the label in perturbed data $\widehat{S}$. We find that these estimates suffice, and expect that a more refined approach would only improve the performance of **Err-Tol**. For all simulations, we set $\Delta := 10^{-2}$.

### E.1.2 SLSQP parameters

For simplicity, we do not implement the algorithm mentioned in Section C, and instead use existing optimization packages in our implementation. We solve both Program (ErrTolerant+) and Program (7) using the SLSQP solver [41] in SciPy [57]. For each optimization problem, we run the solver for 1000 iterations with parameters `ftol` $= 10^{-4}$ and `eps` $= 10^{-4}$, starting at a point chosen uniformly at random. If the solver fails to find a feasible solution, we rerun the solver for up to 10 iterations. If it does not find a feasible solution after 10 iterations, we return the infeasible point reached.

### E.1.3 Implementation details of the adversaries

Across our simulations we consider three $\eta$-Hamming adversaries (which we call $A_{\mathrm{TN}}$, $A_{\mathrm{FN}}$, and $A_{\mathrm{FP}}$). Each adversary has access to the true samples $S$, the fairness metric $\Omega$, and the desired fairness threshold $\tau$. Using these, the adversary computes the "optimal fair classifier" $f^\star$ that has the highest accuracy (on $S$) subject to satisfying $\Omega(f, S) \geq \tau$. $f^\star$ is an optimal solution of Program (2); note that Program (2) is a special case of Program (ErrTolerant) (with $\lambda, \eta, \Delta \to 0$). In practice, we compute $f^\star$ by solving Program (2) on the unperturbed data $S$, using the SLSQP solver in the SciPy package to heuristically solve Program (2) (with the same parameters as described in Section E.1.2).

After computing $f^\star$, $A_{\mathrm{TN}}$ considers the set of all true negatives of $f^\star$ that have protected attribute $Z = 1$, selects the $\eta \cdot |S|$ samples that are furthest from the decision boundary of $f^\star$, and perturbs their protected attribute to $\widehat{Z} = 2$. $A_{\mathrm{FN}}$ and $A_{\mathrm{FP}}$ are identical, except that they consider the set of all false negatives and false positives of $f^\star$ respectively.

### E.1.4 Baseline parameters and implementation

**LMZV.** We use the implementation of the **LMZV** at `https://github.com/AIasd/noise _fairlearn` provided by [44]; where the base classifier is by [1]. **LMZV** takes group-specific perturbation-rates, for each $\ell \in [p]$, $\eta_\ell := \Pr_D[\widehat{Z} \neq Z \mid Z = \ell]$, as input, and controls for additive statistical rate. The desired level of fairness is controlled by $\delta_L \in [0, 1]$, where smaller $\delta_L$ corresponds to higher fairness. We refer the reader to [44] for a description of these parameters. In our simulations, we vary $\delta_L$ over $\{10^{-2}, 4 \cdot 10^{-2}, 10^{-1}\}$. We fix all other hyper-parameters to the ones suggested by the authors for COMPAS.

**AKM.** We use the implementation of **AKM** at `https://github.com/matthklein/equali zed_odds_under_perturbation` provided by [6]; **AKM** is the equalized-odds postprocessing method of [32]. It takes the unconstrained optimal classifier (**Uncons**) as input, and post-processes its outputs to control for equalized-odds constraints.

**WGN+DRO.** This is the distributionally robust framework of [59]; we use the implementation of **WGN+DRO** at `https://github.com/wenshuoguo/robust-fairness-code` provided by [59], which controls for additive false-positive rate. It takes true and perturbed protected attributes as input and computes the required bound on the total variation distance. We use the following learning rates for **WGN+DRO:** $\eta_\theta \in \{10^{-3}, 10^{-2}, 10^{-1}\}$, $\eta_\lambda \in \{1/4, 1/2, 1, 2\}$, and $\eta_{\widetilde{p}_j} \in \{10^{-3}, 10^{-2}, 10^{-1}\}$; these are the same as the learning rates used by the authors (see [59, Table 2]). We refer the reader to [59] for the details of the parameters. The implementation runs **WGN+DRO** for all combinations of learning rates and outputs the classifier that has the best training objective and satisfies their constraints.

**WGN+SW.** This is the "soft-weights" framework of [59]; we use the implementation of **WGN+SW** at `https://github.com/wenshuoguo/robust-fairness-code` provided by [59]. It takes true and perturbed protected attributes as input and controls for additive false-positive rate. We use the following learning rates for **WGN+SW**: $\eta_\theta \in \left\{10^{-3}, 10^{-2}, 10^{-1}\right\}$, $\eta_\lambda \in \{1/4, 1/2, 1, 2\}$, and $\eta_w \in \left\{10^{-3}, 10^{-2}, 10^{-1}\right\}$; these are the same as the learning rates used by the authors (see [59, Table 2]). See [59] for a discussion on the parameters. Their implementation runs **WGN+SW** for all combinations of learning rates and outputs the classifier that has the best training objective and satisfies their constraints.

**CHKV and CHKV-FPR.** We use the implementation of the [14]'s framework at `https://github.com/vijaykeswani/Noisy-Fair-Classification` provided by [14]. We use the implementations for statistical rate and false-positive rate, which we refer to these as **CHKV** and **CHKV-FPR** respectively. Both implementations take group specific perturbation-rates, for each $\ell \in [p]$, $\eta_\ell := \Pr_D[\widehat{Z} \neq Z \mid Z = \ell]$, as input. The desired level of fairness is controlled by $\tau \in [0, 1]$, where a larger $\tau$ corresponds to higher fairness. In our simulations, we vary $\tau$ over $\{0.7, 0.8, 0.9, 0.95, 1.0\}$; other hyper-parameters were the same as those suggested by the authors for COMPAS.

**KL.** This is the framework of [40] which controls for true-positive rate. It takes the perturbation rate $\eta$ and for each $\ell \in [p]$, the probability $p_{1\ell} := \Pr_D[Z = \ell, Y = 1]$ as input; where $D$ is the empirical distribution of $S$. [40] do not provide an implementation of **KL**. We implement **KL** using the logistic loss function. In particular, we solve the following optimization problem

$$\min_{\theta \in \mathbb{R}^d} \quad \frac{1}{N} \cdot \sum_{i=1}^{N} y_i \cdot \log f_\theta(x_i) + (1 - y_i) \cdot \log\left(1 - f_\theta(x_i)\right) \tag{156}$$

$$\text{s.t.,} \qquad \forall \, \ell \in [p], \qquad \frac{\sum_{i \in [N]} \mathbb{I}[f(x_i) = 0, y_i = 1, \widehat{z}_i = \ell]}{\sum_{i \in [N]} \mathbb{I}[y_i = 1, \widehat{z}_i = \ell]} \leq \frac{6\eta}{\min_{\ell \in [p]} p_{1\ell} + 3\eta}$$

We solve Problem (156) using the standard implementation of the SLSQP solver in SciPy [57]; with the same parameters Section E.1.2.

### E.1.5 Computational resources used

All simulations were run on a `t3a.2xlarge` instance, with 8 vCPUs and 32 Gb RAM, on Amazon's Elastic Compute Cloud (EC2).

### E.2 Visualization of synthetic data

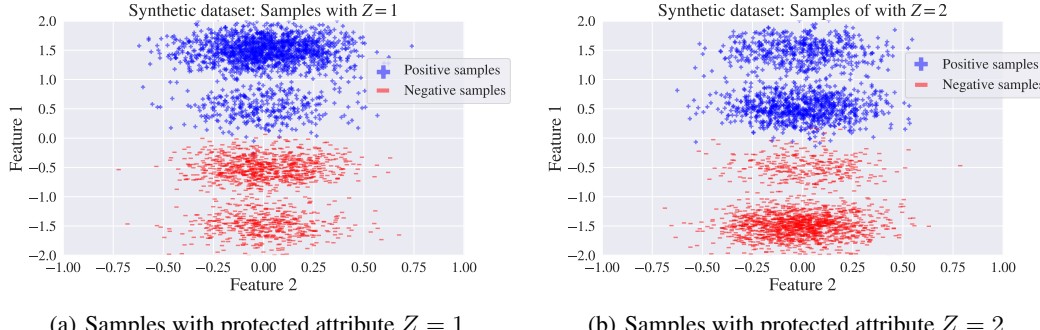

(a) Samples with protected attribute $Z = 1$         (b) Samples with protected attribute $Z = 2$

Figure 4: *Samples from the synthetic data (Section 5)*. Figure 4(a) shows the samples with protected attribute $Z = 1$ and Figure 4(b) shows the samples with protected attribute $Z = 2$. We consider synthetic data with 1,000 samples with two equally-sized protected groups; each sample has a binary protected attribute, two continuous features $x_1, x_2 \in \mathbb{R}$, and a binary label. Conditioned on the protected attribute, $(x_1, x_2)$ are independent draws from a mixture of four 2D Gaussians. The distribution of the labels and features is such that 1) the protected group $Z = 1$ has a higher likelihood of a positive label than the protected group $Z = 2$, and 2) **Uncons** has a near-perfect accuracy ($> 99\%$) and a statistical rate of $0.8$ on $S$.

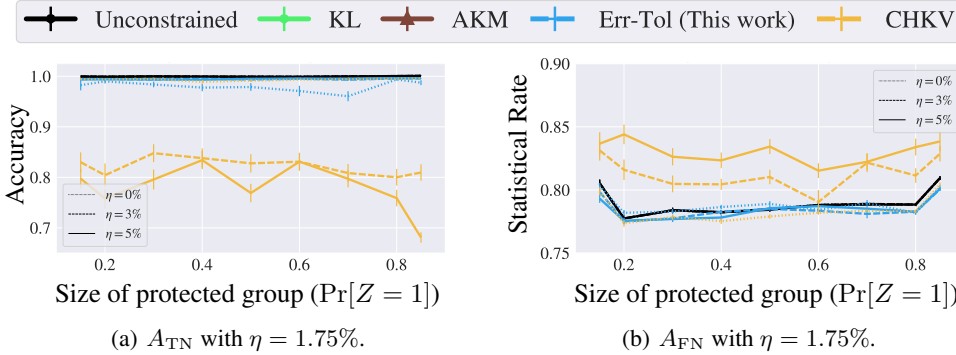

(a) $A_{\text{TN}}$ with $\eta = 1.75\%$.

(b) $A_{\text{FN}}$ with $\eta = 1.75\%$.

Figure 5: *Simulation varying the size of the protected groups on the synthetic data (see Section E.3.1):* We vary the perturbation rate $\eta$ over $\{0, 0.03, 0.05\}$ and the size $\alpha$ of the protected group denoted by $Z = 1$ from 15% to 85% (see Section E.3.1). For each pair of $\alpha$ and $\eta$, we run **CHKV** and **Err-Tol** with $\tau = 0.8$ and report the average accuracy and statistical rate plots Figures 5(a) and 5(b) (respectively). (In the plot, the color of the line identifies the algorithm and its style identifies the value of $\eta$). The $y$-axis depicts accuracy or statistical rate and the $x$-axis depicts $\alpha$; both the accuracy and the statistical rate are computed over the unperturbed test set (we refer the reader to Section 5 for further details). We observe that prior approaches can fail to satisfy their guarantees under the $\eta$-Hamming model, and their deviation from the accuracy guarantee increases as the size of one protected group decreases (i.e., when $\alpha$ approaches 0.85). Error bars represent the standard error of the mean over 100 iterations.

## E.3 Additional empirical results

### E.3.1 Simulation varying the size of the protected groups on synthetic data

In this simulation, we study the effect of varying the relative sizes of the protected groups in the synthetic data on the results of the simulation in Section 5. We vary the size of one protected group $\alpha$ from 0.15 to 0.85 (and the size of the other from 0.85 to 0.15.[10] Recall that the synthetic data in Section 5 is generated by sampling 500 samples with $Z = 1$ from a distribution $\mathcal{D}_1$ and sampling 500 samples with $Z = 2$ from different distribution $\mathcal{D}_2$ (where both $\mathcal{D}_1$ and $\mathcal{D}_2$ are mixtures of four 2D Gaussians). (See Figure 4 for a plot of samples from $\mathcal{D}_1$ and $\mathcal{D}_2$). To vary the size of the protected groups, given an $\alpha \in [0, 1]$, we draw $1000 \cdot \alpha$ samples with $Z = 1$ from $\mathcal{D}_1$ and $1000 \cdot (1 - \alpha)$ samples with $Z = 2$ from $\mathcal{D}_2$. For each value of $\alpha$, we rerun the simulation from Section 5 on the resulting data.

**Results.** We observe that varying $\alpha$ from from 15% to 70% does not change the accuracy and statistical rate of the algorithms significantly (the accuracy changed by <5% and the statistical rate changed by <1%). However, as $\alpha$ approaches 85%, we observe that **CHKV's** accuracy reduced by 15% (to ≈0.68) and its statistical rate increased by 4% (to ≈0.86). In contrast, **Err-Tol** continued to have high accuracy (>0.99) and statistical rate (≥0.77), without large changes in either (its accuracy changed by <1% and statistical rate changed by <2%). Overall, we observe that prior approaches can fail to satisfy their guarantees under the $\eta$-Hamming model, and their deviation from the accuracy guarantee increases as the size of one protected group decreases (i.e., when $\alpha$ approaches 0.85).

### E.3.2 Simulations with stochastic perturbations on the COMPAS data

In this simulation, we evaluate our framework under stochastic perturbations on the COMPAS data, and show that it has a similar statistical rate and accuracy trade-off as approaches tailored for stochastic perturbations (e.g., [44] and [14]). Concretely, we consider a binary protected attribute and the perturbation model studied by [14]: Suppose we have a single binary protected attribute (i.e., $p = 2$). Given values $\eta_1, \eta_2 \in [0, 1]$, the protected attributes of each item with protected attribute $Z = 1$ change to $\widehat{Z} = 2$ with probability $\eta_1$ (independently), and similarly, the protected attributes of each item with protected attribute $Z = 2$ change to $\widehat{Z} = 1$ with probability $\eta_2$ (independently). We consider the COMPAS data as preprocessed by [9], and consider gender (coded as binary) as the protected attribute. We consider four values of $(\eta_1, \eta_2)$: $(0\%, 0\%)$, $(0\%, 3.5\%)$, $(3.5\%, 0\%)$, and $(3.5\%, 3.5\%)$.

---

[10]15% and 85% are (roughly) the smallest and largest group sizes for which $f^\star$ has a sufficient number of true negatives to use $A_{\text{TN}}$ with $\eta = 5\%$.

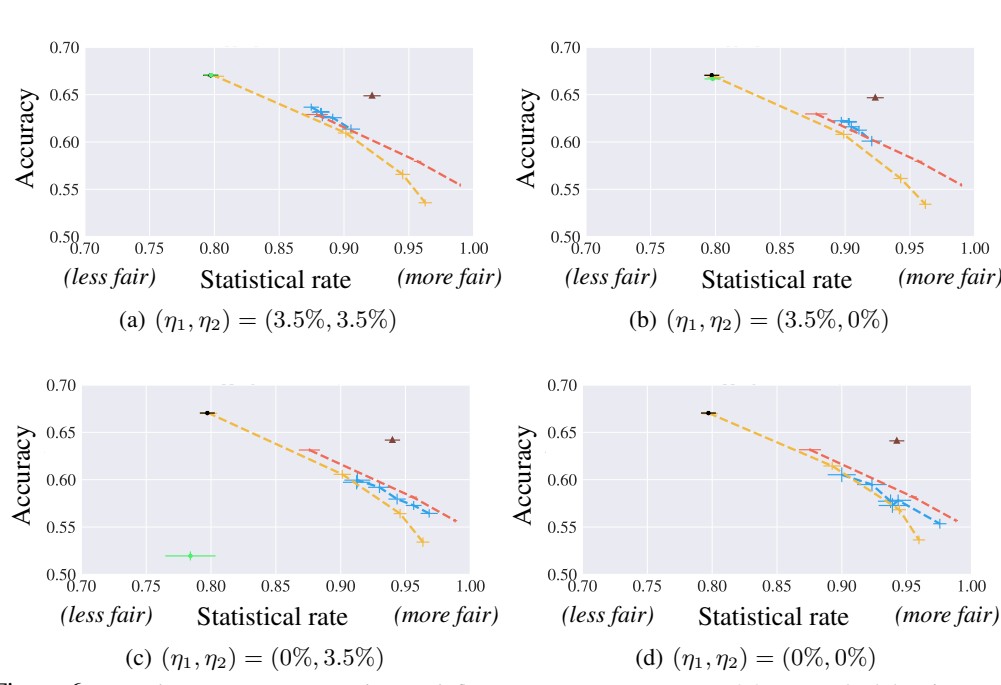

Figure 6: *Simulations on COMPAS data with flipping noise (see Section E.3.2):* Perturbed data is generated using the flipping noise model of [14, Definition 2.3] with $\eta_1$ and $\eta_2$ mentioned with each subfigure. All algorithms are run the perturbed data varying the fairness parameters ($\tau \in [0.7, 1]$ and $\delta_L \in [0, 0.1]$). The $y$-axis depicts the accuracy and the $x$-axis depicts statistical rate; both values are computed over the unperturbed test set. We observe that in each case, our approach, **Err-Tol**, has a similar fairness-accuracy trade-off as approaches tailored for flipping noise [14, 44] Error bars represent the standard error of the mean over 100 iterations.

**Results.** The accuracy and statistical rate of **Err-Tol** and baselines for $\tau \in [0.7, 1]$ and $\delta_L \in [0, 0.1]$, averaged over 100 iterations, are reported in Figure 6. For all settings of $\eta_1$ and $\eta_2$, **Err-Tol** attains a better statistical rate than the unconstrained classifier (**Uncons**) for a small trade-off in accuracy. Further, **Err-Tol** has a similar statistical rate and accuracy trade-off as **CHKV** and **LMZV**. In all cases, **AKM** a better statistical rate and accuracy trade-off than all other approaches. Understanding why **AKM** has a better trade-off than other approaches with flipping noise requires further study. But it is likely because **AKM** does not need to make pessimistic assumptions on the data (as it does not account for perturbations) and outputs a classifier from a richer hypothesis class compared to other approaches. However, we recall that, when the perturbations are adversarial, **Err-Tol** has a better accuracy and statistical rate trade-off than **AKM** (see Figure 1).

### E.3.3 Simulations with adversarial perturbations and false-positive rate on COMPAS data

In this simulation, we evaluate our framework for false-positive rate fairness metric with adversarial perturbations against state-of-the-art fair classification frameworks for false-positive rate under stochastic perturbations: **WGN+SW** [59], **WGN+DRO** [59], and **CHKV-FPR** [14]. We also compare against **KL** [40], which controls for true-positive rate (TPR) in the presence of a Malicious adversary, and **AKM** [6] that is the post-processing method of [32] and controls for equalized-odds fairness constraints.

Similar to the simulation with statistical rate fairness metric (see Section 5), **Err-Tol** is given the perturbation rate $\eta$. To advantage the baselines in our comparison, we provide them with more information as needed by their approaches:

1. **WGN+SW** is given both the true and perturbed protected attributes as input; it internally generates the auxiliary data needed by [59]'s "soft-weights" approach.
2. **WGN+DRO** is given both the true and perturbed protected attributes as input; it internally computes the total variation distances needed by [59]'s distributionally robust approach.
3. **CHKV-FPR** is given group-specific perturbation rates: $\forall \ell \in [p], \eta_\ell := \Pr_D[\widehat{Z} \neq Z \mid Z = \ell]$.
4. **KL** is given $\eta$ and for each $\ell \in [p]$, the probability $\Pr_D[Z = \ell, Y = 1]$; where $D$ is the empirical distribution of $S$.

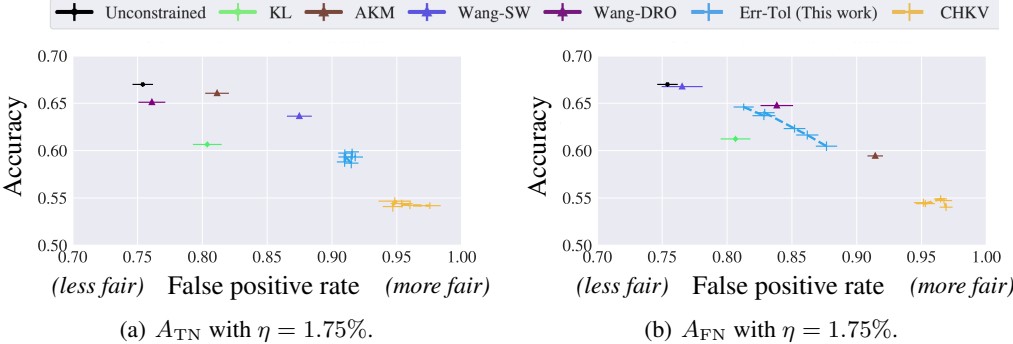

(a) $A_{\text{TN}}$ with $\eta = 1.75\%$.  (b) $A_{\text{FN}}$ with $\eta = 1.75\%$.

Figure 7: *Simulations on COMPAS with false-positive rate (see Section E.3.3):* Perturbed data is generated using adversaries $A_{\text{TN}}$ (a) and $A_{\text{FN}}$ as described in Section E with $\eta = 1.75\%$. All algorithms are run on the perturbed data by varying the fairness parameters ($\tau \in [0.7, 1]$). The $y$-axis depicts accuracy and the $x$-axis depicts false-positive rate computed over the unperturbed test set. We observe that for all adversaries our approach, **Err-Tol**, attains a better fairness than the unconstrained classifier (**Uncons**) with a natural trade-off in accuracy. Further, **Err-Tol** achieves a higher fairness than each baseline except **CHKV** on at least one of (a) or (b). **CHKV** attains a higher fairness but has a low accuracy ($\sim 55\%$) (because it outputs the always-positive classifier). Error bars represent the standard error of the mean over 100 iterations.

**Err-Tol** implements Program (ErrTolerant+) which requires estimates of $\lambda_\ell$ and $\gamma_\ell$ for all $\ell \in [p]$. As a heuristic, we set $\gamma_\ell = \lambda_\ell := \Pr_{\widehat{D}}[Y = 1, Z = \ell]$, where $\widehat{D}$ is the empirical distribution of $\widehat{S}$. More generally, for a general linear-fractional fairness metric, given by $\mathcal{E}$ and $\mathcal{E}'$, the heuristic is to set $\gamma_\ell = \lambda_\ell := \Pr_{\widehat{D}}[\mathcal{E}'(Y), Z = \ell]$, where $\widehat{D}$ is the empirical distribution of $\widehat{S}$ and $Y$ is the label in perturbed data $\widehat{S}$.

**Adversaries.** We consider the same adversaries as in Section 5 (which we call $A_{\text{TN}}$ and $A_{\text{FN}}$). We consider a perturbation rate of $\eta = 1.75\%$. Again, $1.75\%$ is roughly the smallest value for $\eta$ necessary to ensure that the optimal fair classifier $f^\star$ for $\tau = 0.9$ (on $S$) has a false-positive rate less than the false-positive rate of **Uncons** on the perturbed data. This "threshold" perturbation rate is smaller for false-positive rate than for statistical rate because to the number of false positives of a classifier $f$ is smaller than the number of positive predictions of $f$; hence, an adversary perturbing the same number of samples, perturbs a larger fraction of false positives than the fraction of positive predictions of $f$.

**Results.** The accuracy and statistical rate of **Err-Tol** and baselines for $\tau \in [0.7, 1]$ are reported in Figure 7. For both adversaries, **Err-Tol** attains a better false-positive rate than the unconstrained classifier (**Uncons**) for a small trade-off in accuracy. For adversary $A_{\text{TN}}$ (Figure 7(a)), **Uncons** has false-positive rate (0.75) and accuracy (0.65). In contrast, **Err-Tol** achieves a significantly higher false-positive rate (0.92) with accuracy (0.60). In comparison, **CHKV-FPR** has a higher false-positive rate (0.97) but lower accuracy (0.55); this accuracy is close to the accuracy of the all always-positive classifier. Compared to **Err-Tol**, **WGN+SW** has a higher accuracy (0.64) but a lower false-positive rate (0.87), and other baselines have an even lower false-positive rate ($\leq 0.82$) with accuracy comparable to **WGN+SW**. For adversary $A_{\text{FN}}$ (Figure 1(b)), **Uncons** has false-positive rate (0.75) and accuracy (0.67), while **Err-Tol** has a high higher false-positive rate (0.87) and accuracy (0.61). This significantly outperforms **WGN+SW** which has false-positive rate (0.76) and accuracy (0.64). **AKM** achieves the higher false-positive rate (0.92) with a natural reduction in accuracy to 0.59. **CHKV** achieves the higher false-positive rate (0.94) but with a lower accuracy (0.55). Meanwhile, **WGN+DRO** has a comparable false-positive rate (0.84) and a comparable accuracy at the same false-positive rate (0.65), and **KL** has a lower false-positive rate (0.81) and lower accuracy (0.62).

### E.3.4 Simulations with adversarial perturbations on the Adult data

In this simulation, we evaluate our framework on the Adult data [23] with the statistical rate fairness metric. The Adult data consists of rows corresponding to approximately 45,000 individuals, with 18 binary features and a binary class label that is 1 is the individual has an income greater than $50,000 USD and 0 otherwise. Among the binary features, we use gender as the protected attribute.

**Baselines.** Like the simulation with the statistical rate fairness metric on the COMPAS data (Section 5), we compare our framework with:

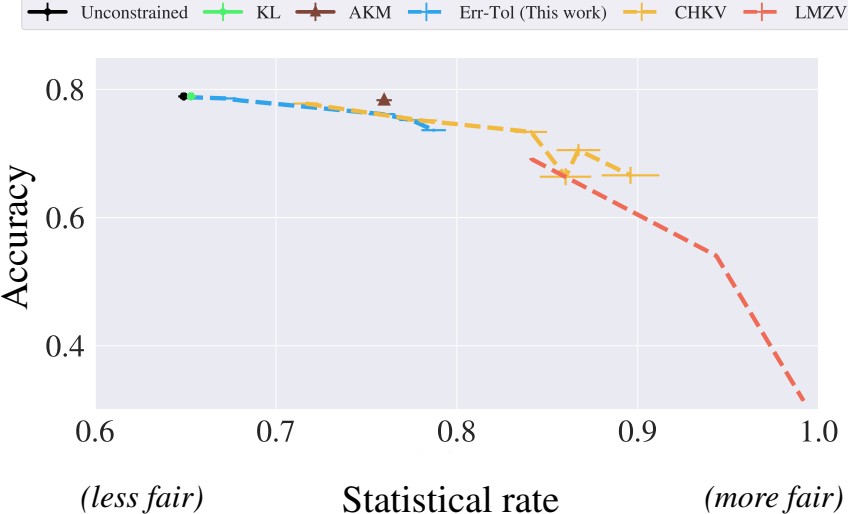

**Figure 8:** *Simulations on Adult data with gender as the protected attribute:* Let $A_{\mathrm{FP}}$ be the adversary that is the same as $A_{\mathrm{TN}}$, except that instead of perturbing the true negatives of $f^\star$, it perturbs the false positives of $f^\star$. The perturbed data is generated using adversary $A_{\mathrm{FP}}$ (as described in Section E.3.4) with $\eta = 1\%$. All algorithms are run on the perturbed data varying the fairness parameters ($\tau \in [0.7, 1]$ and $\delta_L \in [0, 0.1]$). The $y$-axis depicts accuracy and the $x$-axis depicts statistical rate (SR); both values are computed over the unperturbed test set. (Error bars represent the standard error of the mean over 100 iterations.) We observe that for our approach **Err-Tol** attains a better statistical rate than the unconstrained classifier **Uncons** with a small natural trade-off in accuracy. Further, **Err-Tol** achieves a better fairness-accuracy trade-off than **KL** and achieves a similar fairness-accuracy trade-off as **AKM, CHKV, LMZV.**

1. State-of-the-art fair classification frameworks for statistical rate under stochastic perturbations: **LMZV** [44] and **CHKV** [14].
2. **KL** [40], which controls for true-positive rate in the presence of a Malicious adversary.
3. **AKM** [6] that is the post-processing method of [32] and controls for equalized-odds fairness.
4. The optimal unconstrained classifier, **Uncons;** this is the same as [11]'s algorithm for PAC-learning in the Nasty Sample Noise Model without fairness constraints.

**Adversaries and implementation details.** We set $\eta = 1\%$ and consider an adversary $A_{\mathrm{FP}}$ that is the same as $A_{\mathrm{TN}}$, except that instead of perturbing the true negatives of $f^\star$, it perturbs the false positives of $f^\star$.[11] Note that positive labels are rare in the Adult data (e.g., less than 4% of the total samples are positive and annotated as women). Thus, an adversary with $\eta \geq 4\%$ can remove all positive samples from one protected group–thereby, changing the statistical rate on the perturbed data by an arbitrary amount. This suggests that corrupting the positive labels is a hard case for learning fair classifiers on the Adult data. The adversary tries to reduce the performance of $f^\star$ on $Z = 1$ (which is the rarer than $Z = 2$) in $\widehat{S}$ by removing the samples that $f^\star$ predicts as positive. Thus, decreasing $f^\star$'s statistical rate on $\widehat{S}$. All other implementation details were identical to the simulation with the COMPAS data in Section 5.

**Observations.** The accuracy and statistical rate (SR) of **Err-Tol** and baselines for $\tau \in [0.7, 1]$ and $\delta_L \in [0, 0.1]$ and averaged over 100 iterations are reported in Figure 8. We observe that **Err-Tol** attains better fairness than the unconstrained classifier **Uncons** at a small tradeoff to accuracy. Further, **Err-Tol** has a fairness-accuracy tradeoff that is better than **KL** and at least as good as **AKM, CHKV,** and **LMZV.** These observations are consistent with our observations on the COMPAS data in Section 5.

**Remark E.1.** *We also explored the effect of changing the perturbation rate $\eta$ over $\{0.5\%, 1\%, 1.5\%\}$. We report the results from this simulation in Figure 9. We observe that for all values of $\eta \in \{0.5\%, 1\%, 1.5\%\}$, **Err-Tol** achieves a higher statistical rate than the unconstrained classifier*

---

[11] We were unable to implement the analogous adversary $A_{\mathrm{TP}}$, that perturbs the true positives of $f^\star$, because on the Adult data, $f^\star$ does not have sufficient number of true positives with $Z = 1$.

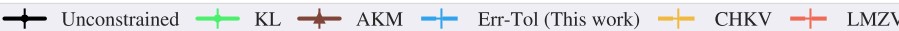

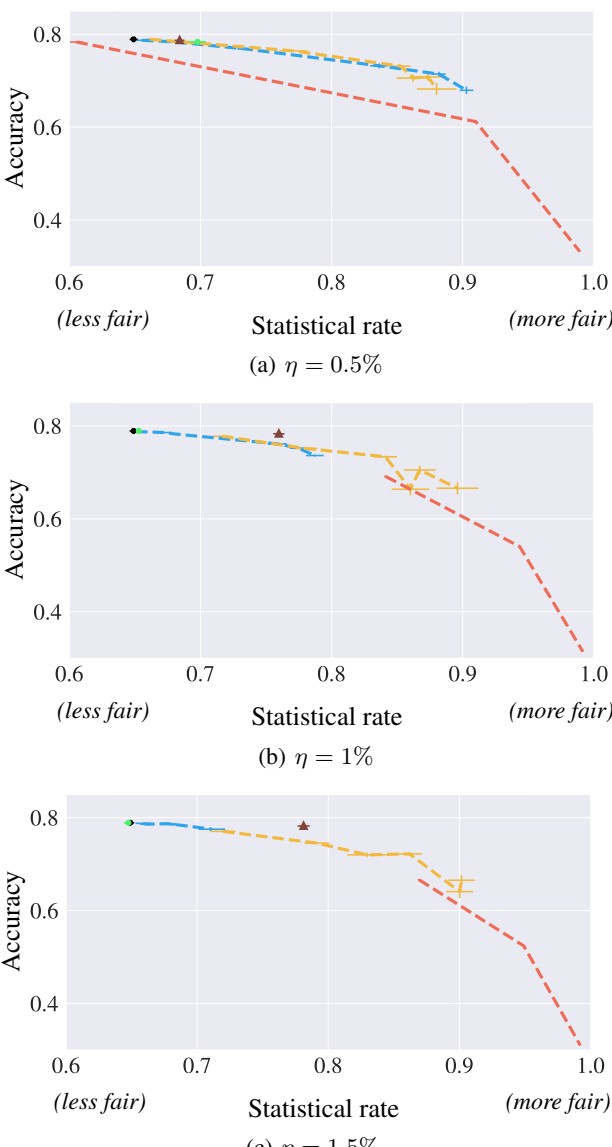

(a) $\eta = 0.5\%$

(b) $\eta = 1\%$

(c) $\eta = 1.5\%$

Figure 9: *Simulations on Adult data with adversarial noise on varying $\eta$ (see Section E.3.2):* In this the results obtained by repeating the simulation in Section E.3.4 by varying $\eta$ over $\{0.5\%, 1\%, 1.5\%\}$. The $y$-axis depicts the accuracy and the $x$-axis depicts statistical rate; both values are computed over the unperturbed test set. (Error bars represent the standard error of the mean over 50 iterations.) We observe that for all values of $\eta \in \{0.5\%, 1\%, 1.5\%\}$, **Err-Tol** achieves a higher statistical rate than the unconstrained classifier **Uncons** at a natural tradeoff to accuracy and **Err-Tol** has a similar (or better) fairness-accuracy tradeoff than other baselines. The only exception is **AKM**, which has a better fairness-accuracy tradeoff than **Err-Tol** at $\eta = 1.5\%$. For further discussion of the results, we refer the reader to Remark E.1.

**Uncons** *at a natural tradeoff to accuracy and* **Err-Tol** *has a similar (or better) fairness-accuracy tradeoff than other baselines. The only exception is* **AKM***, which has a better fairness-accuracy tradeoff than* **Err-Tol** *at $\eta = 1.5\%$. Further, we observe that the highest statistical rate of achieved by* **Err-Tol** *decreases as $\eta$ increases and this decrease is larger than the corresponding decrease in the statistical rate of* **AKM, CHKV,** *and* **LMZV***. We believe this is because $A_{\mathrm{FP}}$ is not the worst-case adversary (for this data), and hence, our approach, which "protects" against the worst-case adversary, outputs a "more robust" classifier which happens to have a low statistical rate on Adult data. In contrast, the prior works do not correct for the worst-case adversaries and are able to output "less robust" classifiers which happen to have a high statistical rate for this adversary, but may perform poorly with the worst-case adversary.*