# OpenReview forum: "Fair Classification with Adversarial Perturbations"
_NeurIPS.cc/2021/Conference — NeurIPS 2021 Poster_

### Official Review · Reviewer_PbrT · 2021-07-06

**Rating:** 7
**Confidence:** 4

**Summary:**

They consider the problem offFair classification in the adversarially perturbed setting — that is, the adversary gets to choose eta fraction of the sample arbitrarily and change their protected attributes arbitrarily.

They show an optimization problem that ensures fairness and accuracy guarantees under a mild assumption and show a mild assumption is necessary for those guarantees.

Finally, they have some experiments in which they consider two well thought out adversaries.

**Limitations And Societal Impact:**

Yes, the authors have addressed the limitations and potential negative societal impact of their work.

**Main Review:**

To my knowledge, this is the first paper that considers fair machine learning with a full adversary that can arbitrarily corrupt the protected attributes of the data unlike previous work that focused on stochastic corrpution.

The solution that the paper proposes is pretty intuitive and hence, it's very easy to follow. For instance, the overall paper is very clear in its description of the problem and its explanation of the solution/proof techniques (e.g. at the end of section 4).

However, the paper only shows an optimization problem that has nice guarantees (Thm 4.3) but doesn't actually show how to solve the optimization problem. More specifically, there isn't a discussion on how to actually solve ErrTolerant regardless of whether it's just a heuristics to solve the problem or an actual algorithm with guarantees. At a glance, the constraints seem convex so one can replace the objective term with a surrogate loss and use any convex optimization approach? In the experiment, it seems that the authors have "solved" ErrTolerant, so some discussion on how to solve and how they were solved in the experiment would be helpful.

Finally, the experiments do show to some extent that the proposed method in the paper is better or observe similar performance as other methods. However, I do have some questions:

-For the synthetic dataset set, why are other methods besides CHKV not tried out? Is it because it's hard to tune the parameters of the other algorithms to have a meaningful comparison?

-I think the easiest way to compare fair-accuracy tradeoffs is by drawing out the pareto curve, so is it possible to just try out much larger sets of parameters for all these algorithms to see a bigger pareto curve instead of the current one that seems to concentrate on a small region?

Questions/Comments on the impossibility results (Thm 4.4, 4.5, and D.1)
-Are the proofs of the impossibility results specific to SR or can one easily generalize the proofs to other fairness metrics?

-Maybe, because it's hard to parse the sentence in line 265 (when eta/gamma is small ... ), it's not clear to me how Thm 4.5 shows that the constants in Thm 4.3 are tight. It would be helpful to have a couple more sentences showing how they are tight.

-Lastly, how dependent are the proofs of Theorem 4.5 and D.1 on tau=1 which requires the fairness metric is exact for any group?





Minor questions/comments:

-Delta = O(blah, blah, blah). Is this supposed to be max with constants hidden?

-definition 4.7 (stability definition) is has not quantified the high probability.

-For adversaries in the Empirical results section A_TN and A_FN, how was f^* obtained?

Typos:
line 265. It's hard to understand what this sentence is saying because I think it's a run-on sentence?
line 270: learnable ⇒ learn

**Time Spent Reviewing:**

3

---

> ### Author Response · Authors · 2021-08-10
> **Review response**
>
> Thanks for your positive comments about this paper and for giving great suggestions!
>
>
> **“At a glance, the constraints seem convex so one can replace the objective term with a surrogate loss and use any convex optimization approach?”**
> Yes, if $\Omega$ is a *linear* metric (e.g., statistical rate, see Definition 3.1), then Prog ErrTolerant is a linear program in $f\in \mathcal{F}$ (where $\mathcal{F}:=\\{0,1\\}^{\mathcal{X}\times p}$). Although Prog ErrTolerant has $|\mathcal{X}|\times p$ variables (where $|\mathcal{X}|$ can be much larger than the size of the dataset), one can solve the dual of Prog ErrTolerant to find an $\varepsilon$-approximate solution to Prog ErrTolerant in $\mathrm{poly}(1/\varepsilon)$ time using standard methods such as stochastic gradient descent as in [15].
>
>
> However, when $\Omega$ is a *linear-fractional* metric (Definition 3.1), Prog ErrTolerant's constraints are *nonconvex*. In this case, building on prior work [15], one can give an algorithm that outputs an  $\varepsilon$-approximate solution of Prog ErrTolerant in $\mathrm{poly}(1/\varepsilon)$ time. At a high level, this algorithm approximates the (nonconvex) feasible region of Prog ErrTolerant up to “$\varepsilon$-error” using $O(1/\varepsilon)$ sets of linear constraints [Theorem 3.1, 15]. For each set of constraints, it finds a feasible classifier whose accuracy is $\varepsilon$-close to the optimal subject to satisfying these constraints, in $\mathrm{poly}(1/\varepsilon)$ time [Theorem 3.2, 15], and outputs the most accurate classifier found across all constraint sets (on the training dataset). Note that the output classifier satisfies at least one set of constraints, and so is $\varepsilon$-approximately feasible for Prog ErrTolerant.
>
> In more detail, each set of linear constraints (constructed by the algorithm) places lower and upper bound constraints on the numerator and denominator of $\Omega$ (see the equation above Line 157). Constraining the  numerator and denominator *separately* (instead of constraining their ratio) crucially ensures that the constraints are linear in $f$.
>
> We will add a discussion on the above algorithm and a proof of its convergence.
>
> **“Some discussion on ... how they were solved in the experiment would be helpful.”**
> We consider the general case (where $\Omega$ can be a linear-fractional constraint) in our implementation, and hence need to solve a constrained optimization problem with nonconvex constraints. For simplicity, we do not implement the algorithm mentioned above, and instead use existing optimization packages in our implementation.  As most of the baselines are implemented in Python, we chose the SLSQP solver in Python's SciPy package (see Section F.1.2). We will include these details in the "Empirical results" (Section 5).
>
>
>
>
> **"For the synthetic dataset set, why are other methods besides CHKV not tried out? Is it because it's hard to tune the parameters of the other algorithms to have a meaningful comparison?"**
> Yes, it is hard to tune the parameters of some methods to have a meaningful comparison. In particular, the implementations of AKM and LMZV (by the respective authors) do not take hyperparameters as input and do not seem to converge to $f^\star$, even on the *unperturbed* synthetic data. Thus, it did not seem fair to include these results in the paper and we removed such comparisons. However, in case you would like to take a look, the results can be found in [FigureC.png](https://github.com/fair-classification-with-perturbations/fair-classification-with-adversarial-perturbations/blob/master/FigureC.png)  in the anonymous Github repository. There we observe that ErrTol’s accuracy and statistical rate are similar to or better than other methods. The only exception is AKM. AKM has a slightly higher statistical rate (≈0.88) than ErrTol (≈0.78), but AKM’s accuracy (≈0.66) is significantly lower than ErrTol (≈0.99).
>
>
> **"I think the easiest way to compare fair-accuracy tradeoffs is by drawing out the pareto curve, so is it possible to just try out much larger sets of parameters for all these algorithms to see a bigger pareto curve instead of the current one that seems to concentrate on a small region?"**
> That's a great question. The problem is that even if we vary the fairness parameters ($\tau$ and $\delta_L$) on their full range ($[0,1]$), we get an identical plot: AKM, KL, and Uncons do not take the desired-fairness value as input, so they appear as points in Figure 1 and cannot be extended into a curve. For the remaining algorithms (CHKV, ErrTol, and LMZV), we vary the fairness parameters starting from the tightest constraints (i.e., $\tau=1$ and $\delta_L=0$) and relax the constraints until all algorithms’ *achieved* statistical rate matches the achieved statistical rate of the unconstrained classifier (this happens around $\tau=0.7$ and $\delta_L=0.1$). However, even when we set the fairness constraints as tight as possible ($\tau=1$ and $\delta_L=0$) the methods do not *attain* perfect fairness; this is why the Pareto curves in Figure 1 cannot be extended to the right. Additionally, relaxing the fairness constraints further is equivalent to the unconstrained problem (because the unconstrained classifier satisfies the fairness constraints for $\tau\leq 0.7$ and $\delta_L\geq 0.1$) and does extend the Pareto curves to the left. We will add a remark explaining this in the paper.
>
> **“Are the proofs of the impossibility results specific to SR or can one easily generalize the proofs to other fairness metrics?”**
> The proofs of impossibility results (Theorem 4.4, 4.5, and D.1) are specific to SR. It is not straightforward to extend our proofs to other fairness metrics. However, one can apply the general approach in Lines 318-331 to other fairness metrics by constructing a suitable set of distributions. We will include a discussion in the final version.
>
> **"It's not clear to me how Thm 4.5 shows that the constants in Thm 4.3 are tight.''**
> The constants in Theorem 4.3 are not tight. Theorem 4.3 gives a learner whose fairness guarantee is $O(\eta/\lambda)$ and error guarantee is $2\eta$. Theorem 4.5 shows that one cannot decrease the fairness guarantee in Theorem 4.3 (below $O(\eta/\lambda)$) without deteriorating the error guarantee from $2\eta$ to $(1/4)-\eta$. This follows because Theorem 4.5 proves that no learner can have a fairness guarantee smaller than $O(\eta/\lambda)$ if the learner's error guarantee is smaller than $(1/4)-\eta$. We will reword Line 265 to clarify this.
>
> **"Lastly, how dependent are the proofs of Theorem 4.5 and D.1 on $\tau=1$ which requires the fairness metric is exact for any group?"**
> Currently, our proofs of Theorems 4.5 and D.1 only work when $\tau$ is 1. We expect these theorems to hold for smaller values of $\tau$.
>
>
> **"Delta = O(blah, blah, blah). Is this supposed to be max with constants hidden?"**
> Yes, this is a maximum with a hidden constant. We will explicitly add the maximum in the expression.
>
> **"Definition 4.7 (stability definition) is has not quantified the high probability."**
> We will quantify the high probability in Definition 4.7. Currently, the formal definition appears as Definition B.5.
>
> **"For adversaries in the Empirical results section A_TN and A_FN, how was $f^\star$ obtained?"**
> We computed $f^\star$ by solving Program (1) using the unperturbed data set (which is available to the adversary). Program (1) is a special case of Program ErrTolerant (with $\lambda,\ \eta,\ \Delta\to 0$). Like Program ErrTolerant, we use the SLSQP solver in the SciPy package to heuristically solve Program (1) (Section F.1.2). We will add a remark about this in the paper.

---

> > ### Comment · Reviewer_PbrT · 2021-08-25
> > **review after the authors' rebuttal**
> >
> > Thanks for answering my questions! I'll keep my score at 7.

---

### Official Review · Reviewer_qeE2 · 2021-07-12

**Rating:** 6
**Confidence:** 3

**Summary:**

In this paper, the authors study fair classification in the presence of an adversary who is capable of choosing an arbitrary-fraction of the training samples and arbitrarily perturbing the corresponding protected attributes. Theoretically, the authors propose an optimization framework to learn fair classifiers with provable guarantees on accuracy and fairness in the adversarial settings. Empirically, the authors evaluate the classifiers produced by our framework for statistical rate on real-world and synthetic datasets against different adversaries.


**Limitations And Societal Impact:**

The authors have described the limitations of the work and discuss the potential negative societal impacts.

**Main Review:**


Overall, I think the paper is well written and the theoretical results are solid and impressive. The proposed problem setup is also important and practical. Assumption 1 also sounds reasonable to me.

However, there exists some problems in the empirical evaluations. First of all, I think the author should provide more justifications in the synthetic experiments. For example, I think the experiment results would be more convincing if the sizes of the protected group are varied by the authors. Besides, more different values of fairness constraint $\tau$ would be more convincing.

Moreover, if the authors could add one more dataset in the empirical evaluation, such as the Adult dataset, which has more data instances than COMPAS, the results would be more convincing.


**Time Spent Reviewing:**

4

---

> ### Author Response · Authors · 2021-08-10
> **Review response**
>
> Thanks for appreciating the theoretical results and the writing. We hope our response addresses your concerns with the empirical evaluation and that you will strengthen your support for the paper.
>
> **“Experiment results would be more convincing if the sizes of the protected group are varied”**
> Great suggestion. We performed a preliminary simulation that varies the size of protected groups and will include it in the final version. In this simulation, we varied the size of one group $\alpha\in [0,1]$ from 15% to 85% (and the size of the other group from 85% to 15%). We observed that varying $\alpha$ from 15% to 70% does not change the accuracy and statistical rate of the algorithms significantly (the accuracy changed by <5% and the statistical rate changed by <1%). However, as $\alpha$ approaches 85%, we observe that CHKV’s accuracy reduced by 15% (to ≈0.68) and its statistical rate increased by 4% (to ≈0.86). In contrast, ErrTol continued to have high accuracy (>0.99) and statistical rate (≥0.77), without large changes in either (its accuracy changed by <1% and statistical rate changed by <2%). The results can be found in [FigureA.png](https://github.com/fair-classification-with-perturbations/fair-classification-with-adversarial-perturbations/blob/master/FigureA.png) on the anonymous Github repository.
>
>
> **“More different values of fairness constraint would be more convincing''**
> We agree that this is an important parameter to evaluate; we consider different values of the fairness constraint $\tau$ in the real-world dataset (Lines 399-400). We also consider a different fairness metric (false-positive rate) for the real-world dataset (Lines 350-351 and Figure 7).
>
>
> **"If the authors could add one more dataset in the empirical evaluation, such as the Adult dataset, which has more data instances than COMPAS, the results would be more convincing."**
> We will add the Adult dataset to our empirical evaluation. In a preliminary simulation linked below, we use gender as the protected attribute, an adversary $A_{\mathrm{FP}}$ that perturbs the false positives of $f^\star$, and $\eta=1.25$%. In this simulation, we observe that ErrTol attains better fairness than the unconstrained classifier at a small tradeoff to accuracy. Further, ErrTol has a fairness-accuracy tradeoff that is better than KL and at least as good as  CHKV, AKM, and LMZV; the results can be found in [FigureB.png](https://github.com/fair-classification-with-perturbations/fair-classification-with-adversarial-perturbations/blob/master/FigureB.png) on the anonymous Github repository. These observations are consistent with our observations on the COMPAS data set. For the final version, we will also explore the effects of changing the value of $\eta$ and using race as the protected attribute.
>
> Note that positive labels are rare in the Adult data (e.g., less than 4% of the total samples are positive *and* annotated as women). Thus, an adversary with $\eta\geq 4$% can remove all positive samples from a protected group--thus, changing the statistical rate on the perturbed dataset by an arbitrary amount. This suggests that corrupting the positive labels is a *hard case* for learning fair classifiers on the Adult data. We would expect our results on $A_{\mathrm{TN}}$ or $A_{\mathrm{FN}}$ to be even better.

---

> > ### Comment · Reviewer_qeE2 · 2021-08-24
> > **Review after Reading Authors' Rebuttal**
> >
> > Thanks for your response. The added experiments results make the paper more convincing. I champion the acceptance of the paper.

---

### Official Review · Reviewer_bYbu · 2021-07-14

**Rating:** 7
**Confidence:** 3

**Summary:**

This paper presents an optimization framework to learn fair and accurate classifiers in the setting where an adversary can perturb the protected attributes. The contribution is mainly theoretical and focuses on Theorem 4.3, which gives guarantees on fairness and accuracy from a solution to their ErrorTolerant program, along with additional impossibility results (4.4, 4.5) which provide tightness and optimality guarantees for the proposed approach. There are also results from a small set of experiments on one simulated and one real dataset showing that the proposed approach generally outperforms baseline methods under the perturbation model.

**Update**: I have read the author response, and it addresses my concerns. Looking at the other reviews and the responses as well, I will keep my initial rating.

**Limitations And Societal Impact:**

Yes.

**Main Review:**

Overall: In general, while the paper is somewhat dense and covers a lot of ground, I found it to be a solid contribution to a problem of great relevance to the community -- versions of this particular problem has the potential to arise in many applied contexts. The paper is generally well-written and the theoretical contributions are made clear enough. I think that the paper could use some reorganization, but absent an issue with the proofs (which I was only able to briefly review) I think there is a good case for acceptance. Detailed comments are below.

Major Comments:

* In general, I found the paper difficult to read at times due to the confusing order of presentation. Many terms are used far in advance of their definitions, and the authors refer to definitions (such as 4.7) many pages before they are introduced. I would suggest to introduce all definitions earlier in the paper, in particular, and to introduce the model itself earlier as well. This will make it much easier to assess the contribution and understand what some ambiguous terms (such as "error") mean when used in context.

* The prior work section is also particularly difficult to read, since the authors mostly focus on immediately defending their territory relative to prior works. Please clearly describe the contributions of these prior works *separate* from your discussion of how the current work is different from them.

Minor Comments:

* The authors' claims about the fragility of imputation methods on L32 seem to me to assume a very specific imputatoin strategy (namely, one based on deep learning). This is certainly not the case for all imputation strategies, such as mean/mode/constant imputation which are sometimes used in practice. Please clarify your claim, or perhaps soften it.

* I am not sure whether SR is ever actually defined in the paper.

* It isn't clear from my reading how the authors actually find the solution to Program ErrTolerant in practice; it would be useful to comment more on how to solve this problem and give more details regarding complexity of this solution, particularly in the sections with experimental results on actual datasets.

Typos etc.

There are several minor typos in the paper; I only list a few here.

* L39: the distribution of --> the distributions of
* L70 -- could "performance" simply be error?
* L80-82 incomplete sentence
* L103: as measured \Omega --> as measured by \Omega

**Time Spent Reviewing:**

1.5

---

> ### Author Response · Authors · 2021-08-10
> **Review response**
>
> Thanks for appreciating the relevance of the problem and this paper’s contribution. We will further improve the paper by introducing the definitions and model earlier, and rewriting the related works section, as you suggest.
>
> **“The authors' claims about the fragility of imputation methods on L32 seem to me to assume a very specific imputation strategy (namely, one based on deep learning)."**
> Thanks, we will clarify that we mean deep-learning-like methods in Line 32.
>
> **"I am not sure whether SR is ever actually defined in the paper."**
> SR is defined in Lines 156-157. We will define SR earlier in the final version.
>
>
> **“It isn't clear from my reading how the authors actually find the solution to Program ErrTolerant in practice”**
> Thanks, we will include details on this in the final version; we summarize the idea here: If $\Omega$ is a linear metric (e.g., statistical rate, see Definition 3.1), then Prog ErrTolerant is a linear program in $f\in \mathcal{F}$ (where $\mathcal{F}:=\\{0,1\\}^{\mathcal{X}\times p}$). Although Prog ErrTolerant has $|\mathcal{X}|\times p$ variables (where $|\mathcal{X}|$ can be much larger than the size of the dataset), one can solve the dual of Prog ErrTolerant to find an $\varepsilon$-approximate solution to Prog ErrTolerant in $\mathrm{poly}(1/\varepsilon)$ time using stochastic gradient descent [15]. However, in the general case, where $\Omega$ is a linear-fractional metric, the constraint in Eq. (3) of Prog ErrTolerant may be nonconvex in $f\in \mathcal{F}$. Thus, to solve Prog ErrTolerant, we need a constrained optimization algorithm that can handle nonconvex constraints. To make our implementation (which will be public) more flexible, we implemented the general case. Because most of the baselines were implemented in Python, we use the SLSQP solver in Python's SciPy package (see Section F.1.2) in our implementation and empirical results. We will include these details in Section 5 (Empirical results).
>
> Note that SLSQP only offers local convergence guarantees, and so, may not converge to the optimal solution of Prog ErrTolerant. However, based on prior work [15], we could also give an algorithm to $\varepsilon$-approximately solve Prog ErrTolerant in $\mathrm{poly}(1/\varepsilon)$ time in the general case. Reviewer PbrT asked a related question; please see that response for more details about this approach. We will also add a discussion and a proof of its convergence to our final version.

---

### Official Review · Reviewer_EtYP · 2021-07-14

**Rating:** 6
**Confidence:** 3

**Summary:**

This work studies learning fair classification under the assumption that an arbitrary eta-fraction of training samples are perturbed on the protected attributes. The authors propose an optimization framework for eta-Hamming model which has the provable guarantees on accuracy and fairness. The experiments with linear models on synthetic and real-world datasets are used to evaluate the proposed approach.

**Limitations And Societal Impact:**

The authors have discussed the limitation of their work, but the further explanation about the above concerns are also needed.

**Main Review:**

Originality: The problem setting of this paper is a bit different from existing ones, that is the eta-fraction of perturbations are arbitrary. The main thought of the techniques seems  inspired by Richard 1950 that has been properly cited.

Quality: I am not knowledgeable enough to check the correctness of every theory result, so I can only basically understand what each theoretical result is used for.  By reading the whole paper, I have the following major concerns:
1. The authors claim their results extend to the case of perturbing both labels and features. This leads me to think about the connection between the proposed setting and eta-fraction over training sample labels. Is the latter one an alternative assumption for learning fair classifier?
2. I know targeting the worst-case adversary makes a strict guarantee on theory. However, for a real case the perturbation type is not known. From this view, should expectation formulation is a better choice for the generalization consideration?
3. I found that eta is a small value in experiments to ensure the optimal fair classifier. For works that are dealing with noisy feature/label, eta might be bigger than used in this work. Back to the setting, when and why is a small proportional perturbations for adversary reasonable?

Clarity: The submission is clearly written and well organized.

Significance: The theoretical results are interesting, while the experimental results seem only comparable. Generally, as the problem setting is not identical with the existing ones,  some modified methods are applied as baselines. But it is fine with me.


**Time Spent Reviewing:**

4h

---

> ### Author Response · Authors · 2021-08-10
> **Review response**
>
> Thank you for your feedback. We answer your questions below and hope that this will increase your support for the paper.
>
> **“This leads me to think about the connection between the proposed setting and eta-fraction over training sample labels. Is the latter one an alternative assumption for learning fair classifier?”**
> Sorry, we are not sure we understood the question. If you meant “are perturbations in labels a reason to learn fair classifiers?” then -- Yes. Undesired and systematic perturbations in the labels are an important reason to learn fair classifiers; e.g., see Blum and Stangl [12]. Please let us know if this was not what you meant and we would be happy to clarify further.
>
>
> **“Should expectation formulation is a better choice for the generalization consideration?,” i.e., are frameworks for stochastic perturbations a better choice?**
> Yes, in some cases, (e.g., when differential privacy techniques have been applied, so the perturbations are independent across samples and identically distributed within protected groups), frameworks for stochastic perturbations could be a better choice. However, in many applications, perturbations arise due to complex models. E.g., if protected attributes are imputed from images, the imputation errors are nonuniform within a protected group [14] and correlated across samples [51]. If iid-like assumptions are not met, frameworks for stochastic perturbations can output classifiers that violate their fairness and/or accuracy guarantees by a significant amount (see Theorem H.2 and Table 1). In these cases, frameworks for adversarial perturbations can be a better choice because they have stronger guarantees and potentially better empirical performance on accuracy and fairness.
>
>
> **"When and why is a small proportional perturbations for adversary reasonable?"**
> Perturbations in datasets arise due to various reasons [Olteanu et al., 2019]. Unintentional errors (such as annotation and duplication errors) arise along the data pipeline, and can affect a large fraction of samples [Ch 1, Venkatesh & Sharma 2013] [Fan 2015]. On the other hand, intentional errors are introduced by external adversarial actors (such as individuals misreporting their credit information). Because external actors only have access to a small fraction of the samples, one expects these errors to have a small perturbation rate; this is supported by prior work on adversarial label perturbations that has primarily focused on small perturbation rates ($\leq$ 10%); e.g., [Xiao et al. 2012].
>
> *Wenfei Fan. "Data quality: From theory to practice." ACM SIGMOD Record (2015).*
>
> *Venkatesh Ganti and Anish Das Sarma. "Data cleaning: A practical perspective." Synthesis Lectures on Data Management (2013).*
>
> *Han Xiao, Huang Xiao, and Claudia Eckert. "Adversarial label flips attack on support vector machines." European Conference on Artificial Intelligence (2012).*

---

> > ### Comment · Reviewer_EtYP · 2021-08-25
> > **Review after reading author's response**
> >
> > Thanks for answering my questions. I do not have any further concerns. I agree to accept the paper.

---

### Decision · Program_Chairs · 2021-09-27

**Decision:**

Accept (Poster)

**Comment:**

This paper provides novel theoretical results regarding fair classification in the presence of adversarial perturbations of the training examples, as well as supporting experiments on real world datasets. The reviewers found the problem formulation to be well motivated by application, and they deemed the theoretical results to be sound, novel, and interesting. Therefore, I recommend acceptance of the paper.

Throughout the discussion there are a number of concerns that should be addressed in the next version of the paper, and I strongly encourage the authors to do so. To name a few: (1) definitions and terminology should be defined early and clearly, (2) the additional experiments discussed in the responses should be added to the paper, and (3) the implementation strategy of solving the proposed programs in practice should be described clearly for the paper to be self-contained.